# Pheromone-based communication influences the production of somatic extracellular vesicles in *C. elegans*

Agata Szczepańska [1,5], Katarzyna Olek [2,5], Klaudia Kołodziejska[1], Jingfang Yu [3], Abdulrahman Tudu Ibrahim [1,4], Laura Adamkiewicz [1], Frank C. Schroeder [3], Wojciech Pokrzywa [2] ✉ & Michał Turek [1] ✉

Extracellular vesicles (EVs) are integral to numerous biological processes, yet it is unclear how environmental factors or interactions among individuals within a population affect EV-regulated systems. In *Caenorhabditis elegans*, the evolutionarily conserved large EVs, known as exophers, are part of a maternal somatic tissue resource management system. Consequently, the offspring of individuals exhibiting active exopher biogenesis (exophergenesis) develop faster. Our research focuses on unraveling the complex inter-tissue and social dynamics that govern exophergenesis. We found that ascr#10, the primary male pheromone, enhances exopher production in hermaphrodites, mediated by the G-protein-coupled receptor STR-173 in ASK sensory neurons. In contrast, pheromone produced by other hermaphrodites, ascr#3, diminishes exophergenesis within the population. This process is regulated via the neuropeptides FLP-8 and FLP-21, which originate from the URX and AQR/PQR/URX neurons, respectively. Our results reveal a regulatory network that controls the production of somatic EV by the nervous system in response to social signals.

Extracellular vesicles (EVs) are lipid bilayer-enclosed particles released by most cell types. Two major EV types can be distinguished based on their biogenesis: endosome-derived exosomes and membrane-derived ectosomes[1]. EVs can be employed by cells to remove unwanted biological material, such as misfolded proteins and damaged organelles, or to transport cargo, including proteins and nucleic acids, enabling exchange and communication between cells. Therefore, they are critical in multiple physiological processes and pathological states involving disrupted cellular homeostasis[2–5]. A recently discovered class of the largest membrane-derived evolutionarily conserved EVs, termed exophers, mediate both the elimination of waste from cells and the exchange of biological material between tissues - they were shown to play a significant role in cellular stress response, tissue homeostasis, and organismal reproduction[6–10]. Previous studies have spotlighted the role of exophers in safeguarding neuronal activity in *C. elegans* against proteotoxic stress by expelling cellular waste into surrounding tissues[7]. This mechanism is also mirrored in mammalian contexts, where the mouse cardiomyocytes release exophers loaded with defective mitochondria, thereby preventing extracellular waste accumulation and consequent inflammations, thus upholding metabolic balance within the heart[9]. However, the biological roles of exophers extend beyond the elimination of superfluous cellular components. In our previous work, we showed that the body wall muscles (BWMs) of *C. elegans* release exophers that transport muscle-synthesized yolk proteins to support offspring development, increasing their odds of development and survival[6]. The precise mechanisms by which external factors impact the regulation of exophergenesis, particularly in relation to animal development and reproduction, remain elusive.

[1]Laboratory of Animal Molecular Physiology, Institute of Biochemistry and Biophysics, Polish Academy of Sciences, Warsaw, Poland. [2]Laboratory of Protein Metabolism, International Institute of Molecular and Cell Biology in Warsaw, Warsaw, Poland. [3]Boyce Thompson Institute and Department of Chemistry and Chemical Biology, Cornell University, Ithaca, NY 14853, USA. [4]Faculty of Chemistry, Warsaw University of Technology, Warsaw, Poland. [5]These authors contributed equally: Agata Szczepańska, Katarzyna Olek. ✉e-mail: wpokrzywa@iimcb.gov.pl; m.turek@ibb.waw.pl

In nematodes, social behaviors are intricately linked to a variety of ascaroside pheromones. These small-molecule chemical signals play pivotal roles in inter-organismal communication and somatic tissue response of *C. elegans*. They regulate a spectrum of behaviors and developmental processes, including gender-specific attraction, repulsion, aggregation, olfactory plasticity, dauer formation, and maternal provisioning, functioning upstream of conserved signaling pathways[11–16]. The biosynthesis of ascarosides involves the activity of acyl-CoA oxidases (ACOXs) and acyl-CoA synthetases (ACSs), which are responsible for the production and modification of specific ascarosides[16–19]. The behavioral responses to ascarosides mirror the complexity of their molecular structures and interactions with diverse G-protein-coupled receptors (GPCRs), indicating a sophisticated network of receptor-specific responses[20]. In this context, our study aims to uncover how exophergenesis is influenced by these nuanced social signals, with a special emphasis on understanding the distinct contributions of pheromones from male and hermaphrodite *C. elegans*.

Our study reveals a complex regulatory network governing exopher production, influenced by both male and hermaphrodite pheromones. The most abundant male pheromone ascr#10 increases exopher production, primarily through the sensory neurons ASK, ADL, and AWB, mediated by the STR-173 GPCR in ASK neurons. Conversely, the predominant hermaphrodite pheromone ascr#3, reduces exopher production. Central to this system are the AQR/PQR/URX neurons, which are directly exposed to the body cavity and monitor the worm's body interior. These neurons release neuropeptides FLP-8 and FLP-21, crucial in downregulating exophergenesis in response to hermaphrodite pheromones, but not affecting the increase of exophergenesis triggered by male pheromones. Thus, exopher production emerges as an integrated response to internal states and external cues, finely balanced to adapt to varying environmental conditions. These findings provide insights into the physiological mechanisms in *C. elegans* and suggest potential conservation of these mechanisms across species.

## Results

### Male pheromones promote exopher generation in hermaphrodite muscles

Since muscle exophers mediate the transfer of maternal resources to offspring supporting their development, we hypothesized that exophergenesis (Fig. 1a) is regulated by metabolites-derived social cues generated within the worm population. We probed this by investigating how the presence of males could impact exopher production. To this end, we examined the number of exophers in *him-5* mutants, characterized by a marked increase in the percentage of males in the population (approximately 33%, compared to 0.3% in the wild type)[21]. Interestingly, *him-5* mutant hermaphrodites, co-cultured with *him-5* mutant males until the L4 stage and then transferred to a male-free plate (Fig. 1b), produce approximately 2.5 times more exophers than wild-type hermaphrodites grown on a male-free plate (Fig. 1c). This augmentation could be mediated by the previously postulated embryo-maternal signaling[6] as *him-5* mutant hermaphrodites contain 26% more embryos *in utero* than their wild-type counterparts (Fig. 1d). To exclude the possibility that the observed increase in exopher numbers resulted from the *him-5* mutation rather than male presence, we cultured wild-type hermaphrodites on plates conditioned with *him-5* mutant or wild-type males for 48 h, subsequently removing them (Fig. 1e). Growing hermaphrodites on male-conditioned plates increased exopher production (Fig. 1f and h) to the same degree as when hermaphrodites were grown with males until the L4 larvae stage (Fig. 1c), regardless of the *E. coli* variant used as a food source (Supplementary Fig. 1a). This elevation in exophers production also coincided with a rise in the number of *in utero* embryos (Fig. 1g), indicating that *C. elegans* male-secreted compounds, e.g., pheromones, can promote embryo retention in the hermaphrodite's uterus and increase

exophergenesis. Longer exposure to male-conditioned plates or co-culture with males beyond larval development did not further increase exopher production in *him-5* mutant hermaphrodites (Supplementary Fig. 1b–d). We next evaluated whether the rise in exopher production was linked to continuous or stage-specific, acute exposure to male secretions by growing hermaphrodites on male-conditioned plates during four 24-hour intervals (Fig. 1i). Results show that 24 h of exposure to male-conditioned plates is sufficient to increase exophergenesis in hermaphrodite, regardless of the life stage at which the exposure occurred (Fig. 1i). This implies that exposure to male secretions solely during worms' larval development can influence exopher level in adult hermaphrodites, with the most potent effect observed in animals exposed during the L4 larval stage to young adults' day 1 stage, mirroring the effects of continuous exposure to male secretions (Fig. 1i). In all experiments, except for exposure during L4 larval stage to young adults' day 1 stage, increased exopher production was observed to coincide with higher embryo retention in the uterus (Fig. 1j). Our data indicate that exophergenesis is finely modulated by perception of male-derived signals, possibly acting through embryo-maternal signaling.

### Elevated population density adversely affects exopher production

In light of our findings demonstrating an increase in exopher production in the presence of males within the population, we sought to examine the effect of hermaphrodite population density on this phenomenon. To this end, we established two distinct experimental conditions. In the first condition, hermaphrodites were cultured individually on plates, eliminating potential social cues. In the second condition, worms were raised as groups of 10 hermaphrodites per plate from the beginning of their development (Fig. 2a). In contrast to the effects of the presence of males, our results reveal that hermaphrodites grown at 10 hermaphrodites per plate consistently released fewer exophers compared to those grown as solitary animals (Fig. 2b, d). Both experimental groups contained an equivalent number of embryos *in utero* (Fig. 2c), suggesting that this modulation of exophergenesis occurs independently of embryo-maternal signaling. Furthermore, we observed that cultivating hermaphrodites in a population as small as five worms per plate was sufficient to decrease exopher production (Fig. 2e). Conversely, escalating the population size to one hundred animals per plate did not cause any further significant reduction in exopher production relative to 10 hermaphrodites per plate (Fig. 2f). Growing a single hermaphrodite on a plate conditioned by other hermaphrodites reduced exopher numbers to a level comparable to that observed at a density of 10 hermaphrodites per plate (Fig. 2g). Furthermore, raising individual adult hermaphrodites in the presence of larvae significantly decreased exopher production (Fig. 2h), highlighting the importance of population density, as even the presence of immature hermaphrodites can influence exophergenesis. The effect of hermaphrodites contrasts with the observed male-induced increase, underscoring a differential response to social cues from different sexes within the population.

### ascr#3 reduces, while ascr#10 enhances exopher production

In the intricate signaling network of *C. elegans*, ascaroside pheromones are pivotal for communicating population density and environmental states[22]. Notably, hermaphrodites predominantly secrete ascr#3, an ascaroside characterized by an α,β-unsaturated fatty acid, which plays a key role in signaling population levels[12]. Whereas, males secrete a wide range of sex-specific metabolites[23,24], including several members of the ascaroside family of pheromones, among which a dihydro-derivative of ascr#3 named ascr#10 is the most abundant[11]. This minor structural modification in ascr#10 compared to ascr#3 has significant implications for its signaling properties: hermaphrodites are highly responsive to male-produced ascr#10, whereas hermaphrodite-

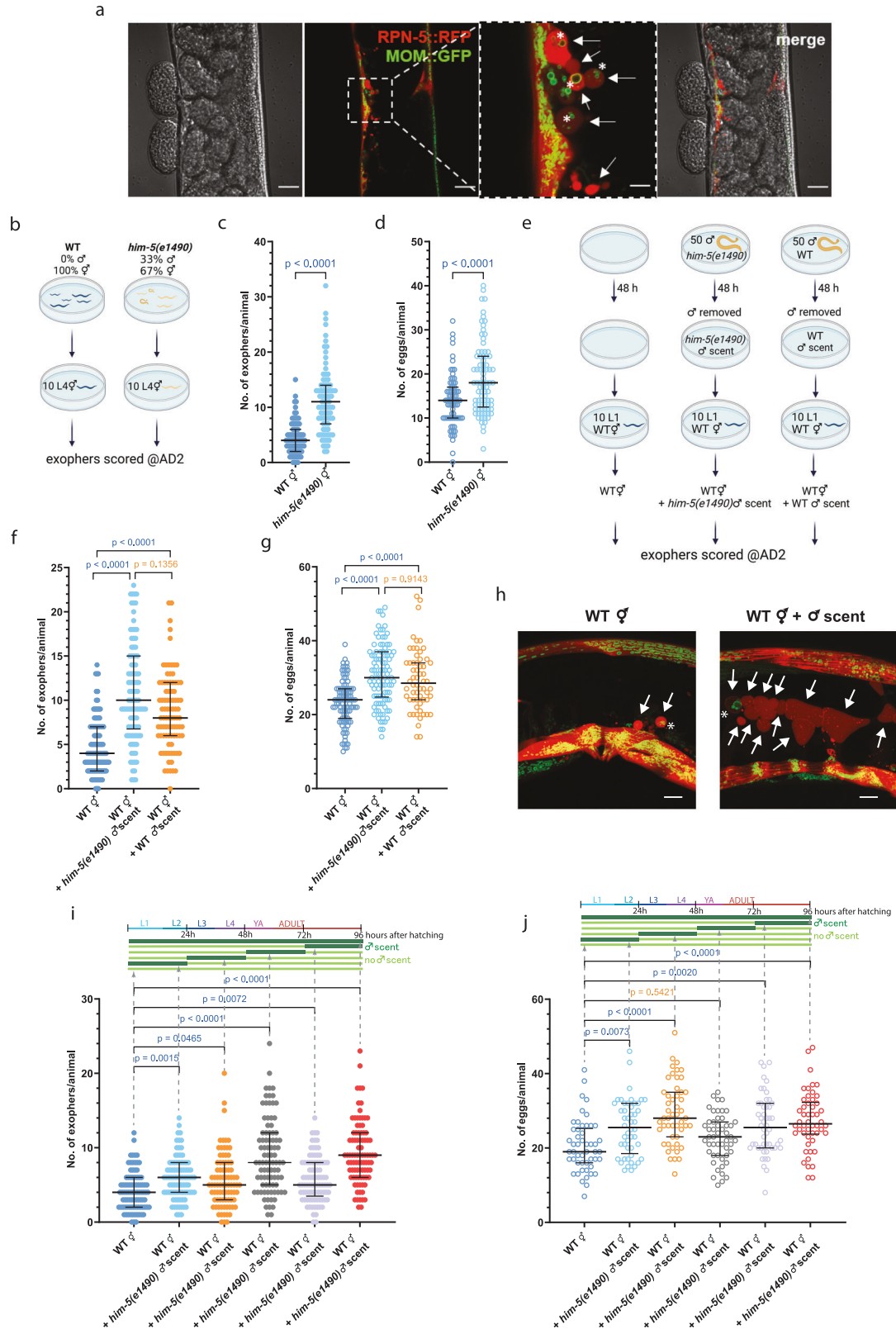

produced ascr#3 has a repellent effect on hermaphrodites and attracts males[11,22]. Based on our observation that male-conditioned plates significantly boost muscle exopher production in hermaphrodites, in stark contrast to the reduced exopher production on hermaphrodite-conditioned plates, we investigated the roles of ascr#3 and ascr#10 in exophergenesis.

We observed that applying ascr#3 at picomolar concentrations to plates containing worms with exopher reporter at the L4 stage resulted in a reduction of exopher production in adulthood day 2 hermaphrodites (Fig. 3a). Notably, this effect was absent when hermaphrodites were continuously exposed to ascr#3 from the egg stage to adult day 2 (Fig. 3b), suggesting a developmental stage-specific response to

**Fig. 1 | Production of *C. elegans* muscle exophers is regulated by male pheromones. a** *C. elegans* muscle exophers featuring mitochondria, shown with red (RPN-5::wrmScarlet proteasome subunit) and green (mitochondrial outer membrane::GFP) fluorescence. Arrows point to exophers, asterisks mark mitochondria-containing exophers (consistent in at least 25 animals from three replicates). **b** Experimental setup schematic: *him-5(e1490)* mutant males were co-cultured from the L1 stage until the L4 stage and then transferred to a male-free plate until AD2 for exopher count assessment. Created with BioRender.com. **c** Increased percentage of males in the population (via *him-5(e1490)* mutation) leads to a higher exophergenesis level in hermaphrodites. *n* = 79 and 83 worms, *N* = 3 independent experiments. **d** Increased percentage of males in the population (via *him-5(e1490)* mutation) causes embryo retention in hermaphrodites. *n* = 79 and 85 worms, *N* = 3 independent experiments. **e** Experimental setup schematic: 50 *him-5*/wild-type males cultured on a plate for 48 h, then removed. Subsequently, 10 hermaphrodites were transferred to these plates and grown until AD2 for exopher number assessment. Created with BioRender.com. **f** Growing wild-type hermaphrodites on *him-5(e1490)* or wild-type male-conditioned plates increases exophergenesis levels. *n* = 106, 98, and 89 (for respective columns), *N* = 3 independent experiments.

**g** Exposing wild-type hermaphrodites to *him-5(e1490)* or wild-type male secretome causes embryo retention in the uterus. *n* = 105, 98, and 60 (for respective columns), *N* = 3 independent experiments. **h** *C. elegans* hermaphrodites increase exopher production upon male pheromone exposure, shown with red (RPN-5::wrmScarlet proteasome subunit) and green (mitochondrial outer membrane::GFP) fluorescence. Arrows point to exophers, asterisks mark mitochondria-containing exophers (consistent in at least 25 animals from three replicates). **i–j** Exposure to male pheromones for 24 h triggers exophergenesis in hermaphrodites, regardless of the life stage at which the exposure occurs. Higher embryo retention in the uterus coincided with an increase in exopher production. **i** *n* = 90, 87, 85, 88, 89, and 90 worms (for respective columns), *N* = 3 independent experiments. **j** *n* = 50, 50, 48, 50, 50, and 50 (for respective columns), *N* = 2 independent experiments. Data information: Scale bars are (**a**) 20 µm and 5 µm (zoom in), **h** 10 µm. @AD2 means "at adulthood day 2". Data are presented as median with interquartile range; not significant *p* values (*p* > 0.05) are in orange color, significant *p* values (*p* < 0.05) are in blue color (**c–d**) two-tailed Mann–Whitney test, (**f–g**, **i**, **j**) Kruskal-Wallis test with Dunn's multiple comparisons test. Source data are provided as a Source Data file.

---

ascaroside signaling. Exposing worms to ascr#3 did not lead to changes in the number of *in utero* embryos (Supplementary Fig. 2a), which is consistent with the data we obtained for hermaphrodites grown under higher population density (Fig. 2c). Conversely, when exposed to low nanomolar levels of ascr#10, hermaphrodites exhibited a rise in exopher production (Fig. 3c, d), while the quantity of embryos in the uterus remained unchanged (Supplementary Fig. 2b). This increase occurred consistently, whether the worms were continuously exposed to ascr#10 from the egg stage to the second day of adulthood (Fig. 3d), or only from the L4 larval stage onwards (Fig. 3c).

Further investigations with worm mutants in which the biosynthesis of the ascaroside side chains via peroxisomal β-oxidation is perturbed[22] revealed complex interactions between ascaroside biosynthesis and exophergenesis. As shown in Fig. 3e, *acox-1(ok2257)*, *maoc-1(ok2645)*, and *daf-22(ok693)* mutants each exhibit distinct ascaroside profiles, including dramatically different amounts of ascr#3 and ascr#10[25]. Similar to the male co-culture experiments, we examined exophergenesis in reporter animals co-cultured with ascaroside biosynthesis mutant hermaphrodites (one reporter worm with nine mutant worms per plate, Fig. 3f). Co-culture with *acox-1(ok2257)* mutant hermaphrodites, in which levels of ascr#10 are dramatically increased relative to ascr#3, increased exopher production in the reporter strain. This suggests that a balance between ascr#3 and ascr#10 may be crucial for exopher regulation, with the presence of a higher concentration of ascr#10 overriding the inhibitory effect of ascr#3 (Fig. 3g). In contrast, *maoc-1(ok2645)* worms present a starkly different biochemical landscape, with very low levels of ascr#3, ascr#10, and other short-chain ascarosides, and an accumulation of long-chain ascarosides typically absent in wild type. Interestingly, when hermaphrodites are co-cultured with these worms, there is a reduction in exopher production (Fig. 3g). This could indicate that other metabolites accumulating specifically in *maoc-1(ok2645)* worms, e.g., side-chain hydroxylated medium-chain ascarosides[25] also affect exophergenesis. The *daf-22(ok693)* mutants add an additional layer of complexity. These worms do not produce the dauer-inducing and male-attracting ascarosides, including ascr#3 and ascr#10[25]. Yet, their presence does not affect exopher production in the reporter strain (Fig. 3g). This phenomenon could potentially be explained by the concurrent lack of both ascr#10, which has a stimulatory role, and ascr#3, which acts as an inhibitor. In the absence of these signaling ascarosides, a neutral impact on the exopher production pathway might occur, suggesting that the presence of either ascaroside is essential for activating the relevant regulatory processes. Furthermore, this situation may indicate a potential redundancy within the exopher production pathway. It is plausible that other metabolites accumulating specifically in *daf-22(ok693)* worms also affect

exophergenesis, thus preserving the equilibrium in regulatory mechanisms governing exopher formation. Additionally, the quantification of embryos within the reporter strain reveals no notable change in response to co-culture with the three different ascaroside biosynthesis mutants (Fig. 3h), indicating that the pheromone-driven modulation of exophergenesis in hermaphrodites primarily functions independently of the signaling pathways involved in embryo-maternal interactions.

## ASK, ASI, and ASH olfactory neurons regulate exopher production in hermaphrodite populations

Given the role of many olfactory neurons in detecting ascarosides[16] (Fig. 4a), we examined whether genetic ablation of all ciliated sensory neurons would abolish hermaphrodite pheromone-regulated exophergenesis inhibition. As shown in Fig. 4b, *che-13(e1805)* mutants, which do not form proper cilia and are incapable of pheromone detection[26], produce a minimal number of exophers (Fig. 4b). To initially determine the impact of sensory neurons on the basal production of exophers, both in the presence and absence of conspecific pheromones, we conducted an analysis of exopher numbers in nematodes where sensory neurons known for pheromone detection were genetically ablated[27]. We found that regardless of the growth conditions, the ablation of ASK, ADL, or AWC leads to significant suppression of exopher production, akin to *che-13(e1805)* mutants, while ASH ablation results in a modest reduction (Fig. 4c, Supplementary Fig. 3a). Notably, worms with genetically ablated neurons, which showed diminished exophergenesis, also displayed fewer eggs *in utero* (Fig. 4d). Considering the prominent effect on exophergenesis observed in worms with genetic ablation of AWC thermo-responsive neurons[28,29], we investigated the influence of temperature on exophers number. When growing worms at 15, 20, or 25 °C, we noted a temperature-related rise in exopher formation (Supplementary Fig. 3b). This increase was diminished in the absence of AWC, substantiating the temperature-dependent control of exophergenesis by these neurons (Supplementary Fig. 3b). Interestingly, the introduction of male-specific, pheromone-sensing CEM neurons in hermaphrodites (via *ceh-30* gain-of-function mutation[30]) did not result in changes in exopher levels when grown under higher population density (Fig. 4c) but caused a decrease in exopher production when animals were grown in isolation (Supplementary Fig. 3a), indicating a hermaphrodite-specific pathway for sensing pheromones that regulates exopher dynamics in response to other hermaphrodites.

Next, we sought to pinpoint the neurons crucial for downregulating exopher production in response to the presence of other hermaphrodites. We grew hermaphrodites with genetically ablated different classes of olfactory neurons either singly or at a population density of 10

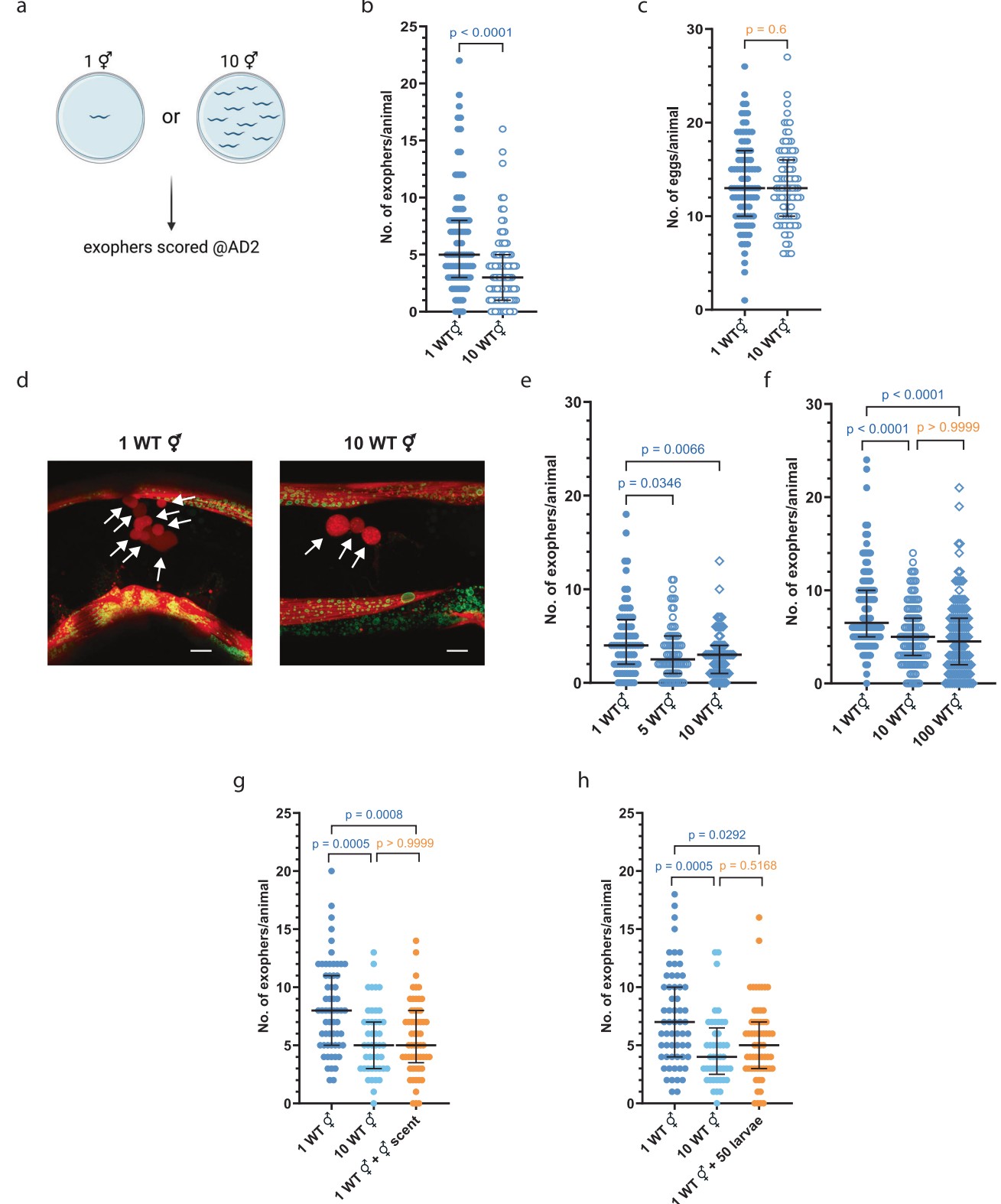

hermaphrodites per plate (Fig. 2a). Our analysis with strains showing impaired olfaction revealed that ASH, ASI, and ASK neurons are pivotal in exophergenesis downregulation in response to the presence of other hermaphrodites (Fig. 4e). We next investigated whether the observed effects were linked to a reduced number of embryos in the uterus of the reporter strain. We examined the embryo count in wild-type worms missing one of the key neurons for exopher regulation in

hermaphrodite populations, ASI, and also considered nematodes without AWB neurons, which did not show a block in exopher reduction at a higher population density, as a comparative control (Fig. 4e). Upon comparing the number of *in utero* embryos in wild-type, and ASI-, and AWB-ablated mutants, we observed a slight increase in wild-type and ASI-ablated worms, but no variation in worms lacking AWB neurons (Supplementary Fig. 3c). This suggests that the absence of decreased

**Fig. 2 | A rise in the population density of hermaphrodites adversely affects exophergenesis. a** Experimental design for determining the effect of the presence of other hermaphrodites on the number of exophers. Created with BioRender.com. **b** *C. elegans* hermaphrodites grown in the presence of other hermaphrodites produce fewer exophers than solitary animals. $n = 88$, and 90 worms (for respective columns), $N = 3$ independent experiments. **c** Embryo-maternal signaling is not responsible for the differences in exophergenesis levels as solitary and accompanied hermaphrodites contain the same number of embryos *in utero*. $n = 88$, and 90 worms (for respective columns), $N = 3$ independent experiments. **d** Representative image shows reduced exopher production in hermaphrodites from high-density populations, with red (RPN-5::wrmScarlet proteasome subunit) and green (mitochondrial outer membrane::GFP) fluorescence. Arrows indicate exophers (consistent in at least 25 animals across three replicates). **e, f** Cultivating as few as five hermaphrodites per plate reduced exopher production, with no

significant additional reduction observed when increasing the population to one hundred per plate. **e** $n = 76$, 70, and 75 worms (for respective columns), **f** $n = 100$, 112, and 124 worms (for respective columns), $N = 3$ independent experiments. **g** A single hermaphrodite on a plate previously occupied by other hermaphrodites yielded exopher numbers comparable to worms in the ten-hermaphrodite population. $n = 60$, 45, and 57 worms (for respective columns), $N = 3-4$ independent experiments. **h** Growing single adult hermaphrodite with developing larval hermaphrodites leads to a significant reduction in exopher production. $n = 60$, 45, and 61 worms (for respective columns), $N = 3-4$ independent experiments. Data information: Scale bars are 10 μm. Data are presented as median with interquartile range; not significant p values ($p > 0.05$) are in orange color, significant $p$ values ($p < 0.05$) are in blue color; (**b, c**) two-tailed Mann–Whitney test, (**e–h**) Kruskal–Wallis test with Dunn's multiple comparisons test. Source data are provided as a Source Data file.

exopher production in worms with ASI-ablated neurons at higher hermaphrodite densities is not related to impaired embryo production. Additionally, worms with ablated ASI and AWB neurons, even when grown with mutants lacking ascaroside side-chain biosynthesis, did not exhibit altered exopher numbers (Supplementary Fig. 3d, e). These findings indicate that the ASK, ASI, and ASH olfactory neurons are key in regulating exopher production within hermaphrodite populations.

## ASK, AWB, and ADL sensory neurons govern exophergenesis in response to males-social cues

Subsequently, we explored which olfactory neurons are essential for detecting male pheromones that enhance exophergenesis. By growing hermaphrodites with genetic ablations of different classes of olfactory neurons on *him-5* mutant male-conditioned plates (as shown in Fig. 1e scheme), we determined that the removal of ASK, AWB, or ADL neurons prevented the increase in the number of embryos *in utero* and nullified the enhancement in exopher production driven by male-emitted pheromones (Fig. 4f; Supplementary Fig. 3f). These results align with previously described roles for ASK, AWB, and ADL neurons in male pheromone sensing[12,31–33]. It is important to note that the influence of these neurons on exophergenesis appears to be tightly intertwined with the male pheromone-dependent accumulation of embryos in the uterus (Supplementary Fig. 3f), further highlighting the complex interplay between sensory neuron responses and social cues in modulating EV production by muscles.

## STR-173 receptor mediates pheromone-dependent exophergenesis modulation through ascr#10

Binding signaling molecules to the relevant receptor is the first step in transducing neuron chemosensory signals. Many of the over 1,300 GPCRs in *C. elegans* may contribute to chemosensory pathways[34]. Internal states and environmental conditions can modulate GPCRs expression to affect worm behavior[35–38]. To identify the receptor(s) responsible for ascaroside-mediated alterations in exopher formation, we performed RNA-sequencing of animals grown either as a single animal or at a density of 10 hermaphrodites per plate (Supplementary Fig. 4a). On the transcriptional level, neither group differed markedly (Supplementary Fig. 4b), and none of the detected GPCRs was significantly up-or down-regulated (Supplementary Table 1). However, *str-173* receptor transcript, which shows among all of the G protein-coupled receptors one of the most evident trends between growth conditions (2.3 fold change), is, according to single-cell RNA-seq data[39] expressed almost exclusively in ASK neurons (Supplementary Fig. 4c). Since we found that ASK is essential for pheromone-mediated modulation of exopher formation, we investigated the role of *str-173* in this pathway.

The wrmScarlet CRISPR/Cas9-mediated transcriptional knock-in for the *str-173* gene confirmed its strong expression during the late development and adulthood in ASK neurons and revealed additional expression in OLQ neurons, the pharynx, vulva muscles, and the tail

(Fig. 5a, b). Next, we created *str-173* null mutants again using CRISPR/ Cas9 editing (Supplementary Fig. 4d). We observed that *str-173* gene knockout neither influences the basal level of exophergenesis in *str-173* mutant hermaphrodites (Fig. 5c) nor robustly impacts population density-mediated exophergenesis decrease (Fig. 5d). Interestingly, contrary to wild-type worms, *str-173* mutants did not exhibit an increase in exopher production in response to male pheromones (Fig. 5e). Recognizing the potential function of STR-173 in male pheromone-mediated signaling, we investigated its participation in the ascr#10 recognition. Contrary to the increase in exopher production observed in wild-type animals, *str-173* mutants exposed to ascr#10 display a decrease in exopher production, emphasizing STR-173's role in translating male pheromone cues into physiological responses (Fig. 5f, g).

The observed decrease in exopher production in *str-173* mutants exposed to ascr#10 suggests the presence of another ascaroside receptor(s) capable of binding ascr#10 that plays an opposite role in exopher regulation to STR-173. Exposing worms with ablated ASK neurons to ascr#10 did not lead to significant changes in exopher counts (Fig. 5h), suggesting that receptor(s) that mediate ascr#10-dependent decrease in exopher production is also located in ASK neurons. Finally, considering the opposite effects of ascr#10 and ascr#3 on exophergenesis, we hypothesized that eliminating ascr#10's stimulatory impact on exophergenesis by ablating ASK neurons might enhance ascr#3's inhibitory effect. Indeed, we observed that worms with ablated ASK neurons produced fewer exophers when exposed to ascr#3, even in early development (Fig. 5i), while this effect in wild-type worms was only evident when exposure occurred from the L4 larval stage onwards (Fig. 3a, b).

## AQR, PQR, and URX neurons activity limit exophergenesis

Among the 118 classes of neurons in *C. elegans*, only four are directly exposed to the pseudocoelomic cavity[40]. Three classes of these neurons, AQR, PQR, and URX, regulate social feeding in worms[41]. Given that exophers are released to the worm's pseudocoelomic cavity and are regulated by social cues, we hypothesized that AQR, PQR, and URX might play a role in exophergenesis. To test for that, we first investigated the effect of genetic ablation of AQR, PQR, and URX neurons on exopher production. Indeed, the removal of these neurons leads to a substantial increase in exophergenesis (Fig. 6a). Notably, the increased number of exophers generated by worms with genetically ablated AQR, PQR, and URX neurons is not the result of boosted embryo-maternal signaling as these animals have a modestly reduced number of eggs *in utero* (Fig. 6b) and smaller brood size (Supplementary Fig. 5a, b) than wild-type control.

To further validate the role of AQR, PQR, and URX neurons in the regulation of exophergenesis, we optogenetically inactivated or activated them at both day 1 (AD1) and day 2 (AD2) of adulthood, using ArchT[42] or ReaChR[43], respectively, and compared the number of exophers before and after the stimulus. We observed that 60 min of AQR, PQR, and URX neurons inactivation at the AD2 stage leads to a

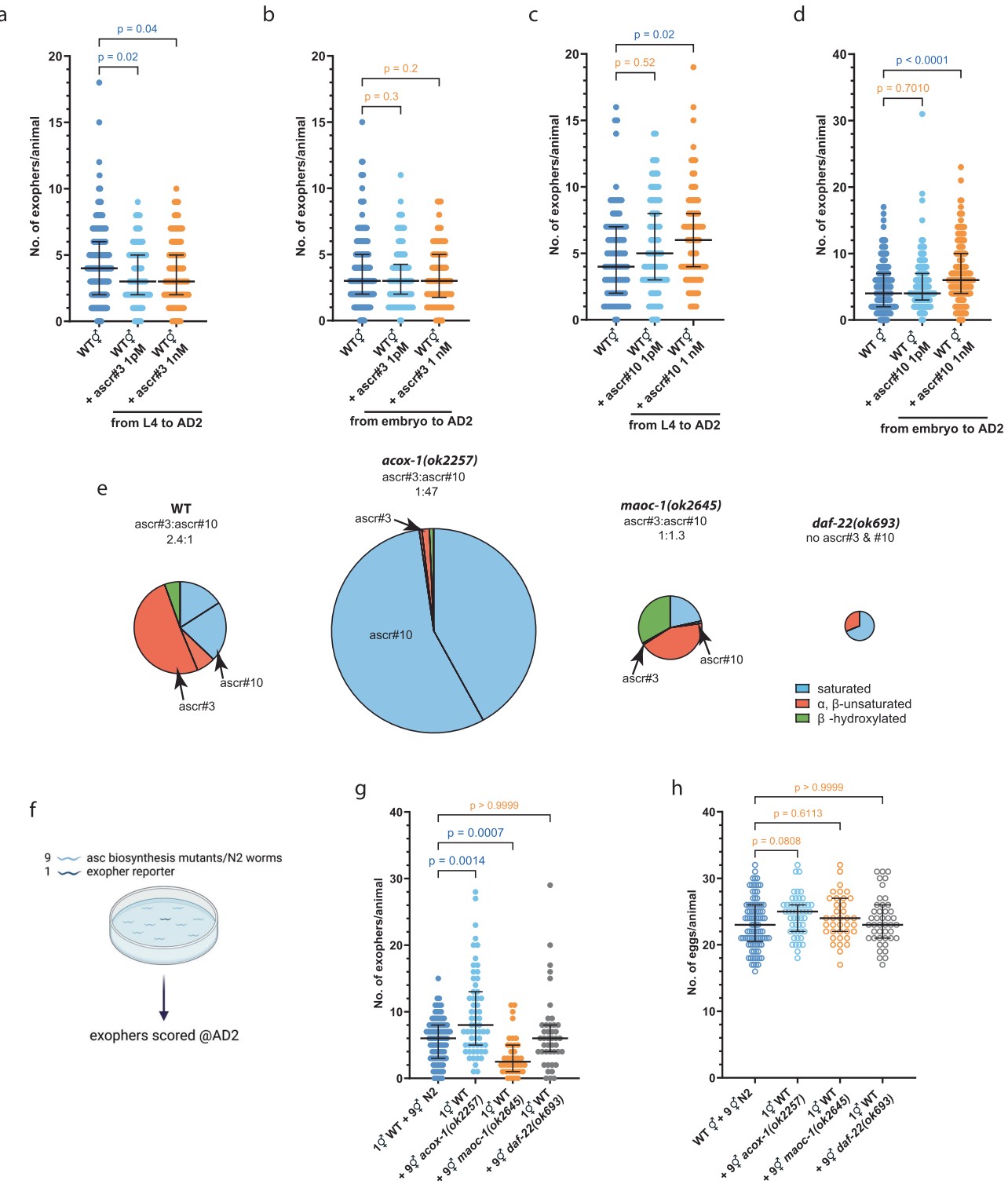

significant increase in exopher release (Fig. 6c and Supplementary Fig. 5c). On the other hand, 60 min of AQR, PQR, and URX neurons activation at the AD2 stage resulted in a significant decrease in exopher release after the stimulus was completed (Fig. 6d and Supplementary Fig. 5d). This modulation was evident at the AD1 stage for ReaChR-based activation but not ArchT-based inactivation, underscoring the potential significance of an optimal embryo count or direct uterus interaction (Supplementary Fig. 5e–h).

Moreover, the contrasting exophergenesis characteristics observed in ASK-ablated worms and those with genetic elimination of AQR, PQR, and URX neurons point to a similar exophergenesis pattern in wild-type worms and those deficient in all four neuron classes (Fig. 6e). We further determined that AQR, PQR, and URX neurons are involved in response to hermaphrodite pheromones (Fig. 6f), but not to male pheromones (Fig. 6g). Interestingly, the increase in exopher production induced by the inactivation of these neurons was still dependent on fertility (Fig. 6h, i). In summary, AQR, PQR, and URX neurons distinctly modulate exopher production in response to hermaphrodite pheromones, underscoring the complexity of neuronally-regulated reproductive signaling.

**Fig. 3 | The production of exophers by hermaphrodites is inhibited by ascr#3 and stimulated by ascr#10 pheromones. a** Hermaphrodites cultured from L4 to AD2 on NGM plates with 1 pM or 1 nM ascr#3 showed reduced exopher production. *n* = 222, 90, and 119 worms (for respective columns), *N* = 3 - 6 independent experiments. **b** Culturing hermaphrodites from embryos to AD2 with 1 pM or 1 nM ascr#3 on NGM plates did not affect exopher production. *n* = 149, 90, and 90 worms (for respective columns), *N* = 3 − 5 independent experiments. **c** Exopher production increased in hermaphrodites raised from L4 to AD2 on NGM plates with 1 nM ascr#10. *n* = 87, 85, and 79 worms (for respective columns), *N* = 3 independent experiments. **d** Raising hermaphrodites from embryos to AD2 on NGM plates with 1 nM ascr#10 led to increased exopher production. *n* = 190, 173, and 164 worms (for respective columns), *N* = 5–6 independent experiments. **e** Mutants deficient in the biosynthesis of short-chain ascarosides exhibit unique ascaroside pheromone profiles, characterized by low levels of ascr#3 and a varying ratio relative to ascr#10. Pie charts are based on the data from[25] that were obtained for mixed-stage

worm populations. Circle's diameter denotes the proportion of produced ascarosides relative to wild-type. **f** Schematic representation of the experimental setup for investigating the influence of ascaroside biosynthesis mutants on exophergenesis level in wild-type worms. Created with BioRender.com. **g** Growing wild-type hermaphrodites in the presence of *acox-1(ok2257)* (high ascr#10 levels) and *maoc-1(ok2645)* (low levels of both ascr#3 and ascr#10) mutants increase and decrease exopher production, respectively. *n* = 93, 43, 60, and 38 worms (for respective columns), *N* = 3 independent experiments. **h** Growing wild-type hermaphrodites with ascaroside biosynthesis mutants does not influence the number of embryos in the uterus. *n* = 93, 43, 50, and 37 worms (for respective columns), *N* = 3 independent experiments. Data information: @AD2 means "at adulthood day 2". Data are presented as median with interquartile range; not significant *p* values (p > 0.05) are in orange color, significant p values (*p* < 0.05) are in blue color; Kruskal-Wallis test with Dunn's multiple comparisons test. Source data are provided as a Source Data file.

## URX-expressed FLP-8 and AQR/PQR/URX-expressed FLP-21 neuropeptides inhibit exophergenesis

To elucidate the mechanism by which AQR/PQR/URX neurons regulate muscle exopher production, we used single-cell RNA-seq data[31], targeting neuropeptides predominantly expressed in these neurons with a limited expression in other cells. These criteria were fulfilled by URX-expressed FLP-8, and AQR, PQR, URX-expressed FLP-21 neuropeptides (Supplementary Fig. 6a–c). We could demonstrate that these neuropeptides negatively regulate exophers production, as evidenced by increased exophers counts in both single and double mutants (Fig. 7a), independent of embryo-maternal signaling (Fig. 7b). Moreover, FLP-8 and FLP-21 act downstream of URX and AQR/PQR/URX, respectively, as optogenetical activation of these neurons in the mutant context failed to suppress exopher release as observed in control animals (Fig. 7c–e, Supplementary Fig. 7a–c). Additionally, in a hermaphrodite-dense culture, the depletion of FLP-8 and FLP-21 did not result in reduced exopher numbers, contrary to wild-type worms (Fig. 7f). Finally, removal of these neuropeptides did not alter the increased exophergenesis seen in a reporter strain grown on male-conditioned plates (Fig. 7g). Both of these results are consistent with the pattern that was observed in worms without AQR/PQR/URX neurons (Fig. 6f, g) which indicates that FLP-8 and FLP-21, released by AQR/PQR/URX neurons, are integral in regulating exophergenesis, particularly in response to hermaphrodite signals.

## Model of exopher regulation mechanism in *C. elegans*

Our investigations have led us to construct the following model detailing the intricate mechanism of exopher regulation:

1. Male pheromones: Males produce ascarosides, including ascr#10, that increase exopher production levels. These signals predominantly act through ASK, ADL, and AWB sensory neurons. In this pathway, the ASK-expressed STR-173 G protein-coupled receptor facilitates the effect of ascr#10 in promoting exophergenesis. An increase in the number of embryos within the hermaphrodite, potentially mediated by male-released pheromones, can initiate additional pro-exophergenesis signals.
2. Hermaphrodite pheromones: Hermaphrodites release ascarosides, including ascr#3, that reduce exopher production levels. These signals predominantly act through ASI, ASH, and ASK sensory neurons.
3. Neuropeptide control: AQR/PQR/URX pseudocoelomic cavity-opened neurons release FLP-8 and FLP-21 neuropeptides that negatively regulate exophergenesis. This type of modulation is important for the decrease in exophergenesis dependent on hermaphrodite pheromones but not for the increase in reliance on male pheromones.
4. Integrated response: Exopher production is influenced by a mix of internal factors and external signals, including secretions from both males and hermaphrodites. This ensures that

exophergenesis adapts to a range of internal and external environmental conditions. See Fig. 8 for a detailed overview.

## Discussion

Our study shows that the regulation of exophergenesis in *C. elegans* presents a sophisticated network of neuroendocrine and pheromone signaling pathways that integrate internal and external environmental cues. Central to our findings are the opposite roles of ascaroside pheromones secreted by males and hermaphrodites. Male-derived ascr#10 significantly enhances exopher formation, via the GPCR STR-173 in the ASK neurons. Conversely, the predominant hermaphrodite-produced pheromone, ascr#3, downregulates exophergenesis. The ASH, ASI, and ASK olfactory neurons, along with the neuropeptides FLP-8 and FLP-21 from AQR, PQR, and URX neurons, play central roles in mediating the hermaphrodite pheromone-based regulation of exophergenesis. This study highlights the delicate equilibrium between environmental and social cues and the nematode's endocrine activities.

The notable increase in exopher production by hermaphrodites either exposed to male metabolites or grown with males underscores the significant influence of male pheromones on cellular processes. This observation aligns well with prior findings suggesting male pheromones channel resource allocation towards the germline[44]. In this context, it is conceivable that exophers are a part of this mechanism, possibly playing a pivotal role in bolstering oocyte and embryo quality. This strategy may enhance reproductive success, ensuring higher quality offspring even at the potential expense of the individual's somatic health. However, this reproductive advantage seems to come at a cost. The elevation in exophergenesis has been linked to decreased exploratory behavior, which could be associated with swifter deterioration of muscle function with age[6]. Aprison and Ruvinsky's observations that hermaphrodites expedite both development and somatic aging in the presence of males[39] further solidify this notion. This underscores the importance of understanding the role and regulation of exophergenesis, as they could serve as a nexus between environmental cues (like male pheromones), reproductive strategies, and age-related degeneration. While our study indicates that exophers are influenced by external cues such as pheromones, it is important to clarify that our current findings do not directly establish exophers as active agents of inter-animal communication. However, the modulation of exopher production by pheromones does hint at a potential indirect role in the broader context of nematode communication and behavior. In drawing parallels with other species like *Drosophila*, where EVs secreted by males are known to play a role in mating behavior and fertility[45,46], our study opens up the possibility of a similar communicative function for exophers. However, this remains a hypothesis that would require further investigation.

While the proximity of males induces an increase in exophergenesis, a heightened density of hermaphrodites elicits the

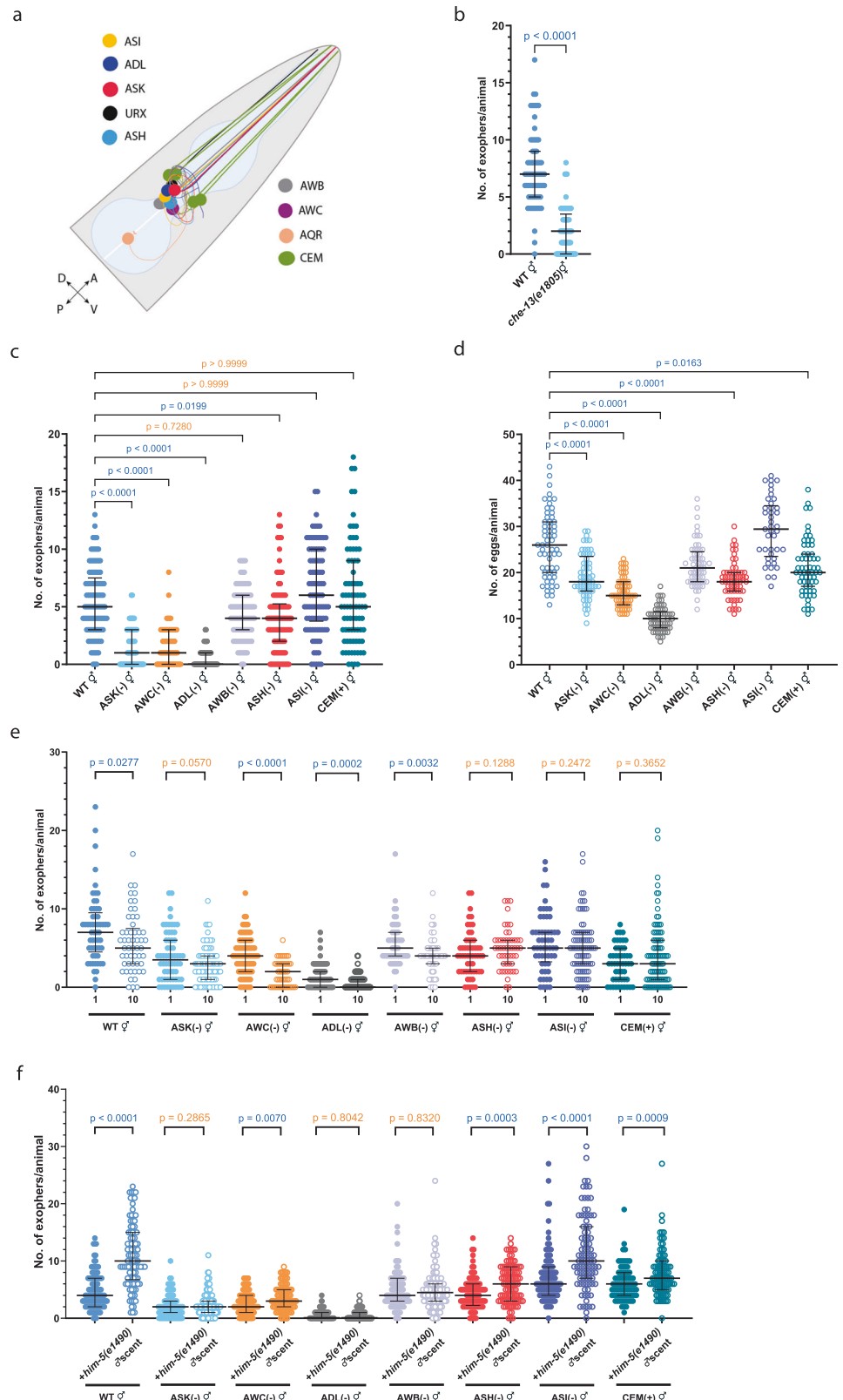

opposite response by diminishing it. There are instances where organisms, in response to densely populated environments, adaptively modulate their metabolic and reproductive functions to address potential resource scarcity[15,47]. Within the context of *C. elegans*, an escalation in hermaphrodite density could indicate imminent resource scarcity. Consequently, this prompts the nematodes to modify survival tactics, including downregulation of exophergenesis. Given the

complexity of cues – a supportive push from embryo-maternal signals and a dampening nudge from dense hermaphrodite surroundings – the resultant exopher production could integrate these counteracting signals. Future investigations should unravel these cues' hierarchical dynamics and identify prevailing signals under diverse conditions. For a comprehensive understanding of this facet, subsequent research should emphasize alterations in neural activity amidst different social

**Fig. 4 | Multiple olfactory neurons regulate exophergenesis levels in response to hermaphrodite and male pheromones. a** Sensory neurons investigated within this study. **b** Hermaphrodites with impaired ciliated sensory neurons exhibit reduced exophergenesis. *n* = 66 and 65 worms (for respective columns), *N* = 3 independent experiments. **c** Genetic ablation of ASK, AWC, ADL, or ASH neurons reduces exopher production in hermaphrodites grown in higher-density populations. *n* = 97, 61, 62, 90, 83, 90, 90, and 79 worms (for respective columns), *N* = 3 independent experiments. **d** Worms with genetically ablated neurons, resulting in reduced exophergenesis, also exhibited fewer eggs *in utero*. *n* = 61, 61, 61, 61, 61, 61, 41, and 61 worms (for respective columns), *N* = 2 - 3 independent experiments. **e** Decreased exophergenesis level due to exposure to hermaphrodite pheromones is mediated by ASH, ASI, and ASK neurons and can be altered by adding CEM male-specific neurons. A schematic representation of the experimental setup is presented in Fig. 2a. *n* = 57, 53, 62, 82, 84, 53, 88, 96, 49, 66, 55, 50, 56, 86, 50, and 88 worms (for respective columns), *N* = 3 independent experiments. **f** Increase in exophergenesis levels due to exposure to male pheromones is mediated by ASK, AWB, and ADL neurons. A schematic representation of the experimental setup is presented in Fig. 1e (without conditioning with WT males). *n* = 106, 98, 99, 108, 101, 100, 98, 106, 94, 98, 92, 95, 101, 90, 77, and 89 worms (for respective columns), *N* = 3 independent experiments. Data information: Data are presented as median with interquartile range; not significant p values (*p* > 0.05) are in orange color, significant *p* values (*p* < 0.05) are in blue color; (**b**, **e**, **f**) two-tailed Mann-Whitney test, (**c**, **d**) Kruskal–Wallis test with Dunn's multiple comparisons test. Source data are provided as a Source Data file.

scenarios and discerning the distinct signaling responses triggered by both male and hermaphrodite stimuli. Investigating how *C. elegans* balances the demand for supporting offspring development (via exophers) with the need to adapt to social stresses in crowded conditions can offer deeper evolutionary insights.

Our study has revealed a complex regulatory system in *C. elegans* that intricately connects ascarosides, GPCR, olfactory neurons, and the production of muscle EVs, all of which dynamically respond to environmental cues. Within this network, ascr#3 and ascr#10 emerge as key ascarosides in exopher regulation, yet their roles are part of a larger system where a variety of ascarosides and possibly other metabolites interact in a complex matrix. The diverse responses observed with different ascaroside biosynthesis mutants emphasize the nuanced interactions within this system, suggesting that a delicate equilibrium of various ascarosides concentrations, rather than the mere presence or absence of specific compounds, is crucial for exopher production. In this context, ascr#10 is linked with increased exopher production, modulated by the STR-173 receptor within ASK neurons. This finding contributes to the still limited knowledge surrounding GPCRs as ascaroside/pheromone receptors, considering the vast number of GPCRs expressed[48] and metabolites secreted by *C. elegans*[49]. The association between the STR-173 receptor and male ascr#10 hints at a receptor-ligand dynamic that may initiate a cascade of intracellular processes influencing exopher production, a pathway that requires further exploration. The complexity of the entire regulatory network is escalated by the fact that the lack of activity of one receptor, achieved either by transcriptional downregulation or post-translational modification, can drastically alter the biological potential of its ligands. This was demonstrated in our analysis, where knockout of *str-173* reversed the pro-exopher potential of ascr#10, implying that multiple receptors may transduce different physiological responses to a single compound. Identifying signaling components antagonizing exopher production will be pivotal in understanding the full spectrum of regulatory effects mediated by ascarosides and their receptors. Additionally, our study exemplifies the multilayered interplay between pheromones and cellular and organismal responses. The significant increase of embryos in the hermaphrodite's uterus upon exposure to the male secretome, contrasting with the lack of egg retention induced by ascr#10, shows that male pheromone-induced exopher production can be achieved with and without increased *in utero* egg retention. This also suggests that ascr#10 is just one among multiple small molecule signals secreted by males that influence exopher formation, potentially with or without engaging embryo-maternal signaling pathways. Furthermore, exploring how ascr#10- and ascr#3-mediated signaling interacts with other male-derived signals may uncover new pathways and responses. In conclusion, our findings underscore the necessity of additional research into the complex interactions among metabolites (pheromones/ascarosides), receptors, and intracellular processes, particularly their modulation by social and environmental factors, to enhance our understanding of the mechanisms controlling exophergenesis in *C. elegans*, which may

also offer insights into similar cellular processes and environmental adaptations in other organisms. Our study sheds light on the function of specific sensory neurons AQR, PQR, and URX and their associated neuropeptides, FLP-8 and FLP-21, in modulating muscle exopher production. The impact of FLP-21 on socially-driven exophergenesis, in particular, resonates with its function in modulating social interactions[50]. Moreover, neuropeptides, such as FLP-18, are involved in controlling muscle function and worm behavior. Specifically, the co-release of FLP-18 with tyramine from the RIM neurons and its targeting of body wall muscles via the NPR-5 receptor to modulate muscle excitability underscores the multifaceted interactions of neuropeptides and neurotransmitters in these processes[51]. Given this, it is plausible to assume that FLP-8 and FLP-21 may function along similar pathways. Their involvement might influence muscle excitability in a manner that subsequently impacts the release dynamics of EVs. Exploring the precise receptors and pathways through which FLP-8 and FLP-21 act will be instrumental in further elucidating their specific roles in exophers release and muscle functionality.

Understanding the mechanisms behind exopher encapsulation of damaged organelles, e.g., mitochondria, or aggregated proteins in *C. elegans* could provide valuable insights into protein aggregation and clearance processes. This nematode model, with its unique neuropeptides and neuronal populations, may reveal fundamental principles applicable to broader biological contexts. For instance, if similar mechanisms are discovered in more complex organisms, this could enhance our understanding of neurodegenerative diseases such as Alzheimer's or Parkinson's, where protein aggregation plays a key role. Furthermore, our research highlights the important role of the worm's olfactory system in controlling muscle exophergenesis. Intriguingly, in humans, olfactory impairments have been linked to an increased risk of cardiovascular disease (CVD)[52,53]. This association suggests that disruptions in processes similar to exopher-driven homeostasis in the heart[9] might be occurring. However, verifying the presence of a similar olfactory-based mechanism for regulating exophers in mammals requires further investigation.

## Methods
### Worm maintenance and strains
Worms were maintained on nematode growth medium (NGM) plates seeded with bacteria *Escherichia coli* OP50 or HT115 at 20 °C unless stated otherwise[54]. A list of all chemicals used in the study can be found in Supplementary Data 1. A list of all *C. elegans* strains used in the study can be found in Supplementary Data 2.

### Scoring exophers and fluorescence microscopy
For the assessment of exophers number, the protocol described in Turek et al.[6] and Banasiak et al.[55] (Bio-Protocol, detailed, step-by-step guide) was applied. Briefly, exophers were scored using stereomicroscope Zeiss Axio Zoom.V16 with filter sets 63 HE and 38 HE. For each exopher scoring assay, worms were age-synchronized either from pretzel-stage embryos, or larval stages L1 and L4. On adulthood day-2,

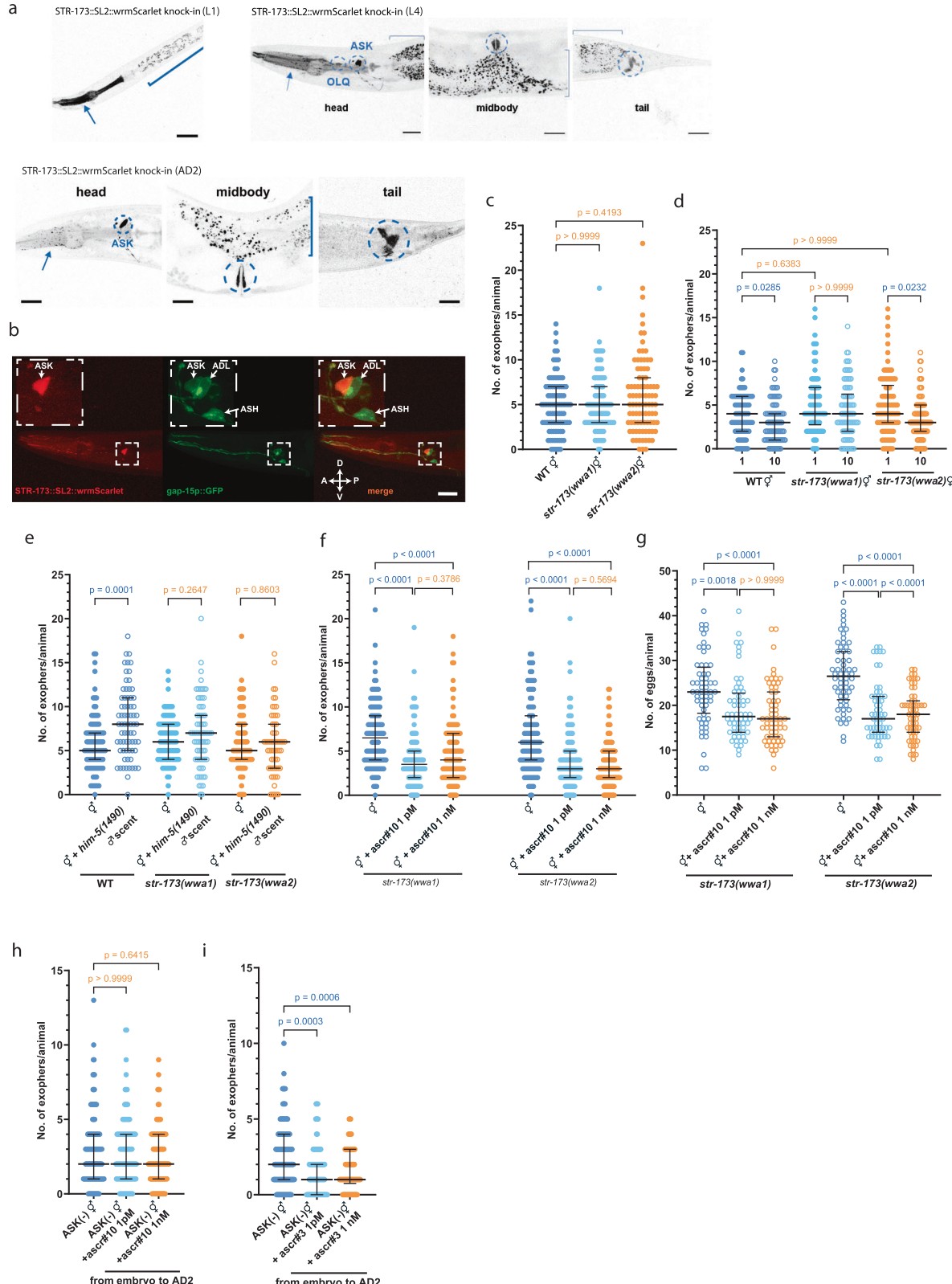

animals were visualized on NGM plates, and the number of exophers was counted in each freely moving worm.

The representative pictures presented in the manuscript were acquired with an inverted Zeiss LSM800 laser-scanning confocal microscope with a 40x oil immersion objective. To excite the GFP and RFP fluorescent proteins, 488- and 561-nm lasers were used. For visualization, animals were immobilized on 3% agarose pads with 6 μl

of PolySciences 0.05 μm polystyrene microspheres or 25 μM tetramisole.

**RNA interference assay**

RNA interference in *C. elegans* was performed using the standard RNAi feeding method and RNAi clone[56], previously described in Turek et al.[6]. Briefly, NGM plates, supplemented with 12,5 μg/ml tetracycline,

**Fig. 5 | The STR-173 receptor facilitates the pheromone-dependent modulation of exophergenesis via ascr#10. a, b** The *str-173* 7TM receptor is expressed in neurons (ASK and probably[#] OLQ) and non-neuronal tissues (pharynx marked with an arrow, vulva, and probably[##] rectal gland marked with circles). Square brackets mark gut autofluorescence. [#]based on the position and scRNAseq data[39] [##]based on the position and shape, consistent in at least 20 animals from three replicates. **c** Exophergenesis level in *str-173* mutants co-cultured with other hermaphrodites is unchanged compared to wild-type worms. *n* = 90 worms, *N* = 3 independent experiments. **d** Knockout of the *str-173* gene does not robustly affect the decrease in exophergenesis mediated by changes in population density. *n* = 90, 90, 86, 90, 86, and 90 worms (for respective columns), *N* = 3 independent experiments. **e** Increase in exophergenesis levels due to exposure to male pheromones depends on the STR-173 7TM receptor. *n* = 90, 70, 90, 72, 90, and 60 worms (for respective columns), *N* = 2 - 3 independent experiments. **f, g** When exposed to ascr#10, *str-173*

mutants exhibit a reduction in exopher production and *in utero* embryo number, underscoring the importance of STR-173 in mediating ascr#10 function in exophergenesis regulation. **f** *n* = 106, 104, 113, 107, 104, and 107, *N* = 4 independent experiments; **g** *n* = 60 worms, *N* = 2 independent experiments. **h** Removal of ASK neurons abolished the effect of ascr#10 on exophergenesis. *n* = 135, 109, and 117 worms (for respective columns), *N* = 3 independent experiments. **i** Removal of ASK neurons potentiates the inhibitory effect of ascr#3 on exophergenesis. *n* = 150, 90, and 90 worms (for respective columns), *N* = 3-5 independent experiments. Data information: Scale bars are (**a**) 30 μm and (**b**) 20 μm. Data are presented as median with interquartile range; not significant *p* values ($p > 0.05$) are in orange color, significant *p* values ($p < 0.05$) are in blue color; (**c, d, f–i**) Kruskal–Wallis test with Dunn's multiple comparisons test, (**e**) two-tailed Mann-Whitney test. Source data are provided as a Source Data file.

---

100 μg/ml ampicillin, and 1 mM IPTG, were seeded with *E. coli* HT115 bacteria containing dsRNA against the gene of interest. Control group animals were fed with bacteria containing the empty vector L4440. Age-synchronized pretzel-stage embryos or L1 larvae were placed on freshly prepared plates and cultured until day 2 of adulthood, when exophers were scored. The age of the worms was verified at the L4 larvae stage, either younger or older worms were removed from the experiment.

### Pre-conditioning of the plates with *C. elegans* males
50 males from the strain with *him-5(e1490)* mutation were transferred on a fresh, 35 mm NGM plate with *E. coli* OP50 or HT115 at the L4 developmental stage and older. After 48 h, males were removed from the plates.

**Culturing worms from L1 to adulthood day 2 on plates pre-conditioned with males.** L1 larvae were transferred to plates preconditioned with males in a group of 10 hermaphrodites per plate. For the control group, L1 larvae were transferred to fresh, 35 mm NGM plates seeded with *E. coli* OP50 or HT115 without pre-conditioning with males. Worms were cultured up to adulthood day 2 when muscle exophers were counted in each animal.

**Culturing worms in 24 h intervals from L1 to adulthood day 2 on plates pre-conditioned with males.** Worms were kept on plates preconditioned with males only for 24 h at different developmental stages. All hermaphrodites were picked at L1 stage and grouped into 10 worms per plate population. The first group was cultured on preconditioned plates for 24 h, starting from hatching from the egg. After 24 h, worms were transferred to a fresh NGM plate seeded with OP50 bacteria. The second group was grown for 24 h after hatching from the egg on the NGM plate, then the worms were moved to preconditioned with males plates for 24 h, after that, they were transferred to fresh NGM plates seeded with OP50 bacteria. The third group was cultured for the first 48 h on NGM plates, then transferred to plates preconditioned with males for 24 h, and then replated on fresh NGM plates. The fourth group was cultured from the L1 stage for 72 h on standard NGM plates, after which they were transferred to plates preconditioned with males for 24 h. All experimental groups were cultured from L1 to adulthood day 2 when muscle exophers were counted.

### Egg retention assay
Firstly, using the stereomicroscope, muscle exophers were counted in each worm. In the following step, hermaphrodites were exposed to a 1.8% hypochlorite solution. When they dissolved, embryos retained in the uterus were counted.

### Metabolic inactivation of bacterial food source
The preparation of plates with a metabolically inactive food source for the worms to determine its effects on exophergenesis was done

according to the protocol described in Beydoun et al.[57]. Briefly, a single *E. coli* HT115 or OP50 colony was inoculated overnight and then the bacterial culture was split into two flasks. Next, paraformaldehyde (PFA) was added to a final concentration of 0.5% in only one of them, and the flask was placed in the 37 °C shaking incubator for 1 h. Afterward, the aliquots were transferred to 50 mL tubes, centrifuged at approximately 3000 × g for 20 min, and washed with 25 mL of LB five times. Control and PFA-treated bacteria were later concentrated accordingly and seeded on the NGM plates.

As control of PFA treatment, new bacterial cultures in LB were set in the 37 °C shaking incubator overnight to ensure the replication was blocked and there was no bacterial growth.

Plates were used for experiments at least 5 days after their preparation to ensure no bacteria-derived metabolites were left on plates with PFA-treated *E. coli*. A new batch of bacterial food source was prepared for each biological repetition.

### Brood size quantification
Age-synchronized hermaphrodites at the L4 developmental stage were picked for the experiment. Worms were cultured as single worms per 60 mm NGM plate, seeded with *E. coli* OP50 or HT115. Ten worms were counted as one biological repeat. Worms were transferred to fresh plates every day from adulthood day 1 until the end of the egg-laying period.

The number of eggs laid over the worms' reproductive lifetime was counted manually daily. The data is presented as the total number of eggs laid by each animal.

### Quantifying the number of exophers in worms grown in different temperatures
Worms were confronted with a range of physiological temperatures: low 15 °C, optimal 20 °C, and high 25 °C throughout their development until exophers were scored. To compare the corresponding stages of development at various temperatures, worms were additionally sorted at the L4 stage. The exopher number was assessed based on the timepoint of maximal egg laying, approx. 140 h or 78 h from egg hatching at low or high maintenance temperatures, respectively. Timing calculations were based on the *C. elegans* development timeline at different temperatures[58].

### Culturing worms at different population densities
L1 larvae were transferred to 35 mm NGM Petri dishes seeded with *E. coli* OP50 or HT115 bacteria strains (100 μL of bacteria from 10 mL overnight culture grown in 50 mL Erlenmeyer flask) immediately after hatching as a single larva or in a group of 5, 10 or 100 animals. In total, thirty plates with single worms, six plates with 5 worms, three plates with 10 worms, and one plate with 100 worms were used per biological repeat. Exophers were scored on adulthood day 2 in the worms expressing RFP in BWM.

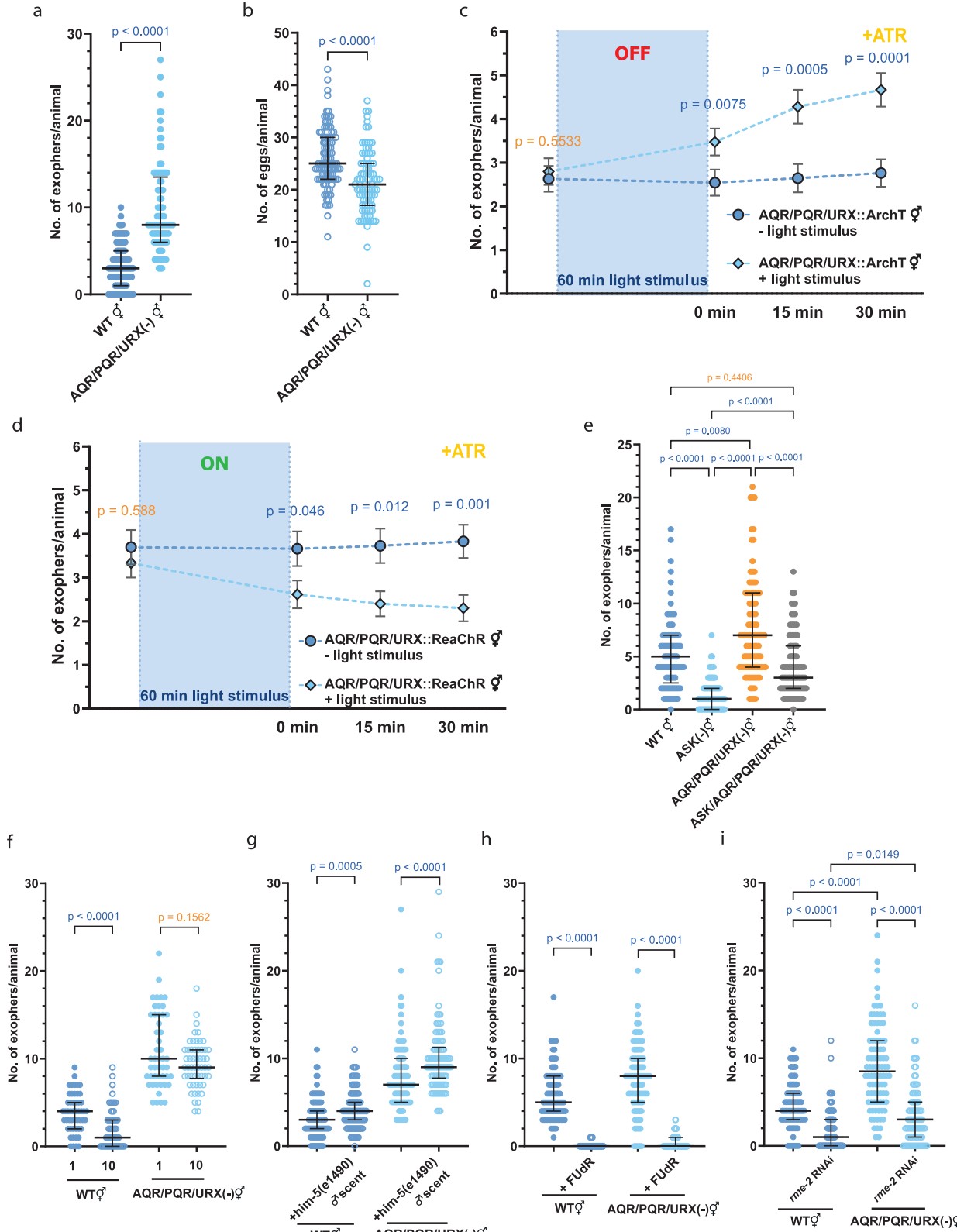

### Quantifying the influence of excreted metabolomes of ascaroside biosynthesis mutants on exopher production in wild-type worms

Nine freshly hatched L1 larvae from selected ascaroside biosynthesis mutant strains (*maoc-1(ok2645), daf-22(ok693)*, or *acox-1(ok2257)*) or wild-type hermaphrodites (as a control) were transferred to fresh NGM plates. Next, one L1 larva of a reporter strain expressing RFP in BWM was added to each plate and 10 worms in total were grown together on 35 mm Petri dish seeded with *E. coli* OP50 or HT115 bacteria strains. On adulthood day 2 the number of exophers was quantified in the worms expressing RFP in BWM, according to description.

**Fig. 6 | Pseudocoelom-exposed neurons negatively regulate exopher production. a** Genetic ablation of pseudocoelom-exposed AQR, PQR, and URX neurons increases exopher production. $n = 100$ and 89 worms (for respective columns), $N = 3$ independent experiments. **b** Genetic ablation of pseudocoelom-exposed AQR, PQR, and URX neurons causes a decrease in the number of embryos present in the uterus. $n = 100$ and 89 worms (for respective columns), $N = 3$ independent experiments. **c** ArchT-mediated optogenetic inactivation of AQR, PQR, and URX neurons increases exopher production. $n = 59$ worms, $N = 6$ independent experiments. **d** ReaChR-mediated optogenetic activation of AQR, PQR, and URX neurons decreases exopher production. $n = 59$ worms (- light stimulus), 60 worms (+ light stimulus); $N = 6$ independent experiments. **e** The opposing exophergenesis levels observed in animals with genetic ablation of ASK neurons (low exophergenesis) and AQR, PQR, and URX neurons (high exophergenesis) converge to an intermediate level in animals with all four neurons removed. $n = 89$, 90, 95, and 90 worms (for respective columns), $N = 3$ independent experiments. **f** Decreased exophergenesis levels due to exposure to hermaphrodite pheromones are partially mediated by AQR, PQR, and/or URX neurons. $n = 51$, 60, 48, and 54 worms (for respective columns), $N = 3$ independent experiments. **g** Increase in exophergenesis levels due to exposure to male pheromones is not altered in animals with genetic ablation of AQR, PQR, and URX neurons. $n = 103$, 105, 102, and 102 worms (for respective columns), $N = 3$ independent experiments. **h** Genetic ablation of AQR, PQR, and URX neurons does not rescue the inhibition of exophergenesis caused by FUdR-mediated worm sterility. $n = 90$, 89, 89, and 90 worms (for respective columns), $N = 3$ independent experiments. **i** Genetic ablation of AQR, PQR, and URX neurons only partially rescues the inhibition of exophergenesis caused by *rme-2* (yolk receptor) knockdown. $n = 90$ worms, $N = 3$ independent experiments. Data information: +ATR means "with all-trans-retinal". Data are presented as median with interquartile range (**a**, **b**, **e**–**i**) or mean with SEM (**c**–**d**); not significant *p* values ($p > 0.05$) are in orange color, significant *p* values ($p < 0.05$) are in blue color; (**a**–**d**, **f**–**h**) two-tailed Mann–Whitney test, (**e**–**i**) Kruskal–Wallis test with Dunn's multiple comparisons test. Source data are provided as a Source Data file.

## Generation of *str-173* mutant strains

The *str-173* gene mutants (*str-173(wwa1)* and *str-173(wwa2)*) were generated using CRISPR/Cas9 method as previously described[59]. The crRNA sequence used was ATAATTGGTGGATATACAAATGG. The *str-173* gene locus was sequenced, and deletions were mapped to the first exon (Supplementary Fig. 4d). Both mutations cause frame shifts and are most likely molecular null alleles. Strains were backcrossed twice (strain *str-173(wwa1)*) and 6 times (*str-173(wwa2)*) against N2 wild-type worms.

## Generation of optogenetic strains

Optogenetic strains created for this paper contain red-shifted Channel Rhodopsin (ReaChR) or archaerhodopsin from Halorubrum strain TP009 (ArchT). To generate these strains, firstly, mKate2-unc-54 3'UTR was amplified from the template and cloned into pCG150 to create a pAZ03 plasmid. Next, the *gcy-36* promoter was amplified from pMH389, and ReaChR and ArchT were amplified from respective templates. The *gcy-36* promoter was then cloned into pAZ03 plasmid with ReaChR and ArchT separately. As a result, two plasmids were created: gcy-36 promoter::ReaChR::mKate2-unc-54 3'UTR in pCG150 and gcy-36 promoter::ArchT::mKate2-unc-54 3'UTR in pCG150. Plasmids were sequenced to verify the correct sequence of the cloned constructs. All constructs generated for this study were made using the SLiCE method[60].

Transgenic strains with extrachromosomal arrays were generated by microinjection. DNA was injected into exopher reporter strain worms with muscle exopher RFP and mitochondrial GFP marker. For injection, DNA was prepared as follows: construct 90 ng/μL and co-injection marker 10 ng/μL. Positive transformants were selected according to the presence of co-injection markers (myo-2 promoter::mNeonGreen).

A list of all constructs created for this study and primers used for the amplification can be found in Supplementary Data 3.

## Optogenetics assays

**On adulthood day 2.** For optogenetic activation or inhibition, 35-mm NGM plates seeded with HT115 *E. coli* bacteria were covered with 0.2 μM all-trans retinal (ATR). Control plates were not covered with ATR. Ten to twelve age-synchronized worms were picked per plate from optogenetic strains (expressing ReaChR or ArchT in AQR/PQR/URX neurons) at adulthood day 1. After 24-hour incubation at 20 °C and in darkness, muscle exophers extruded by worms were counted **(1)**. Next, experimental plates were placed on the stereomicroscope and illuminated for 1 h with green light (HXP 200 C illuminator as a light source, band-pass filter Zeiss BP 572/25 (HE), the green light intensity measured at 561 nm = 0.07 mW/mm²). The muscle exophers were counted immediately after illumination **(2)**. Subsequently, exophers scorings were performed 15 min **(3)** and 30 min **(4)** after the illumination with a green light was completed. Worms were kept in the darkness in between counts. The control group was not illuminated. To provide similar environmental conditions, control plates were placed next to the experimental plate but were shielded from light. Control and treated groups were randomized before the start of the experiment.

**On adulthood day 1.** The assay was conducted similarly as on adulthood day 2 with a difference in time of exposure to the light. Twelve age-synchronized worms from optogenetic strains were picked at the L4 developmental stage and transferred on a 35-mm NGM plate covered with ATR (no ATR for the control group). Worms were then incubated for 24 h at 20 °C and darkness. Experimental plates were placed on the stereomicroscope and illuminated for 1 h with green light (HXP 200 C illuminator as a light source, band-pass filter Zeiss BP 572/25 (HE), the green light intensity measured at 561 nm = 0.07 mW/mm²). Exophers extruded from BWM were counted before the exposure to the light **(1)**, immediately after the exposure to the light **(2)**, and 15 min **(3)**, 30 min **(4)**, and 24 h **(5)** after the exposure to the light.

## FUdR assay

Age-synchronized young adult animals (day 0) were placed on NGM plates containing 25 μM fluorodeoxyuridine (FUdR) or control NGM plates without FUdR. Exophers number were scored when worms reached adulthood day 2 using a stereomicroscope.

## Transcriptome analysis

RNA extractions, library preparations, and sequencing were conducted at Azenta US, Inc (South Plainfield, NJ, USA) as follows:

**RNA extraction.** Total RNA was extracted using Qiagen RNeasy Plus mini kit following the manufacturer's instructions (Qiagen, Hilden, Germany).

**Library preparation with polyA selection and Illumina sequencing.** Extracted RNA samples were quantified using Qubit 2.0 Fluorometer (Life Technologies, Carlsbad, CA, USA) and RNA integrity was checked using Agilent TapeStation 4200 (Agilent Technologies, Palo Alto, CA, USA).

RNA sequencing libraries were prepared using the NEBNext Ultra II RNA Library Prep Kit for Illumina following the manufacturer's instructions (NEB, Ipswich, MA, USA). Briefly, mRNAs were first enriched with Oligo(dT) beads. Enriched mRNAs were fragmented for 15 min at 94 °C. First strand and second strand cDNAs were subsequently synthesized. cDNA fragments were end-repaired and adenylated at 3'ends, and universal adapters were ligated to cDNA

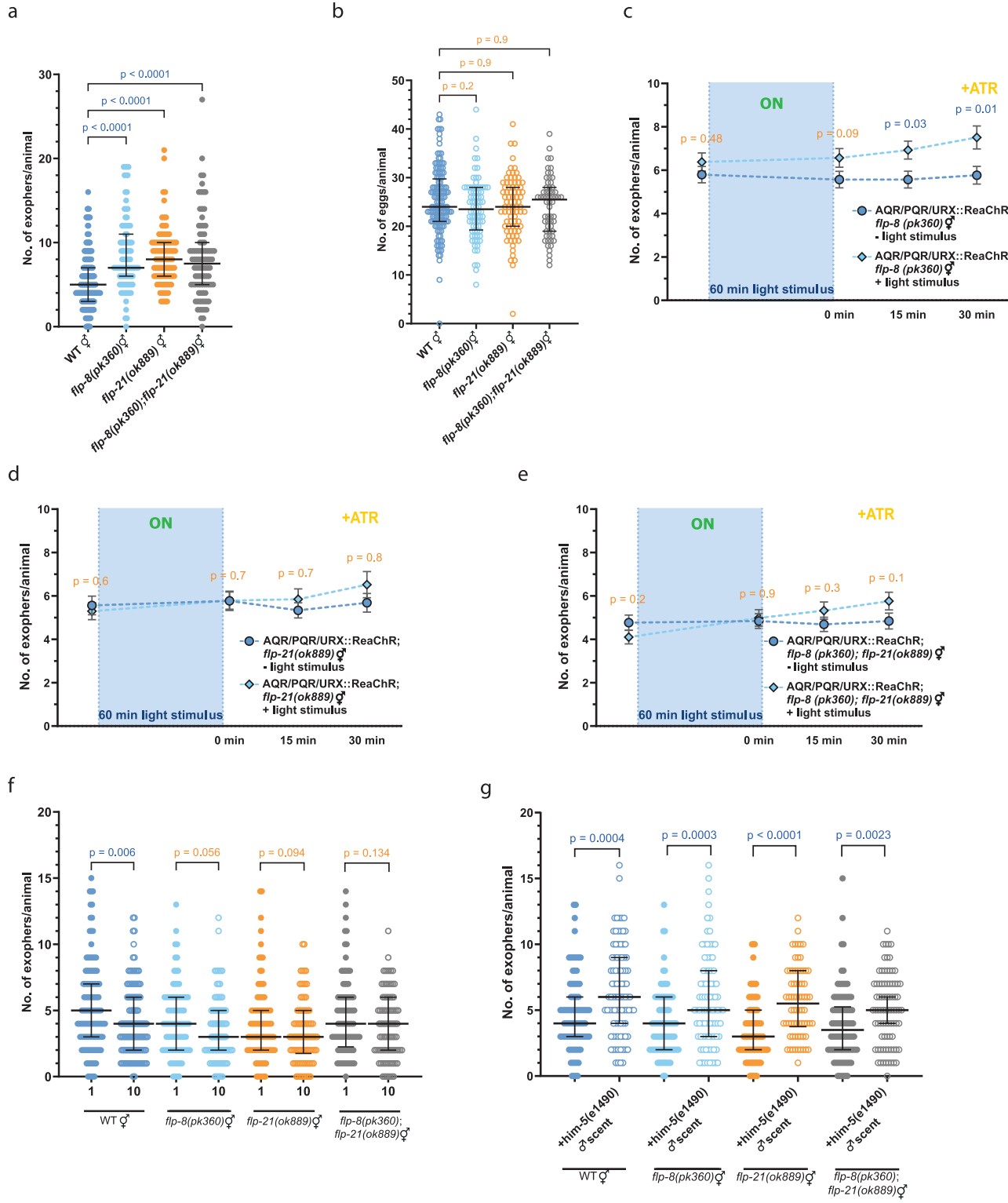

fragments, followed by index addition and library enrichment by limited-cycle PCR. The sequencing libraries were validated on the Agilent TapeStation (Agilent Technologies, Palo Alto, CA, USA), and quantified by using Qubit 2.0 Fluorometer (Invitrogen, Carlsbad, CA) as well as by quantitative PCR (KAPA Biosystems, Wilmington, MA, USA).

The sequencing libraries were multiplexed and loaded on the flowcell on the Illumina NovaSeq 6000 instrument according to the manufacturer's instructions. The samples were sequenced using a 2×150 Pair-End (PE) configuration v1.5. Image analysis and base calling

were conducted by the NovaSeq Control Software v1.7 on the NovaSeq instrument. Raw sequence data (.bcl files) generated from Illumina NovaSeq was converted into fastq files and de-multiplexed using Illumina bcl2fastq program version 2.20. One mismatch was allowed for index sequence identification.

**Sequencing data analysis.** After investigating the quality of the raw data, sequence reads were trimmed to remove possible adapter sequences and nucleotides with poor quality using Trimmomatic v.0.36. The trimmed reads were mapped to the *Caenorhabditis elegans*

**Fig. 7 | Neuropeptides FLP-8 released by URX neurons and FLP-21 released by AQR, PQR, and/or URX neurons inhibit exophergenesis. a** *flp-8(pk360)* and *flp-21(ok889)* single and double mutants have increased exophergenesis levels. *n* = 180 worms, *N* = 6 independent experiments for WT; *n* = 90 worms, *N* = 3 independent experiment for mutants. **b** Mutations in *flp-8(pk360)* and *flp-21(ok889)* genes do not influence embryo retention *in utero*. *n* = 140 worms, *N* = 5 independent experiments for WT; *n* = 80, 80, and 60 mutant worms (for respective columns), *N* = 3 independent experiments. **c**–**e** Optogenetic activation of AQR, PQR, and URX, which do not produce FLP-8 and FLP-21 neuropeptides, does not lead to a decrease in exopher production. (**c**) *n* = 68 (- light stimulus), and 69 worms (+ light stimulus), *N* = 6 independent experiments. (**d**) 71 worms, *N* = 6 independent experiments; (**e**) *n* = 69 (- light stimulus), and 71 worms (+ light stimulus), *N* = 6 independent experiments.

**f** Decreased exopher production in hermaphrodites grown in the presence of other hermaphrodites depends on FLP-8 and FLP-21 neuropeptides. *n* = 119, 120, 86, 90, 90, 90, 88, and 90, *N* = 3 independent experiments. **g** Increased exopher production in hermaphrodites grown on wild-type male-conditioned plates does not depend on FLP-8 and FLP-21 neuropeptides. *n* = 85, 90, 90, 84, 90, 70, 90, and 81 worms (for respective columns), *N* = 3 independent experiments. Data information: +ATR means "with all-trans-retinal". Data are presented as median with interquartile range (**a** - **b**, f - g) or mean with SEM (**c** - **e**); not significant p values (p > 0.05) are in orange color, significant p values (p < 0.05) are in blue color; (c - g) two-tailed Mann-Whitney test, (a - b) Kruskal-Wallis test with Dunn's multiple comparisons test. Source data are provided as a Source Data file.

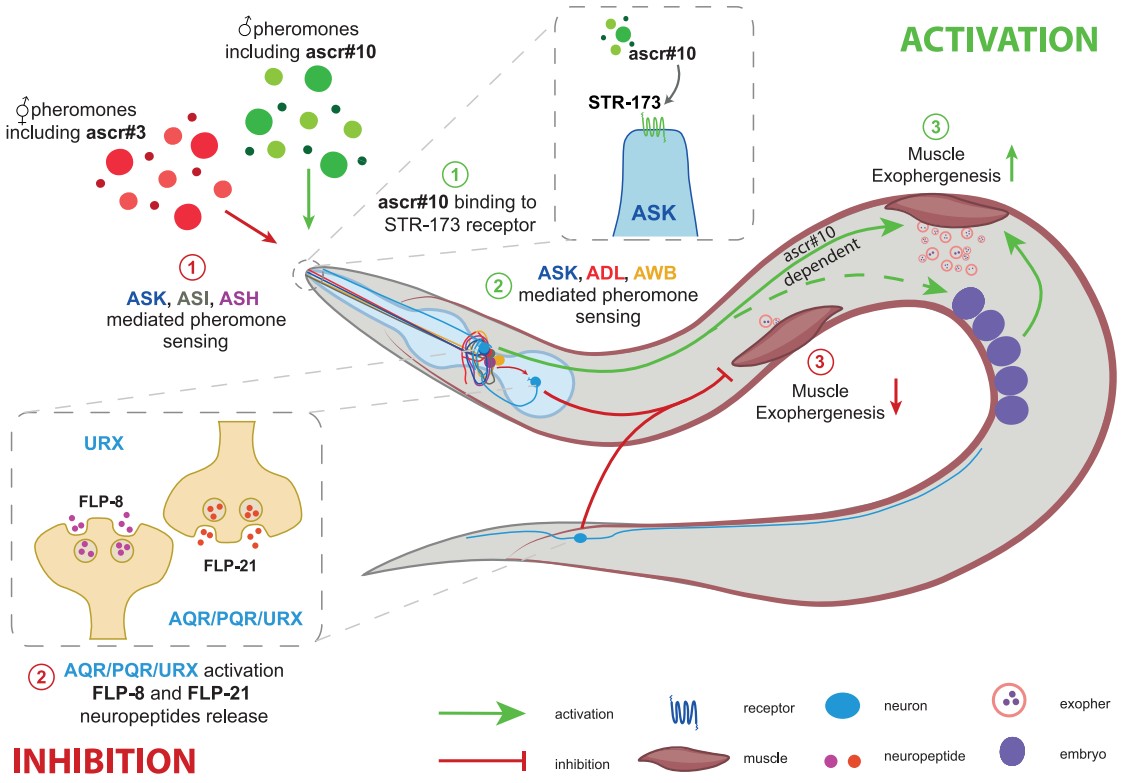

**Fig. 8 | Model.** Males produce ascarosides, including ascr#10, which promote exopher production through ASK, ADL, and AWB signaling, whereby the ASK-expressed STR-173 GPCR enhances exopher production in response to ascr#10. In contrast, hermaphrodites release a different set of ascarosides, dominated by ascr#3, which reduces exopher production, primarily acting via the ASI, ASH, and ASK sensory neurons. The AQR/PQR/URX neurons in the pseudocoelomic cavity release FLP-8 and FLP-21 neuropeptides, negatively regulating exophergenesis. This modulation is critical for decreasing exophergenesis due to hermaphrodite pheromones but not for the increase driven by male pheromones.

reference genome available on ENSEMBL using the STAR aligner v.2.5.2b. The STAR aligner is a splice aligner that detects splice junctions and incorporates them to help align the entire read sequences. BAM files were generated as a result of this step. Unique gene hit counts were calculated by using feature Counts from the Subread package v.1.5.2. Only unique reads that fell within exon regions were counted.

After the extraction of gene hit counts, the gene hit counts table was used for downstream differential expression analysis. Using DESeq2, a comparison of gene expression between the groups of samples was performed. The Wald test was used to generate p-values and Log2 fold changes. Genes with adjusted p-values < 0.05 and absolute log2 fold changes > 1 were called as differentially expressed genes for each comparison.

RNAseq data was deposited in the GEO database (GSE241786) and can be accessed using the following links: https://www.ncbi.nlm.nih.gov/geo/query/acc.cgi?acc=GSE241786.

### Growing hermaphrodite on plates pre-conditioned with other hermaphrodites

50 wild-type hermaphrodites at the L4 stage were placed on a fresh, 35 mm NGM plate seeded with *E. coli* OP50 bacteria (100 μL of bacteria from 10 mL overnight culture grown in 50 mL Erlenmeyer flask). After 12-16 h, hermaphrodites were removed from the plate and a single pretzel-stage embryo was transferred to the hermaphrodite-conditioned plate. For the control groups, single pretzel-stage embryos or ten pretzel-stage embryos were situated on a fresh, 35 mm NGM plate with *E. coli* OP50 bacteria without pre-conditioning with hermaphrodites. When worms reached the second day of adulthood, the number of exophers was scored in each animal.

### Quantifying the influence of larval pheromones on exopher production in hermaphrodites

A single pretzel-stage embryo was placed on a fresh, 35 mm NGM plate seeded with *E. coli* OP50 bacteria (100 μL of bacteria from 10 mL

overnight culture grown in a 50 mL Erlenmeyer flask). The worm was cultured until the late L4 stage, when 50 L1 larvae were added to a single worm. For the control groups, single pretzel-stage embryos or ten pretzel-stage embryos were placed on a fresh, 35 mm NGM plate with *E. coli* OP50 bacteria. Muscle exophers were scored in each animal at adulthood day 2.

## Growing hermaphrodites on plates supplemented with synthetic ascr#10 or ascr#3

Synthetic, concentrated stock solution of ascr#10 was stored in DMSO at -80 °C. The stock was diluted to working solutions with water. A total of 100 μL of diluted solution (containing 1 ng or 1 pg) was applied and rubbed into the 35-mm NGM plate with a sterile glass rod. Plates were incubated at 20 °C overnight. The following day, plates were seeded with 100 μL of *E.coli* OP50 bacteria, and incubated for an additional 24 h at 20 °C. Control plates were prepared similarly using water instead of an ascaroside solution. Pretzel-stage eggs / L4 worms were transferred on plates supplemented with ascarosides and solvent control plates in groups of twelve eggs per plate. On day 2 of adulthood, the number of exophers was quantified in the worms expressing RFP in BWM.

Synthetic, concentrated stock solution of ascr#3 was stored in ethanol at -80 °C. The stocks were diluted to working solutions with water. A total of 50 μL (containing 1 nM or 1pM) was applied on the 35-mm NGM plate seeded with E.coli OP50 bacteria and left in RT for 3 h until dry. Pretzel-stage eggs / L4 worms were transferred on plates supplemented with ascarosides and solvent control plates in groups of twelve eggs per plate. On day 2 of adulthood, the number of exophers was quantified in the worms expressing RFP in BWM.

## Data analysis and visualization tools

Data analysis was performed using Microsoft® Excel® and GraphPad Prism 9 software. Graphical representation of data was depicted using GraphPad Prism 9.

## Statistical analysis

No statistical methods were used to predetermine the sample size. Worms were randomly allocated to the experimental groups for all the data sets and experiments were performed blinded for the data sets presented in the following figures: Fig. 1c–g, i, j; Fig. 3a–d, g, h; Fig. 4d, f; Fig. 5c–i; Fig. 7a, b, f, g; Supplementary Fig. 3c, d; Supplementary Fig. 7d, e. Non-Gaussian distribution of residuals was assumed; therefore, nonparametric statistical tests were applied: two-tailed Mann–Whitney (in comparison between two groups) or Kruskal-Wallis test with Dunn's multiple comparisons test (in comparison between more than two groups). $P$-value < 0.05 is considered significant.

## Reporting summary

Further information on research design is available in the Nature Portfolio Reporting Summary linked to this article.

## Data availability

The authors declare that the main data supporting the findings of this study are available within the article and its supplementary files. Source data are provided with this paper. The RNAseq data generated in this study have been deposited in the GEO database under accession code GSE241786. All raw data have been deposited at Zenodo repository and can be accessed at https://doi.org/10.5281/zenodo.10718455[61]. Source data are provided with this paper.

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

## Acknowledgements

Some strains were provided by the CGC, which is funded by NIH Office of Research Infrastructure Programs (P40 OD010440). We thank Henrik Bringmann and Lukas Kapitein for plasmids; Peter Askjaer, Henrik Bringmann, and Antonio Miranda Vizuete for discussions and comments on the manuscript; Zofia Olszewska, Marta Niklewicz, and Monika Woźniak for assistance with worms maintenance and Natalia A. Szulc for assistance in depositing data at the Zenodo repository. Work in the MT Laboratory was mainly funded by a National Science Centre SONATA grant (2019/35/D/NZ3/04091 to MT) and additionally supported by a National Science Centre SONATA BIS grant (2021/42/E/NZ3/00358 to MT). Work in the WP Laboratory was funded by the Foundation for Polish Science co-financed by the European Union under the European Regional Development Fund (grant POIR.04.04.00-00-5EAB/18-00 to WP), and additionally supported by the Norwegian Financial Mechanism

2014-2021 operated by the Polish National Science Centre, Poland (project contract number 2019/34/H/NZ3/00691 to WP). Work in the FCS Laboratory was supported by the NIH (R35 GM131877 to FCS).

## Author contributions

Conceptualization: M.T., W.P.; Data curation: M.T., W.P.; Formal analysis: A.S., K.O., K.K., M.T., W.P., F.C.S., J.Y., A.T.I., L.A.; Funding acquisition: M.T., W.P., F.C.S.; Investigation: A.S., K.O., K.K., Y.I., A.T.I., L.A., M.T.; Methodology: M.T., W.P., K.O., A.S., K.K., F.C.S., Y.I.; Project administration: M.T., W.P.; Resources: M.T., W.P.; Supervision: M.T., W.P.; Validation: M.T., W.P., F.C.S., A.S., K.O., K.K.; Visualization: A.S., K.O., K.K., M.T.; Writing – original draft: M.T., W.P.; Writing – review & editing: M.T., W.P., F.C.S., A.S., K.O., K.K.

## Competing interests

The authors declare no competing interests.
