## [Peer Review File · Nature Communications]

Pheromone-based communication influences the production of somatic extracellular vesicles in *C. elegans*REVIEWER COMMENTS

Reviewer #1 (Remarks to the Author):

The genesis and function of cell-derived vesicles is a hot area of current biology, and as the authors point out, details of the extracellular signals that might promote their release are generally lacking. This work asks whether large vesicle exopher production from young adult *C. elegans* muscle can be modulated by environmental exposure to hermaphrodites or males. In showing that exopher production levels are responsive to the external “social” environment, the authors identify ascaroside biosynthesis as a factor in some modulation and map out specific neurons and molecular receptors that may contribute to particular cues and responses.

Exopher formation is suppressed by exposure to hermaphrodites; but is stimulated by male exposure, or male-conditioned media. The male effect is correlated with the presence of eggs in the hermaphrodite gonad—males influence egg retention AND enhanced muscle exopher production. Hermaphrodite influence on the individual can be perturbed via exposure to mutants for ascaroside biosynthesis. Starved swimming animals can produce secreted factors that elevate muscle exophers; and this effect is altered in extracts derived by deletion in the *maco-1* biosynthesis mutant. The authors also show that exopherogenesis levels in multiple paradigms (male conditioned medium; single hermaphrodite culture; possibly response to elevated hermaphrodite level) are perturbed by mutations in the STR-173 GPCR protein. Ciliated neurons are required for WT baseline exopher production and, drilling down, ASK AWB ADL are also needed for baseline. In the 9+1 reporter test for hermaphrodites, ablation of ASK, ASH, ASI disrupts responsiveness to exposure. For male conditioned media impact on exophers, ADL ASK and AWB are important. Oxygen-sensing neurons with connection to the internal environment of the pseudocoelom are also able to modulate muscle exopherogenesis levels.

The overarching theme of this paper is that secreted social cues can modulate the extent of muscle EV production (which influences reproductive fitness), which is interesting. One question for publication at Nature Communications is novelty; a fast search indicates little, if any, implication of pheromones in large vesicle (exopher) extrusion; so the work has high impact on that front.

The issue is that data support that social exposures, identified sensory neurons, a GPCR, have roles in multiple socially-directed muscle exopher responses but the paper does not establish details of specific mechanistic links. Instead included are fascinating and reasonably documented fragments that refer to social signals relevant to particular exopher outcomes. Thus, work remains to be done. Ideally the precise connection of particular ascaroside or secreted signal through an identified receptor through a particular neuron with muscle signal delineated would be the goal. Admittedly, this is a tall order; to follow are suggestions for improving the manuscript that might move to meet recommendation standard. Asterisks indicate high importance.

**In general, the legends, and parts of the Methods lacked details required to fully understand the experiments and the significance of outcome. Specific examples of this are cited below but revision with general attention to having figures and legends including minimal information that enables the figures and legends to report on their own would enhance readability of this work.

The paper jumps back and forth between the hermaphrodite and male signals, which the authors convincingly argue are distinct, although there is some overlap. Possibly the messages and conclusions might emerge more clearly if all the hermaphrodite and then all the male related data were presented in the text. The downside of this is that the hermaphrodite effect is somewhat small, so the early showing of the larger male secretion effect helps convince that the biology is interesting. A table summary of all the phenotypes suggestive of “social” impact along with cells and molecules that influence each phenotype might help organize the final message.

In general the authors might better concentrate on defining “social-induced” pathways in response to different social interactions in a way more focused on the male pathway, the hermaphrodite pathway, and possibly the “starvation” culture pathway.

Fig. 1 and related SFig. 1

**Supplementary Figure 1a. and Line 94 related text on dose dependence. Data do not show dose dependence 5 vs. 10 or 10 vs. 100. For this reason the statement of dose dependence should be removed from the paper text.

**Text line 89. The 44% difference cited is a difference of an average of 5 vs. an average of 3. Statement as a percentage, although correct, impresses as somewhat misleading. Better to note levels are lower and eliminate the statement on %.

*In general, the magnitude of the differences for the hermaphrodite exposure effects studied is a concern--the hermaphrodite inhibition effect is on the order of 5 exophers per animal down to 2 or 3 per animal. The authors do not indicate if scoring is done blind to experimental condition, but doing so would enhance confidence on seemingly modest differences. In the other hand, statistics appear appropriate and large numbers of animals are scored.

**The authors should at least comment on hermaphrodite plate conditioning experiments, which are missing here. Does hermaphrodite culture condition the media such that a chemical signal confers an exopher suppression? Even if this did not work, the authors should discuss.

-The details of the plate environment in social experiments should be better provided. In the experiments where animals are reared from L1 up to adulthood, progeny will be generated by the test animals prior to the Ad2 scoring. It is not clear from Methods, but the implication is that the 1 vs. 10 test animals might be differentially swarmed with progeny—many more progeny should populate the 10-hermaphrodite plates at the time of scoring. If so, this experiment should be repeated with animals moved from progeny to eliminate the impact of their progeny on the outcome.

-Is it the male exposure as larvae or into adulthood that matters? The question of the male lifestage for effective conditioning and the lifestage at which the hermaphrodite exposure is interesting but not clearly addressed/presented. For example text in line 123—"showed no further increase"—compared to what? Sfig1e—"above the L4 stage" is not clear if this means in addition to or after (as in longer then).

So, the experimental design of temporal male exposure during development to the L4 but not into adult life is not well explained. It appears the male exposure is during development, but males are removed at the L4 stage after which exposed animals progress to day 2 of adult life—the work should report whether the early exposure is a required element—can older animals respond to the exopher induction cues—for example expose L4 to Ad1? If males are not removed is the outcome different? This is an important point in considering the signaling as being assessed over a few developmental days vs. more acute exopher induction in adult life. A bit more discussion is warranted for clarity. The careful delineation of sensitive stage for production and reception would be a strong addition to the paper.

-Regarding the male conditioning experiments in Figure 1e-j, studies were done with him-5 mutants and comparison appears to be to WT hermaphrodites. Rigor and generality could be expanded by conducting the same experiment of adding WT males (easily generated in bulk by a simple crossing of WT males to WT hermaphrodites) to WT cultures.

-Legend b, e useful to indicate the time/stage animals were added to the original growth plate (from L1) in the legend for clarity.

-1e,f,g. Information should be added as to what control is—the experiment shows WT vs. him-5 mutant, which would be more clear by indicating WT in the image and graph axes. There could easily be confusion as to whether him-5 hermaphrodites are the control.

-h should indicate the genotypes used to condition and also used to test—him-5 or WT.

- Line 109 better stated him-5 mutants rather than him-5 animals

-Line 116 in text—WT hermaphrodites but him-5 males? This should be clear in the text by adding the genotype.

-Line 123 Please add that this statement refers to him-5 mutants, not WT. Supplemental 1e does not match text description—do the authors mean after rather than above ?

-Line 124 exopher generation

-Supplementary 1d indicate what strain the control is.

-Supplementary 1e—explain experimental details better.

-Line 287 shows not show

Figure 2 and associated text

**1) Paradoxical is that in the biosynthetic pathway in Figure 2a ACOX-1 is upstream of MAOC-1, which is upstream of DAF-22, but genetic disruption of maoc-1 suppresses, but disruption of acox-1 and daf-22 enhances muscle exopherogenesis. The mixing of ascaroside biosynthesis mutants with WT cultures is a good addition to the paper. WT + WT should give the modest suppression quantitated in Figure 1C and supplemental Figure 1a. The experiments in Figure 2e indicate that in daf-22 mutants the hermaphrodite suppression effect is lacking. maoc-1 mutants confer a suppression that is normal or a bit enhanced; acox-1 hermaphrodites confer a little boost. The authors should comment in relation to the biosynthetic pathway and ascaroside biology as to what they hypothesize is going on.

The literature notes that most ascarosides with fatty acid sidechains less than 9 carbons are not made in maoc-1 and daf-22, whereas precursor VLCFA and LCFA- conjugated ascarylose precursor accumulates. ACOXs produce enoyl-CoA from acyl-CoA--some ascr pheromones are not synthesized in the acox-1 (ok2257) background, but the synthesis of others is elevated.

The point is that a bit of discussion that the overall constellation of ascarosides is likely to be shifted in particular ways by the mutants tested, but not eliminated, should be made.

**2) The question of whether the influence of male exposure are dependent on ascaroside biosynthesis is not addressed here. The authors should address this biology; males for ascaroside biosynthesis can easily be generated.

**3) The starving swimming populations are likely to make a different range of ascarosides from those on the well fed plate environments of most of the study. The authors should make more of a clear point on how different the ascaroside populations would be expected to be and better note this is an experiment in which exopher production is changed by a different chemical exposure, with distinctive associated ascarosides.

Figure 3 identifies specific subsets of ciliated neurons needed for normal response to hermaphrodite (d,e) and male secreted factors.

**3d,e,f--There is no description in Methods or legends on how the genetic ablations in specific neurons were executed. There is a hint from the strain list that cell specific caspase 1 was used, but the details are lacking. This information should be added to methods and described briefly in figure legends. Useful would be the addition of a few word titles to the figure panels that summarize what was done in the experiment.

**3f study looks at neurons needed for normal upregulation of via co-culture with males. Since the model presented in the paper is that males are associated with egg retention, it is of value to test if the egg number phenotype correlates with the impact of neuronal ablations. The implication of early data is that exposure of male pheromones increases egg retention which enhances muscle exophers. These data provide an opportunity to separate/link egg retention from neuronal perception of male-produced pheromone. The authors should add measures of egg counts in these studies.

** There is a disconnect with the earlier reported studies here, does male extract depend on ascaroside biosynthesis to work? If so, it is of interest to confirm whether the tested ablations are sensitive to that biology. Regarding hermaphrodite differential response for ascaroside mutants, the involvement can be confirmed by testing in ablation backgrounds.

**3g addresses temperature response in the level of muscle exopher production. There are concerns with text discussion of data. First, data for the control WT strain do not show an increase from 15 to 20oC so the summary statement lines 180 181 is not accurate. Second the statement that AWC removal “aggravates” the response also should be rewritten. Data show that there is a strong suppression of the muscle exopher level at each temperature in the absence of AWC. AWC does play a role in the response but a more precise description of outcome is needed.

Supplemental Figure 2— identify the significantly changed transcript; one wonders why this was not chosen as a candidate mediator of exopher level change.

Supplemental Figure 3 describes the biology of the STR-173 GCR.

S3a, b legend, rather than having question marks indicate uncertainty of identification the authors should indicate likely identification based on position or whatever the assignment criteria was.

**S3b. Authors should add an explanation of what the gap-15 expression pattern is. The example image is strange in that red and green appear adjacent and non-overlapping, yet the conclusion is co-expression in the same cell. Is there an explanation? Or a more representative figure? Providing data on how typical this image is of how many observations would be helpful.

**What is the age of the animals scored for expression reporting? The animal exposure can be from L1-L4 and yet the responses are measured in adult—when does the GPCR signaling (or at least expression) take place, and in what cells is the receptor over this critical time period? The question is whether the receptor is in the right place at the right time to be a direct receptor as implied. Authors could figure that out to enhance mechanistic understanding.

**S3c. , text line 201. It is not clear the str-173 alleles are null alleles. The authors can easily expand the allele descriptions here to include a description impact on coding region. Both mutations appear to confer frame shifts, but indication as to whether shift is likely to extend to downstream coding region of the next exon or whether translation is likely to shift back in to frame consequent to the downstream splice event is important for assessing the likely severity of the alleles. This is especially important give that the two alleles do not confer exactly the same phenotypes.

S3d. Add to legend a description of what basal level reflects—one presumes standard culture of a mixed population.

*S3e. It appears that solitary animals might exhibit modestly lower levels of muscle exophers when str-173 is disrupted, in addition to the failure to turn down levels in response to “social” exposure. Authors should address this with statistical tests and add to text. They should note clearly to remind readers that standard mass growth is different from solitary growth which is different from a 10 animal culture. Does this receptor act at one or all? This is an interesting question.

S3f. Given the changes in response to male conditioned plates, authors might consider examining expression of STR-173 in hermaphrodites co-cultured with males or their conditioned media.

Figure 4 considers whether internal signaling based in the pseudocoelom -exposed neurons AQR, PQR, URXL/R might influence social responses in exopher production. Genetic ablation of these neurons results in an increase in exophers, without a large change in egg load but if eggs are eliminated, any increase in exopher numbers is suppressed. Fertilized eggs are needed as part of the modulation pathway, and thus eggs may be positioned downstream of the neuronal function.

**Figure 4b, S4ab—better to note that eggs and progeny production are modestly reduced; changes appear unlikely to themselves modulate exopher numbers associated with genetic ablations of AQR, PQR and URX. Thus, “reduced” seems an overstatement—there is not much difference that may matter here, which is fine!!!

**Figure 4e would be stronger if the wild type no-ablation control in this experiment was included. The “double” ablation level is likely at WT levels.

Legend 4e and text—outcome of “double” ablation is a phenotype that is in between either single disruption. Each may contribute independently to a summed outcome. Discuss more clearly--The terms “counterbalanced” and “equalized” are vague and should be clarified.

**4c. The 60 minute neuronal inactivation/activation studies are quite interesting, but the implications on the timing are not discussed. The 60 minute disruption with capacity to increase exophers shortly thereafter implies a temporally tight functional connection between these neurons and the muscle exopher response. How do the authors think this works? Does the stage at which the activation/inactivation is delivered matter? (additional comments on clarification of exposure and outcome timing are given above).

Figure 4f is on hermaphrodite impact, as measured by solitary vs. group rearing. An outcome that seems obvious (but no statistics are indicated in the panel) is that baseline muscle exophers in the solitary and group are both elevated consequent to ablation. This can be interpreted to indicate that AQR/PQR/UBX action normally inhibits baseline exophers which should be better pointed out. High baseline is also evident in the ablation + male scent assay. Hermaphrodite downregulation in the G10 condition is lost—AQR, PQR, UBX might mediate the normal suppression by hermaphrodites as well, which the authors conclude.

Figures 4J,I model—where does STR-173 fit in these models? Minimally there has to be some note/discussion of this issue. Mutant *str-173* alleles do not appear to impact baseline, might mediate the hermaphrodites 9/1 suppression, might influence the male conditioned medium response. Baseline impact should be discussed as well as what the data might imply for the specific outcomes assayed.

*Figure 4I—To test this model, the authors should show that “pathway specific” ADL, AWB influence egg retention.

The Discussion is quite speculative and could benefit from addition of a summary clarification of what can and cannot be concluded about multiple responses and factors in the external environment and their impact on muscle exophers. Hermaphrodite and male influences, how much is actually shown to require proper ascaroside synthesis, *str-173*, eggs? Can male conditioned medium work but not hermaphrodite? The authors might consider a summary table as there are multiple implications of data and multiple pathways, but the precise model for a given influence does not easily emerge.

**Line 255. Although the authors document changes in muscle exopher production in response to environmental conditions, the leap to assuming exophers play a likely role in inter-animal communication impresses as a step too far. Ascarosides or other chemicals are implicated in communication by data but exopher exchange, per se, is not; this aspect of the discussion should be toned down.

**Line 262. It is an overstatement to say that the authors have shown that neuroendocrine signals are secreted; the authors showed AQR/PQR/and URX negatively regulate baseline but the molecular mechanism is not addressed. Vesicle release by these neurons is not documented. Authors should revise discussion on this point.

Reviewer #2 (Remarks to the Author):

Banasiak et al. investigated how secreted metabolites that are possibly related to ascaroside pheromone affect the genesis of large extracellular vehicles (exophergenesis), a biologically important process that remains very poorly understood. The authors further report the involvement of GPCR STR-173 and sensory neurons AQR, PQR, and URX.

Major points:

- In the first two sections, the authors showed that worms grown in the presence of male secretome had an effect on exophergenesis that is opposite to that of hermaphrodite-derived secretome, and further, that exophergenesis is affected by peroxisomal beta-oxidation genes and secreted molecules. These observations suggest that ascaroside pheromones are involved, since ascaroside biosynthesis requires peroxisomal β -oxidation. However, ascarosides are a highly diverse class of molecules - more than 100 different structures have been reported, many of which are produced in a sex-specific manner. In addition, the production (and possibly their secretion) many other lipids likely depends on peroxisomal β -oxidation. Therefore, whether ascarosides are involved or not must be tested. Based on the existing knowledge of *C. elegans* sex-specific ascaroside biosynthesis, the authors could have developed testable hypotheses on the molecular identities of potentially involved ascarosides to perform validation assays. Without such validation experiments, I don't think any firm conclusions can be drawn from the data presented here.
- Importantly, the observation that *maoc-1* mutants display a reduction in exopher production, whereas the *daf-22* and *acox-1* mutants display an increase, may actually speak against the involvement of ascarosides, since most production of ascarosides is abolished in both *maoc-1* and *daf-22* worms. Thus, while the data clearly show that peroxisomal β -oxidation is involved, it is entirely unclear whether ascarosides are involved or not.

Minor points:

- Though only included in a supplementary figure, one of the most interesting claims in this manuscript is that STR-173 may be involved in sensing of "pheromone" (though it's unclear whether this is ascarosides). However, to support that STR-173 functions as a GPCR sensor in the exopher context one would expect more molecular evidence – at least testing isolated metabolites or synthetic candidate compounds.
- STR-173 does not appear to be expressed in the relevant neurons, therefore a logical connection is missing.
- Figure 2a: using 'very long chain ascarosides' as the starting point might match better in this case.

Taken together, the authors aimed to demonstrate a pheromone (ascaroside)-sensor (GPCR)-neuron axis regulating exophergenesis; however the findings presented in this fairly concise manuscript are insufficient to support this model.

Reviewer #3 (Remarks to the Author):

Summary: This is an exciting paper on the role of exophers in ascaroside signaling. The authors discovered that the hermaphrodites grown with hermaphrodites produce fewer exophers, but have the same number of eggs in utero. Conversely, hermaphrodites grown with males or on a male-conditioned plate produce more exophers and also produced more eggs in utero. These experiments allowed the authors to conclude that exopher formation is influenced by the pheromones worms sense. Since pheromones influence exopher production, the authors wanted to see if genes involved in ascaroside synthesis, *maoc-1*, *daf-22*, and *acox-1* affect exopher production. The authors found that *maoc-1* worms had decreased exopher and egg production while *daf-22*, and *acox-1* worms had an increase in both exopher and egg production. This caused the authors to conclude that MAOC-1 plays a role in exopher-mediated pheromone synthesis. Since neurons involved in ascaroside detection are known, the authors wanted to see if these neurons also mediate pheromone-induced exopher production. They additionally discovered that the ASK, AWC, and ADL are the main neurons required for normal exopher production. The authors also found that the ASK, ASH, and ASI neurons were required for a hermaphrodite pheromone-induced decrease in exopher production whereas the ASK, ADL, and AWB were required for male pheromone-induced increase in exopher production and identified *str-173* as a candidate gene because it was differentially expressed in worms grown alone and worms grown with conspecifics.

Overall, this paper supports that exopher formation is influenced by the ascarosides a worm is sensing, and this effect may be related to the role that ascaroside signaling plays in reproduction.

I RECOMMEND THE PAPER FOR SOME REVISIONS.

My concerns are as follows:

1. In figure 1, the authors indicate that there is a relationship between pheromone exposure, exophergenesis, and embryogenesis. The exposure to male pheromones increases both exopher formation and egg count (Fig 1e-j). Since various sensory neurons are implicated in the male pheromone dependent increase (ASK, ADL, and AWB) of exopher formation, do these same neurons affect embryogenesis too? If these neurons were found to also modulate the increased egg production upon exposure to male pheromones, I think it would help support that these phenotypes are related.

2. In Figure 2a, there is a schematic showing how each of the genes tested is involved in ascaroside synthesis. This figure implies that each of these genes functions in the same pathway. If this were true, it would seem that all of the mutant secretions would create the same phenotype. Could you add text explaining why you think why the secretions of each mutant creates different exopher numbers (fig 2e) when all three mutants are deficient in ascaroside production? In other words, what is a specific difference between the mutant secretions could these results (in fig 2e) be attributed to?

3. In Figure 3d-3f, the authors examine how ablating various cells affects the exopher response. They also show how masculinizing the hermaphrodite nervous system with the addition of the CEM neurons changes exopher response. I would prefer if the CEM data were separated because it feels like it's answering a different question with this data (which part of the male nervous system is required for exopher response?) Whereas the ablation data is trying to answer which part of the hermaphrodite nervous system is involved in exopher pheromone response. With this, if you're going to include the CEM data, it might make sense to just include the WT male response because I'm not really sure what that would look like as a baseline. If you see that WT males and hemaphrodites with CEMs have the same response, then you can conclude that the CEM neurons are required for exopher response in males.

4. In the intro, could there be more text about how exophers are related to reproduction? It would be helpful to have more background on this when reading the paper (unless more is not known).

Reviewer #4 (Remarks to the Author):

This work demonstrates that signals released to the environment by conspecifics regulate the production of muscle exopheres in *C. elegans* hermaphrodites. While signals from hermaphrodites decrease exosphere production, signals from males increase production. Furthermore, the authors identify 2 distinct groups of sensory neurons mediating either signal. Overall, I don't feel that the results presented here contribute significantly to advance the field of extracellular vesicle communication. The mechanisms by which the signals affect exosphere production are unknown. In addition, it is not clear at all what would be the adaptability of this decrease or increase according to the sex of the conspecifics.

More specific comments that the authors may consider addressing in future submissions are below:

- The data in this manuscript should be pretty straight forward to follow and understand. However, the paper is written in such an unclear manner that is extremely hard to interpret the experiments performed and follow the logic and conclusions of the authors. An example in point is all the data in figure 1. The manipulations carried out are very unclear to me. On growing worms in presence of males and having an impact on exospherogenesis: why do the authors say male scent? It may be contact by males during development, or mating in adulthood, etc. at this stage of the data presented they don't know what aspect of male presence is the cause. Is it having male essence after L4? Or is it growing with males? When is the effect produced? I just don't understand what they did in line 117, are they changing genotype (now N2) and protocol (removing the males which they didn't with him-5 mutants)? Then you can't distinguish between the two.

I think the sentence "Growing hermaphrodites on male-conditioned plates increased exosphere production to the same degree as when hermaphrodites were grown with males until the L4 larvae stage (Fig. 1i)" doesn't make sense because they condition the plates by growing herms with males, so the sentence should say "growing N2 herms with males increased exosphere production to the same degree as when growing him-5 hermaphrodites", right? Otherwise, I don't understand what they did or how many factors they changed. And how is this different from "Furthermore, adult hermaphrodites exposed to males' secretions as larvae showed no further increase in exosphere production" ?

- It is also unclear why the authors purify/extract secretions from starving animals in dauer to mimic the signals from well-fed adult conspecifics that regulate exosphere production in their essays.

- The results with the mutants in ascaroside production are also complex to interpret. Is the effect on exosphere production due to an increase in long-chain ascarosides or a decrease in short-chain ascarosides (both result from manipulating the synthesis pathway)?

- In all the experiments of sensory neuron manipulation the authors should just present the data set where they compare mutant animals raised singly and mutant animals raised with 10 others. This is the correct experiment to assess the contribution of sensory neurons to sensing the exospherogenesis-regulating signal without having confounding autonomous effect of sensory neurons on exosphere production. The presence of the other data set where they assess exospherogenesis in a mutant population does not add any useful information and makes data interpretation confusing.

- In figure 3d, why is ASH ablation significantly different from control but AWB ablation is not? The graph would suggest otherwise. And why do the authors single out the effect of CEMs in the text to say that adding CEMs leads to a reduction in exosphere production in single animals? It also happens by removing ADL, ASK, AWC but they do not mention it. In addition they need statistical analysis comparing to control if they want to make that claim on CEMs. Also why do they say for pheromone response ASK has less of a role than ASH and ASI? How do they reach that conclusion?

- Finally, regarding the role of AQR, URX and PQR in exospherogenesis, their data suggests that the effect goes through egg production, as egg production is required for these neurons to increase the production of exospheres (Fig 4h and i). Please state this accordingly. And, could they assess the same for the effect

of male pheromones on exosphere production? Male pheromones increase both egg production and exosphere production. Could the authors dissect whether these are linked or not?

We thank the reviewers for constructive comments that improved our manuscript, and we are happy to present this revised version. In response to reviewers' comments, significant experimental work was undertaken, the results of which are shown in new panels in **Fig. 1f, g, h, i; Fig. 2f, g; Fig. 3e, f, g, h; Fig. 4d; Fig. 5a, c, d, e, f, g, h, i, j; Fig. 7a, b, c, d, e; Supplementary Fig. 1c; Supplementary Fig. 3a, b, c, d; Supplementary Fig. 5e, f, g, h; Supplementary Fig. 6a, b, c; Supplementary Fig. 7a, b, c.** Thanks to the obtained results, we were able to draw some exciting new conclusions that significantly strengthened the molecular aspect of our manuscript. Our new results have shown that the male pheromone *ascr#10* has the ability to upregulate exophergenesis through the STR-173 receptor in the ASK neurons. Moreover, we revealed that hermaphrodite pheromone *ascr#18* decreases exophergenesis via ASK neurons. Furthermore, we have discovered that inhibition of exophergenesis is effectively regulated through the FLP-8 and FLP-21 neuropeptides released by URX and AQR/PQR/URX neurons, respectively. In addition, we have made all requested experiments, suggested changes to the text, and analyses.

Please find a detailed description of the edited paragraphs below (the reviewers' comments are in *italics* and our responses are in **blue font**):

Reviewer #1 (Remarks to the Author):

The genesis and function of cell-derived vesicles is a hot area of current biology, and as the authors point out, details of the extracellular signals that might promote their release are generally lacking. This work asks whether large vesicle exopher production from young adult C. elegans muscle can be modulated by environmental exposure to hermaphrodites or males. In showing that exopher production levels are responsive to the external “social” environment, the authors identify ascaroside biosynthesis as a factor in some modulation and map out specific neurons and molecular receptors that may contribute to particular cues and responses.

*Exopher formation is suppressed by exposure to hermaphrodites; but is stimulated by male exposure, or male-conditioned media. The male effect is correlated with the presence of eggs in the hermaphrodite gonad—males influence egg retention AND enhanced muscle exopher production. Hermaphrodite influence on the individual can be perturbed via exposure to mutants for ascaroside biosynthesis. Starved swimming animals can produce secreted factors that elevate muscle exophers; and this effect is altered in extracts derived by deletion in the *maco-1* biosynthesis mutant. The authors also show that exophergenesis levels in multiple paradigms (male conditioned medium; single hermaphrodite culture; possibly response to elevated hermaphrodite level) are perturbed by mutations in the STR-173 GPCR protein. Ciliated neurons are required for WT baseline exopher production and, drilling down, ASK AWC ADL are also needed for baseline. In the 9+1 reporter test for hermaphrodites, ablation of ASK, ASH, ASI disrupts responsiveness to exposure. For male conditioned media impact on exophers, ADL ASK and AWB are important. Oxygen-sensing neurons with connection to the internal environment of the pseudocoelom are also able to modulate muscle exophergenesis levels.*

The overarching theme of this paper is that secreted social cues can modulate the extent of muscle EV production (which influences reproductive fitness), which is interesting. One question for publication at Nature Communications is novelty; a fast search indicates little, if any, implication of pheromones in large vesicle (exopher) extrusion; so the work has high impact on that front.

The issue is that data support that social exposures, identified sensory neurons, a GCPR, have roles in multiple socially-directed muscle exopher responses but the paper does not establish details of specific mechanistic links. Instead included are fascinating and reasonably documented fragments that refer to social signals relevant to particular exopher outcomes. Thus, work remains to be done. Ideally the precise connection of particular ascaroside or secreted signal through an identified receptor through a particular neuron with muscle signal delineated would be the goal. Admittedly, this is a tall order; to follow are suggestions for improving the manuscript that might move to meet recommendation standard. Asterisks indicate high importance.

General remarks

Major points:

1. ***In general, the legends, and parts of the Methods lacked details required to fully understand the experiments and the significance of outcome. Specific examples of this are cited below but revision with general attention to having figures and legends including minimal information that enables the figures and legends to report on their own would enhance readability of this work.*

Response: We have carefully considered and incorporated your suggestions in the new manuscript version. Specifically, we have revised the results and discussion section to provide a more detailed and nuanced interpretation of the findings, their broader implications, and the potential avenues for future research. Additionally, we have revised the figures, legends, and methods sections to ensure they provide comprehensive and stand-alone descriptions of the experiments and outcomes. We believe these revisions enhance the manuscript's clarity, comprehensiveness, and readability.

Minor points:

2. *The paper jumps back and forth between the hermaphrodite and male signals, which the authors convincingly argue are distinct, although there is some overlap. Possibly the messages and conclusions might emerge more clearly if all the hermaphrodite and then all the male related data were presented in the text. The downside of this is that the hermaphrodite effect is somewhat small, so the early showing of the larger male secretion effect helps convince that the biology is interesting. A table summary of all the phenotypes suggestive of “social” impact along with cells and molecules that influence each phenotype might help organize the final message. In general the authors might better concentrate on defining “social-induced” pathways in response to different social interactions in a way more focused on the male pathway, the hermaphrodite pathway, and possibly the “starvation” culture pathway.*

Response: As the reviewer suggested, we have rewritten the manuscript in such a manner that it first focuses on male-related regulation of exophers formation, followed by the data on hermaphrodites regulation of exopher formation. Next, we present the data on molecular mechanisms responsible for exopher formation. Finally, we prepared a table (Supplementary Table 2) summarizing all phenotypes presented in the manuscript and their regulators.

Remarks to Fig. 1, related SFig.1, and associated text

Major points:

3. ****Supplementary Figure 1a. and Line 94 related text on dose dependence. Data do not show dose dependence 5 vs. 10 or 10 vs. 100. For this reason the statement of dose dependence should be removed from the paper text.**

Response: We have removed the statement regarding dose dependency and have replaced it with sentences: "Furthermore, we observed that cultivating hermaphrodites in a population as small as five worms per plate was sufficient to decrease exopher production (Fig. 2d). Conversely, escalating the population size to one hundred animals per plate did not cause any further significant reduction in exopher production relative to 10 hermaphrodites per plate (Fig. 2e)."

4. ****Text line 89. The 44% difference cited is a difference of an average of 5 vs. an average of 3. Statement as a percentage, although correct, impresses as somewhat misleading. Better to note levels are lower and eliminate the statement on %.**

Response: We have removed the statement on the percentage difference. The current version of the sentence: "In contrast to the effects of the presence of males, our results reveal that hermaphrodites grown at 10 hermaphrodites per plate consistently released fewer exophers compared to those grown as solitary animals (Fig. 2b)"

5. ***In general, the magnitude of the differences for the hermaphrodite exposure effects studied is a concern-the hermaphrodite inhibition effect is on the order of 5 exophers per animal down to 2 or 3 per animal. The authors do not indicate if scoring is done blind to experimental condition, but doing so would enhance confidence on seemingly modest differences. In the other hand, statistics appear appropriate and large numbers of animals are scored.**

Response: We appreciate your concern regarding the magnitude of the differences in the hermaphrodite exposure effects studied. We understand that the observed differences in exopher production per animal might seem modest; however, the statistical analysis, as detailed in the "Statistical analysis" section of the Materials and Methods, confirms that these differences are indeed significant. Additionally, this section clarifies which experiments were conducted blindly to ensure objectivity in scoring and data analysis. We believe that the combination of statistically significant results, large numbers of animals scored, and appropriate blinding of experiments, where indicated, collectively support the robustness and validity of our findings.

6. ****The authors should at least comment on hermaphrodite plate conditioning experiments, which are missing here. Does hermaphrodite culture condition the media such that a chemical signal confers an exopher suppression? Even if this did not work, the authors should discuss.**

Response: We appreciate the reviewer's comment on hermaphrodite plate conditioning experiments. Our revised results indeed include experiments addressing this issue. Briefly, we found that growing a single hermaphrodite on a plate conditioned by other hermaphrodites reduced exopher numbers to a level comparable to that in a ten-hermaphrodite population, indicating that hermaphrodite culture does condition the media in a way that influences exopher production. Data are presented in Fig. 2f and this is part of a larger set of experiments elucidating the interplay between social cues and exophogenesis, which is discussed in detail in the revised manuscript.

Minor points

7. *The details of the plate environment in social experiments should be better provided. In the experiments where animals are reared from L1 up to adulthood, progeny will be generated by the test animals prior to the Ad2 scoring. It is not clear from Methods, but the implication is that the 1 vs. 10 test animals might be differentially swarmed with progeny—many more progeny should populate the 10-hermaphrodite plates at the time of scoring. If so, this experiment should be repeated with animals moved from progeny to eliminate the impact of their progeny on the outcome.*

Response: Thank you for raising an important consideration regarding the plate environment in our social experiments.

You correctly pointed out that the progeny generated by the test animals prior to the AD2 scoring could potentially affect the experiment outcomes, given that many more progenies should populate the 10-hermaphrodite plates at the time of scoring.

Thank you for recommending the transfer of animals away from their progeny to negate potential influences. However, we were concerned that doing so could affect our experimental setup by moving animals to new unconditioned plates. In response to your concern, we have conducted a different experiment that still addresses your issue. Specifically, we tested the hypothesis you formulated about the influence of larvae presence on exophogenesis in hermaphrodites. To this end, we grew a single reporter worm at the L4 stage with 50 wild-type L1 larvae. By the time the reporter worm reached the AD2 stage, the remaining population was at the L4 stage. We found that aggregating a population of 50 larvae did indeed affect exophogenesis, lowering its level to roughly that observed when aggregating 10 adult worms (Fig. 2g).

This experiment allowed us to conclude that the presence of a larger number of larvae (progeny) impacts the number of exophers. It also strengthens one of our main conclusions, that increasing hermaphrodite population density, both with mature and immature animals, inhibits exophogenesis.

8. *Is it the male exposure as larvae or into adulthood that matters? The question of the male life stage for effective conditioning and the life stage at which the hermaphrodite exposure is interesting but not clearly addressed/presented. For example text in line 123—"showed no further increase"—compared to what? Sfig1e—"above the L4 stage" is not clear if this means in addition to or after (as in longer then). If males are not removed is the outcome different? This is an important point in considering the signaling as being assessed over a few developmental days vs. more acute exopher induction in adult life. A bit more discussion is warranted for clarity. The careful delineation of sensitive stage for production and reception would be a strong addition to the paper.*

Response: We appreciate the reviewer's keen observations and the request for further clarification on the impact of the male life stage on effective conditioning and the life stage at which the hermaphrodite is exposed. Our revised results section provides some insights into this question. We found that 24 hours of exposure to male pheromones, regardless of the life stage at which the exposure occurred, was sufficient to increase exophogenesis in hermaphrodites. This was observed even when hermaphrodites were exposed only during the larval development, indicating that exposure to male pheromones during the larval stage can influence exopher levels in adult hermaphrodites. Specifically, the most potent effect was observed in animals exposed during the L4 larval stage to young adults' day 1 stage, mirroring the effects of continuous exposure to males' secretions (Fig. 1h-i).

Regarding the question of the male life stage for effective conditioning, it is difficult to assess as males are distinguishable from hermaphrodites at the L4 stage, and it would be hard to test if ascariosides produced in earlier male stages are inducing an increase in exophogenesis in hermaphrodites. Especially since our experimental setup requires 48 hours of male conditioning, which is long enough for males to go through different developmental stages.

We have included a more detailed discussion of these points in the revised manuscript to provide a more precise delineation of the sensitive stages for production and reception.

9. Regarding the male conditioning experiments in Figure 1e-j, studies were done with *him-5* mutants, and comparison appears to be to WT hermaphrodites. Rigor and generality could be expanded by conducting the same experiment of adding WT males (easily generated in bulk by a simple crossing of WT males to WT hermaphrodites) to WT cultures.

Response: Thank you for your insightful suggestion regarding the rigor and generality of our experiments. To address your comment, we have repeated the experiment with hermaphrodites grown on plates conditioned with wild-type (WT) males. The data from this experiment are presented in Fig. 1f-g of the revised manuscript.

In the revised experiment, we cultured wild-type hermaphrodites on plates conditioned with either *him-5* or wild-type males for 48 hours before removing them. The results from this experiment indicated that the conditioning elevated exopher production equivalently to when hermaphrodites were co-cultured with males until the L4 larval stage. This finding supports our original conclusion that male pheromones promote exopher generation in hermaphrodite muscles, regardless of whether the males are *him-5* mutants or wild-type.

We believe this additional experiment strengthens our study by demonstrating that the effect is consistent across different genetic backgrounds and is not an artifact of the *him-5* mutation.

10. Legend Fig. 1b, e useful to indicate the time/stage animals were added to the original growth plate (from L1) in the legend for clarity.

Response: Absolutely, clarity in the figure legends is crucial. The revised legend for Figure 1b and 1e would read as follows:

Fig. 1b: "Schematic representation of the experimental setup for investigating the influence of increased male presence in the population. *Him-5* mutant hermaphrodites and *him-5* mutant males were co-cultured from the L1 stage until the L4 stage and then transferred to a male-free plate. Hermaphrodites were grown on male-free plates until adulthood day 2 (AD2) when the number of exophers was assessed."

Fig. 1e: "Schematic representation of the experimental setup for investigating the influence of male's secretome on exophogenesis level. 50 *him-5* or wild-type males were grown on a plate for 48 hours before removal. Next, 10 hermaphrodites were transferred to plates previously occupied by males and grown until adulthood day 2 (AD2) when the number of exophers was assessed."

11. Fig. 1e,f,g. Information should be added as to what control is—the experiment shows WT vs. *him-5* mutant, which would be more clear by indicating WT in the image and graph axes. There could easily be confusion as to whether *him-5* hermaphrodites are the control.

Fig. 1h should indicate the genotypes used to condition and also used to test—*him-5* or

WT. Supplementary Fig. 1d indicate what strain the control is.

Response: For all the figures, we have added information about the genetic background of hermaphrodites and males used in experiments.

12. Line 109 better stated *him-5* mutants rather than *him-5* animals

Response: We have made appropriate changes in the text.

13. Line 116 in text—WT hermaphrodites but *him-5* males? This should be clear in the text by adding the genotype.

Response: We have made appropriate changes in the text.

14. Line 123 Please add that this statement refers to *him-5* mutants, not WT. Supplementary 1e does not match text description—do the authors mean after rather than above ?

Response: We have made appropriate changes in the text. The revised part of the manuscript read as follows: “Longer exposure to male-conditioned plates or co-culture with males beyond larval development did not further increase exopher production in *him-5* mutant hermaphrodites (Supplementary Fig. 1a-c).”

15. Line 124 exopher generation.

Response: We have made appropriate changes in the text.

16. Supplementary Fig. 1e—explain experimental details better.

Response: For Supplementary Fig. 1b (previously Supplementary Fig. 1e), we have provided a schematic representation of the experimental presented in Supplementary Fig. 1a.

17. Line 287 shows not show.

Response: We have made appropriate changes in the text.

Remarks to Fig. 2 and associated text

Major points:

18. ****Paradoxical** is that in the biosynthetic pathway in Figure 2a ACOX-1 is upstream of MAOC-1, which is upstream of DAF-22, but genetic disruption of *maoc-1* suppresses, but disruption of *acox-1* and *daf-22* enhances muscle exophergenesis. The mixing of ascaroside biosynthesis mutants with WT cultures is a good addition to the paper. WT + WT should give the modest suppression quantitated in Figure 1C and supplemental Figure 1a. The experiments in Figure 2e indicate that in *daf-22* mutants the hermaphrodite suppression effect is lacking. *maoc-1* mutants confer a suppression that is normal or a bit enhanced; *acox-1* hermaphrodites confer a little boost. The authors should comment in relation to the biosynthetic pathway and ascaroside biology as to what they hypothesize is going on.

The literature notes that most ascarosides with fatty acid sidechains less than 9 carbons are not made in *maoc-1* and *daf-22*, whereas precursor VLCFA and LCFA- conjugated ascarylose precursor accumulates. ACOXs produce enoyl-CoA from acyl-CoA--some ascr pheromones are not synthesized in the *acox-1* (*ok2257*) background, but the synthesis of others is elevated.

The point is that a bit of discussion that the overall constellation of ascarosides is likely to be shifted in particular ways by the mutants tested, but not eliminated, should be made.

Response: Thank you for your insightful feedback, highlighting the areas requiring deeper exploration and analysis concerning the biosynthetic pathway and ascaroside biology in the context of our research.

To offer a more detailed representation, we have revised Fig. 3a (previously Fig. 2a) to portray the shifts in the ascaroside profile brought on by the different mutants, essentially to illustrate that in *acox-1* and *maoc-1* mutants, the ascaroside profiles are modified, but production of ascaroside pheromones is not abolished, explaining the different effects on muscle exophergenesis.

We upheld the fundamental experiments encompassing the *acox-1(ok2257)*, *maoc-1(ok2645)*, and *daf-22(ok693)* mutants, facilitating the delineation of the distinct ascaroside profiles of each,

including a substantial increase of the male pheromone ascr#10 levels in *acox-1* mutants, and a decrease in *maoc-1* mutants (as depicted in Fig. 3a). These experiments were pivotal in demonstrating the differential impact of these mutants on wild-type exopher levels in a co-culture setup, an influence that stood independent of embryo-maternal signaling (highlighted in Fig. 3b-d). Moreover, the differences in the ascaroside profiles between hermaphrodites, males, and the ascaroside biosynthesis mutants led us to uncover the differential roles of ascr#10 and ascr#18 in modulating exopherogenesis, using synthetic samples of these ascarosides. These new data confirm that sex-specific ascarosides play an important role for exopher formation and underscored the profound relationship between the chemical structure and biological function of ascarosides (Fig. 3e-h). This investigative trajectory necessitated an enriched discussion in the manuscript to foster a nuanced understanding of the complex interplay involved in ascaroside biology and its role in exopherogenesis.

While constructing a comprehensive picture, we took the strategic decision to remove the analysis of exopher production in ascaroside mutants from our manuscript (previously Fig. 2b-c). This decision stems from the realization that these mutants did not bring any novel insights compared to what had already been elucidated through our co-culturing experiments. Including data from these mutants would introduce additional complexity, potentially obscuring the central narrative, given the presence of other factors influencing exopherogenesis in these mutants. Furthermore, we observed that animals with defective peroxisomal 13-oxidation, one of the main lipid metabolic pathways, might inherently possess other physiological characteristics influencing exopherogenesis, thereby complicating the storyline without adding substantial value. Thus, to maintain a focused narrative and avoid diluting the core findings, we opted to streamline our presentation by excluding exopher production in ascaroside biosynthesis mutants from the discussion.

19. ***The question of whether the influence of male exposure are dependent on ascaroside biosynthesis is not addressed here. The authors should address this biology; males for ascaroside biosynthesis can easily be generated.*

Response: Thank you for bringing to our attention the pivotal role the relationship between male exposure and ascaroside biosynthesis could potentially play in our study.

In light of your feedback, we revisited our experiments to assess this relationship more thoroughly. Our revised results indeed delve into the implications of ascaroside biosynthesis on the impact of male exposure.

To provide a nuanced understanding of this dynamic, we turned our focus towards the ascarosides, ascr#10, and ascr#18, which are predominantly featured in the biochemical profile of *acox-1(ok2257)* hermaphrodites. Importantly, ascr#10 is the predominant ascaroside excreted in large amounts upon sexual maturation by males, whereas ascr#18 is produced constitutively by developing and adult hermaphrodites. Our findings establish that a 1 nanomolar concentration of ascr#10, facilitates an increase in exopher production without altering the number of embryos in the uterus, as illustrated in Fig. 3e-f. Contrarily, exposing reporter hermaphrodites to 1 picomolar and 1 nanomolar concentrations of ascr#18 led to a decline in exopher formation and reduced egg retention in utero, detailed in Fig. 3g-h. These observations underscore a differential regulatory role of these ascarosides in exopherogenesis, spotlighting the intricate connection between the chemical structure and activity of ascarosides.

While it might seem like a straightforward step to further this study using males of ascaroside biosynthesis mutants, we deemed it unsuitable due to the unknown metabolome of such males, which poses a significant limitation in interpreting the results and drawing substantial conclusions. Recent work demonstrated that males excrete vast number of sex-specific metabolites (Burkhardt et al., Nat. Commun. 2023), some of which have already been shown to affect hermaphrodite physiology (Ludewig et al., Nat. Chem. Biol. 2019 and Burkhardt et al., Nat. Commun. 2023), and it is unclear how production of these compounds is affected in peroxisomal 13-oxidation mutants. We agree that characterization of the metabolomic profile of males of peroxisomal 13-oxidation mutants and testing their effects on exopher formation would be fascinating; however, this will be a complex endeavor beyond the scope of the current study.

We believe that by demonstrating the potent effects of two example ascarosides, ascr#10 and ascr#18, we here significantly advance our understanding of exopherogenesis regulation while

highlighting the complex interplay of different ascarosides and likely other small molecule signals in this process.

20. ***The starving swimming populations are likely to make a different range of ascarosides from those on the well fed plate environments of most of the study. The authors should make more of a clear point on how different the ascaroside populations would be expected to be and better note this is an experiment in which exopher production is changed by a different chemical exposure, with distinctive associated ascarosides.*

Response: Your comment about the starving swimming populations making different ascarosides from well-fed plate environments is very insightful. While our study has begun to unravel the intricacies of ascaroside-induced exopher modulation, it does not cover all possible scenarios. The environmental conditions, including nutritional status, could lead to changes in chemical exposure and, consequently, distinctive associated ascaroside profiles, affecting exopher production differently. As you pointed out, we do not know precisely to what extent the pheromone profiles of the starving swimming population differ in ascr#10 and ascr#18 amounts and ratios, so we removed these data from the manuscript.

We value this observation as it opens a fascinating avenue for further exploration where the environment-induced pheromone variations and their role in regulating exophogenesis can be studied in depth. Despite their intriguing potential, we believe that presenting these data in the current version might detract from the core focus of our manuscript, introducing elements of uncertainty and speculation.

Remarks to Fig. 3 and associated text

Major points:

21. ***Fig. 3d,e,f--There is no description in Methods or legends on how the genetic ablations in specific neurons were executed. There is a hint from the strain list that cell specific caspase 1 was used, but the details are lacking. This information should be added to methods and described briefly in figure legends. Useful would be the addition of a few word titles to the figure panels that summarize what was done in the experiment.*

Response: Thank you for mentioning the lack of description regarding the methodology of genetic ablations in specific neurons. All the genetic ablations were previously described in the literature and are now properly cited in the Materials and Methods part of the manuscript, where we describe strains used in the study. We have also added a brief description to the figure panels to summarize what was done in the experiment, as suggested.

22. ***Fig. 3f study looks at neurons needed for normal upregulation of via co-culture with males. Since the model presented in the paper is that males are associated with egg retention, it is of value to test if the egg number phenotype correlates with the impact of neuronal ablations. The implication of early data is that exposure of male pheromones increases egg retention which enhances muscle exophers. These data provide an opportunity to separate/link egg retention from neuronal perception of male-produced pheromone. The authors should add measures of egg counts in these studies.*

Response: We agree with your suggestion on assessing the correlation between egg number phenotype and the impact of neuronal ablations. Indeed, our data in the revised manuscript directly address this point.

In the study, we observed that worms with genetically ablated neurons, which showed diminished exophogenesis (Fig. 4c), also displayed fewer eggs in utero (Fig. 4d). Additionally, the removal of ASK, AWB, or ADL neurons prevented the increase in the number of embryos *in utero* and nullified the enhancement in exopher production driven by male-emitted pheromones (Fig. 4f; Supplementary Fig. 3a). This indicates a direct link between egg retention, neuronal perception of male-produced pheromone, and exopher production, highlighting the essential role of ASK, AWB, and ADL neurons in this process. On the other hand, ascr#10 identified as a potent inducer of exophogenesis (as

shown in Fig. 3e), does not lead to a rise in the number of eggs retained in hermaphrodites (as demonstrated in Fig. 3f). This shows that the increase in exopher release resulting from exposure to pheromones occurs with (in case of *ascr#10*) or may occur without (other unidentified pheromones) the contribution of egg retention in the uterus. These conclusions have been incorporated into our model and are discussed in detail in the manuscript.

23. ***There is a disconnect with the earlier reported studies here, does male extract depend on ascaroside biosynthesis to work? If so, it is of interest to confirm whether the tested ablations are sensitive to that biology.*

Response: Thank you for pointing out the lack of clear confirmation of whether neuronal ablations are sensitive to ascaroside biology. To prove it, we have shown that *ascr#10* induces exopher production (Fig. 3e) and that this increase is mediated via STR-173 G protein-coupled receptor (Fig. 5f) expressed in ASK neurons (Fig. 5b). Next, when we exposed ASK ablation mutants to *ascr#10* we could not observe any change in exopher production (Fig. 5j). These results indeed show that there is a direct connection between ascaroside-dependent regulation of neuronal activity and exophergenesis.

24. ***Fig. 3g addresses temperature response in the level of muscle exopher production. There are concerns with text discussion of data. First, data for the control WT strain do not show an increase from 15 to 20oC so the summary statement lines 180 181 is not accurate. Second the statement that AWC removal “aggravates” the response also should be rewritten. Data show that there is a strong suppression of the muscle exopher level at each temperature in the absence of AWC. AWC does play a role in the response but a more precise description of outcome is needed.*

Response: We have revised the text to focus exclusively on the temperature response and the role of AWC neurons:

“When growing worms at 15, 20, or 25°C, we noted a temperature-related rise in exopher formation (Fig. 4e). This increase was inhibited in the absence of AWC, substantiating the temperature-dependent control of exophergenesis by these neurons (Fig. 4e).”

Minor points:

25. *Regarding hermaphrodite differential response for ascaroside mutants, the involvement can be confirmed by testing in ablation backgrounds.*

Response: Thank you for your constructive feedback. Indeed, testing the involvement of ascarosides in ablation backgrounds is a valid aspect of confirming their role in the differential response of hermaphrodites.

In our revised manuscript, we conducted experiments with hermaphrodites that had genetic ablations of different classes of olfactory neurons. We chose ASI and AWB ablation mutants as examples of neurons that are and are not, respectively, crucial for hermaphrodite pheromones-dependent decrease in exophergenesis. When we co-cultured them with ascaroside biosynthesis mutants, these neuronal ablation worms failed to exhibit the typical alterations in exophergenesis (Supplementary Fig. 3c, d). This further confirms that the ascaroside response is mediated through these specific olfactory neurons, as their absence led to a lack of response to the ascaroside biosynthetic mutants.

Remarks to Supplementary Fig. 2 and associated text

Minor point:

26. *Supplemental Fig. 2— identify the significantly changed transcript; one wonders why this was not chosen as a candidate mediator of exopher level change.*

Response: Thank you for your insightful observation regarding the significantly changed transcript in Supplementary Fig. 2 (now Supplementary Fig. 4). You are correct to question why this transcript was not chosen as a candidate mediator of exopher level change. We did not describe or experiment with this specific transcript in the current manuscript because the mutant associated with this gene exhibits intriguing phenotypes related to exophers. We believe this transcript and its associated gene warrant a more comprehensive and focused analysis, which is beyond the scope of the current study. Therefore, we decided not to expose this information in the current manuscript to avoid preemptive conclusions and to allow a more thorough investigation in our ongoing project.

Remarks to Supplementary Fig. 3 and associated text

Major points:

27. ***Supplementary Fig. 3b. Authors should add an explanation of what the gap-15 expression pattern is. The example image is strange in that red and green appear adjacent and non-overlapping, yet the conclusion is co-expression in the same cell. Is there an explanation? Or a more representative figure? Providing data on how typical this image is of how many observations would be helpful.*

Response: In our study, we used gap-15p::GFP as a marker for ASK, ASH, and ADL neurons. In the revised manuscript, we have included images with a higher level of zoom to better visualize the colocalization of STR-173 expression with ASK-expressed GFP from gap-15 promoter (Fig. 5b). These modified panels convincingly show that STR-173 is expressed in ASK neurons, providing a more representative and more transparent illustration of our findings.

28. ***What is the age of the animals scored for expression reporting? The animal exposure can be from L1-L4 and yet the responses are measured in adult—when does the GPCR signaling (or at least expression) take place, and in what cells is the receptor over this critical time period? The question is whether the receptor is in the right place at the right time to be a direct receptor as implied. Authors could figure that out to enhance mechanistic understanding.*

Response: We have taken additional images at the AD2 stage, a period where the receptor is necessary, and similar to the L4 stage, the transcriptional reporter indicates the presence of STR-173 in ASK neurons (Fig. 5a). This is critical as it demonstrates the presence of the receptor at a necessary time for signaling.

Additionally, our data shows that STR-173 is needed for transmitting signals from ascr#10, which implies that the receptor must be active at the right time and place to mediate the observed physiological responses. Specifically, we found that contrary to wild-type worms, there was a decrease in exopher quantity in *str-173* mutants exposed to ascr#10, emphasizing the role of STR-173 in translating male pheromone cues into physiological responses (Fig. 5f). Moreover, Fig. 5e demonstrates that the exposure to male scent does not lead to an increase in exophogenesis in *str-173* mutants. This underscores the crucial role of the receptor in identifying male pheromones and triggering the mechanisms that regulate exophers.

29. ***S3c. , text line 201. It is not clear the str-173 alleles are null alleles. The authors can easily expand the allele descriptions here to include a description impact on coding region. Both mutations appear to confer frame shifts, but indication as to whether shift is likely to extend to downstream coding region of the next exon*

or whether translation is likely to shift back in to frame consequent to the downstream splice event is important for assessing the likely severity of the alleles. This is especially important give that the two alleles do not confer exactly the same phenotypes.

Response: We appreciate the importance of clarifying whether the alleles we studied are null and the potential implications of the mutations on the coding region. We have once again investigated closely mutations and can confirm that both are null mutations. Moreover, to address this issue, we conducted an outcross of these lines and repeated all the experiments with backcrossed strains. Following numerous experiments using both *str-173* mutants, the results were consistently reliable. Notably, we observed a reduction in exopher quantity in *str-173* mutants exposed to ascr#10 (as shown in Fig. 5g), unlike wild-type worms, and there was no increase in exopher production after exposure to male pheromones (as demonstrated in Fig. 5e). These findings highlight the importance of STR-173 in interpreting male pheromone signals and producing physiological responses.

30. * *Supplementary Fig. 3e. It appears that solitary animals might exhibit modestly lower levels of muscle exophers when str-173 is disrupted, in addition to the failure to turn down levels in response to "social" exposure. Authors should address this with statistical tests and add to text.*

Response: We agree that assessing the role of *str-173* in solitary animals and the failure to downregulate levels in response to "social" exposure is important. However, the results of the experiment performed on backcrossed mutants show no significant difference in exopher number between *str-173* mutants and wild-type control (Fig. 5d). Appropriate statistical comparison is presented on the graph.

Minor points:

31. *Supplementary Fig. 3a, b legend, rather than having question marks indicate uncertainty of identification the authors should indicate likely identification based on position or whatever the assignment criteria was.*

Response: We appreciate your suggestion to clarify the identification of the *str-173* expression in Supplementary Fig. 3a, b (now Fig. 5a) legend. In the initial version of the manuscript, we used question marks to indicate uncertainty in identifying *str-173* expression in specific tissues. We take great care when describing the expression of a newly discovered gene to avoid any confusion for those who may use our research as a foundation. In order to be transparent, we indicate in the figure legend which neurons and non-neuronal tissues definitively express *str-173* and which ones we are describing its likely expression.

Revised Fig. 5a legend: "*str-173* 7TM receptor is expressed in neurons (ASK and probably[#] OLQ) and non-neuronal tissues (pharynx marked with an arrow, vulva, and probably^{##} rectal gland marked with circles). Square brackets mark gut autofluorescence. [#]based on the position and scRNAseq data cit. ^{##}based on the position and the shape."

32. *Supplementary Fig. 3d. Add to legend a description of what basal level reflects—one presumes standard culture of a mixed population.*

Response: We agree that the term "basal level" in the figure legend may not be self-explanatory to all readers. In the revised manuscript, we have updated the legend to clarify that the basal level refers to the standard culture condition of AD2 worms.

The revised legend for Fig. 5c (previously Supplementary Fig. 3d) reads as follows: The exophogenesis level in *str-173* mutants co-cultured with other hermaphrodites is unchanged compared to wild-type worms. n = 90; N = 6.

33. *They should note clearly to remind readers that standard mass growth is different from solitary growth which is different from a 10 animal culture. Does this receptor act at one or all? This is an interesting question.*

Response: Our study demonstrates the significant role of the STR-173 receptor in mediating exophogenesis in response to male pheromones and ascr#10. While our experiments mainly focused on the responses triggered by these specific cues, it is indeed an interesting question whether the receptor also acts in other contexts.

We have noted that the exophogenesis in *str-173* null mutants is comparable to wild-type controls when grown as solitary animals as well as in standard growing conditions (i.e., 30-50 age synchronized hermaphrodites per plate). This indicates that the receptor's activity is particularly relevant in the context of exposure to male pheromone and ascr#10. Under other conditions, the effect of STR-173 may be masked by other receptors for different ascarosides in the mixture.

We agree that understanding the receptor's activity under different growth conditions (mass growth, solitary growth, 10 animal culture, etc.) is an important and intriguing question, and our current study provides a foundation for exploring this in future research.

34. *Supplementary Fig. 3f. Given the changes in response to male conditioned plates, authors might consider examining expression of STR-173 in hermaphrodites co-cultured with males or their conditioned media.*

Response: Upon reflecting on your recommendation, we agree that analyzing STR-173 expression in the described settings would offer an enriched perspective on the interplay of different factors governing the observed phenomena.

Despite recognizing the value this addition could bring to our study, we encountered a substantial barrier in the current methodology, which is the utilization of the wrmScarlet protein as a reporter for STR-173 expression in our STR-173::SL2::wrmScarlet reporter strain. The wrmScarlet protein is characteristically stable, meaning that any changes at the transcript level may not be accurately reflected at the protein level. This is especially relevant as an increase in wrmScarlet expression can unambiguously indicate an elevation in the transcript level; however, the stability of wrmScarlet means that a lack of change in its levels can't conclusively delineate whether the transcript level remained constant or decreased, posing a risk of drawing misleading conclusions.

In light of these considerations, we consciously decided not to pursue this angle to prevent introducing data that might not accurately represent the underlying dynamics in the manuscript.

Remarks to Fig. 4, Supplementary Fig. 4, and associated text

Major points:

35. ***Fig. 4b, Supplementary fig. 4a,b—better to note that eggs and progeny production are modestly reduced; changes appear unlikely to themselves modulate exopher numbers associated with genetic ablations of AQR, PQR and URX. Thus, “reduced” seems an overstatement—there is not much difference that may matter here, which is fine!!!*

Response: We agree that the term "reduced" may be an overstatement, given the modest differences observed. In the revised manuscript, we have carefully rephrased our description to note that eggs and progeny production are "modestly reduced" rather than simply "reduced." We have also clarified that these changes appear unlikely to themselves modulate exopher numbers associated with genetic ablations of AQR, PQR, and URX, as you pointed out.

36. ***Fig. 4e would be stronger if the wild type no-ablation control in this experiment was included. The “double” ablation level is likely at WT levels.*

Response: In response to your comment, we have included the wild-type no-ablation control in the revised manuscript. This addition strengthens the conclusion drawn from Fig. 4e (now Fig. 6e).

37. ***Fig. 4c. The 60 minute neuronal inactivation/activation studies are quite interesting, but the implications on the timing are not discussed. The 60 minute disruption with capacity to increase exophers shortly thereafter implies a temporally tight functional connection between these neurons and the muscle exopher response. How do the authors think this works?*

Response: We appreciate your interest in the 60-minute neuronal inactivation/activation studies and the implication regarding the temporally tight functional connection between the AQR, PQR, and URX neurons and the muscle exopher response.

Our observations indeed suggest a rapid response mechanism. The significant increase in exopher release following a 60-minute inactivation and the marked reduction in exopher release post a 60-minute activation of AQR, PQR, and URX neurons underscore a temporally tight and dynamic regulation of exopher production by these neurons.

We speculate that this rapid response is mediated through the neuropeptides FLP-8 and FLP-21, which we identified as key exopher production regulators and are expressed predominantly in URX and AQR, PQR, and URX neurons, respectively. Our results demonstrate that these neuropeptides negatively regulate exopher production, as evidenced by increased exopher counts in both single and double mutants (Fig. 7a), independent of embryo-maternal signaling (Fig. 7b). Most importantly, activation of AQR, PQR, and URX neurons in wild-type worms leads to the production of exopher, which is mitigated in *flp-8* and *flp-21* single and double mutants (Fig. 7c-e). This shows that the signaling from AQR, PQR, and URX neurons via FLP-8 and FLP-21 can directly modulate exopher production in a relatively short time.

While our studies delineate the role of these neurons and neuropeptides in modulating exophogenesis, the exact mechanism of how this rapid signaling is transduced to the muscle cells remains to be elucidated. It is plausible that these neurons and neuropeptides exert their effect via a fast-signaling pathway that may involve other unidentified molecular players. Further studies will be needed to dissect the detailed molecular mechanisms and signaling pathways in this rapid response.

38. **Figure 4I—To test this model, the authors should show that “pathway specific” ADL, AWB influence egg retention.*

Response: In our revised manuscript, we have included new data that directly addresses your comment. We observed that worms with genetically ablated neurons, which showed diminished exophogenesis, also displayed fewer eggs in utero (Fig. 4c-d). Additionally, we found that the removal of ASK, AWB, or ADL neurons prevented the increase in the number of embryos in utero and nullified the enhancement in exopher production driven by male-emitted pheromones (Fig. 4f; Supplementary Fig. 3a).

Your suggestion to incorporate this information into the graphic exopher regulation model is well-taken. We understand the importance of differentiating between ascaroside-mediated and egg signaling pathways, especially given the observation that *ascr#10* at 1 nanomolar concentration increased exopher but not eggs. Accordingly, in the revised model, we have included two potential scenarios for ascaroside-mediated exopher production with and without the involvement of embryo-maternal signaling.

Minor points:

39. *Legend Fig. 4e and text-outcome of “double” ablation is a phenotype that is in between either single disruption. Each may contribute independently to a summed outcome. Discuss more clearly--The terms “counterbalanced” and “equalized” are vague and should be clarified.*

Response: Thank you for your feedback on the legend of Fig. 4e and the corresponding text. We agree that the terms "equalized" and "equilibrated" may not precisely convey the observed phenomena. In response to your feedback, we propose the following revision to the manuscript: Revised Legend: "The opposing exophergenesis levels observed in animals with genetic ablation of ASK neurons (low exophergenesis) and AQR, PQR, and URX neurons (high exophergenesis) converge to an intermediate level in animals with all four neurons removed. n = 90 - 106; N = 3." Revised Text: "Moreover, our data underscores that the contrasting exophergenesis characteristics observed in ASK-ablated worms and in worms with genetic elimination of AQR, PQR, and URX neurons resemble those of wild-type worms in animals lacking all four neuron classes (Fig. 6e)."

40. *Does the stage at which the activation/inactivation is delivered matter? (additional comments on clarification of exposure and outcome timing are given above).*

Response: Thank you for your question regarding the stage at which the ReaChR-based activation and ArchT-based inactivation is delivered. In the revised manuscript, we indeed clarify the importance of the timing of the activation/inactivation of AQR, PQR, and URX neurons. The experiment was started from the adult day 2 stage because, as our results indicate, the modulation of exopher release by ArchT-based inactivation was not evident at the adult day 1 (AD1) stage. This suggests that the stage of the worm does indeed matter, and the effect observed could be due to the potential significance of an optimal embryo count or direct uterus interaction.

To ensure clarity, we have provided the detailed timing in the manuscript: " To further validate the role of AQR, PQR, and URX neurons in the regulation of exophergenesis, we optogenetically inactivated or activated them using ArchT or ReaChR, respectively, and compared the number of exophers before and after the stimulus. We observed that 60 min of AQR, PQR, and URX neuron inactivation leads to a significant increase in exopher release (Fig. 6c and Supplementary Fig. 5c). On the other hand, 60 min of AQR, PQR, and URX neuron activation resulted in a significant decrease in exopher release after the stimulus was completed (Fig. 6d and Supplementary Fig. 5d). This modulation was evident at the adult day 1 (AD1) stage for ReaChR-based activation but not ArchT-based inactivation, underscoring the potential significance of an optimal embryo count or direct uterus interaction (Supplementary Fig. 7e-h)."

41. *Fig. 4f is on hermaphrodite impact, as measured by solitary vs. group rearing. An outcome that seems obvious (but no statistics are indicated in the panel) is that baseline muscle exophers in the solitary and group are both elevated consequent to ablation. This can be interpreted to indicate that AQR/PQR/UBX action normally inhibits baseline exophers which should be better pointed out. High baseline is also evident in the ablation + male scent assay. Hermaphrodite downregulation in the G10 condition is lost—AQR, PQR, UBX might mediate the normal suppression by hermaphrodites as well, which the authors conclude.*

Response: Thank you for your insightful comments and observations. Indeed, our results indicate that AQR/PQR/URX neurons play a significant role in inhibiting baseline exophers, as evidenced by the substantial increase in exophergenesis upon the removal of these neurons (Fig. 6a). Additionally, the increased number of exophers in genetically ablated AQR, PQR, and URX neurons is independent of embryo-maternal signaling (Fig. 6b), reinforcing the notion that these neurons have a direct inhibitory effect on exopher production.

Moreover, the loss of hermaphrodite downregulation in the 10 animal conditions suggests that AQR, PQR, and URX might mediate the normal suppression by hermaphrodites. This is further supported by our finding that these neurons are involved in response to hermaphrodite pheromones (Fig. 6f) but not to male pheromones (Fig. 6g).

In summary, our data underscore the complexity of neuronally-regulated reproductive signaling and highlight the distinct roles of AQR, PQR, and URX neurons in modulating exophergenesis in response to hermaphrodite pheromones. We have attempted to clarify this in the revised manuscript, and we appreciate your feedback in helping us clarify this.

42. *Fig. 4J,I model—where does STR-173 fit in these models? Minimally there has to be some note/discussion of this issue. Mutant str-173 alleles do not appear to impact baseline, might mediate the hermaphrodites 9/1 suppression, might influence the male conditioned medium response. Baseline impact should be discussed as well as what the data might imply for the specific outcomes assayed.*

Response: The reviewer has raised a valid point regarding the positioning of STR-173 in the models depicted in Fig. 4J, I. Our additional data on the STR-173 receptor clarifies all of the questions mentioned above and shows that STR-173 is one of the key factors in inducing exopherogenesis by male pheromones, specifically via *ascr#10*. These findings are now incorporated into our model presented in Fig. 8.

Discussion

Major points:

43. ***Line 255. Although the authors document changes in muscle exopher production in response to environmental conditions, the leap to assuming exophers play a likely role in inter-animal communication impresses as a step too far. Ascarosides or other chemicals are implicated in communication by data but exopher exchange, per se, is not; this aspect of the discussion should be toned down.*

Response: We thank you for your insightful comments. We recognize that our initial discussion may have overextended the role of exophers in inter-animal communication, and we have revised the model and discussion sections accordingly.

Revised Model Section: Our revised model underscores the role of male-derived ascaroside pheromones and hermaphrodite volatile signals in modulating exopher production via the STR-173 receptor and the ASH, ASI, and ASK olfactory neurons. We have toned down the implication of exophers in inter-animal communication, focusing instead on their potential role in resource allocation towards the germline and somatic health. Additionally, we highlighted the role of FLP-8 and FLP-21 neuropeptides and their associated neurons, URX and PQR, in regulating muscle exopher production.

Revised Discussion: In our revised discussion, we have carefully reevaluated the role of exophers in inter-animal communication. We emphasize the strategic adaptation of *C. elegans* in response to reproductive endeavors and potential resource scarcity, indicated by the opposing effects of male pheromones and high hermaphrodite density on exopher production. We also discuss the potential implications of our findings on understanding diseases like Alzheimer's or Parkinson's, where protein aggregation is a prominent concern. Finally, we acknowledge the potential parallels between our findings in *C. elegans* and similar mechanisms in humans, while recognizing the need for further research to ascertain these connections.

44. ***Line 262. It is an overstatement to say that the authors have shown that neuroendocrine signals are secreted; the authors showed AQR/PQR/and URX negatively regulate baseline but the molecular mechanism is not addressed. Vesicle release by these neurons is not documented. Authors should revise discussion on this point.*

Response: In response to the reviewer's criticism, we acknowledge that it is an overstatement to say that neuroendocrine signals are secreted solely based on the regulation of baseline by AQR, PQR, and URX neurons, as the molecular mechanism was not addressed in the initial discussion, and vesicle release by these neurons was not documented. In the revised discussion, we refrained from making broad claims about neuroendocrine secretion and focused on the roles of specific neuropeptides and neurons identified in our study.

Revised Discussion Section: The revised discussion delves into the nuanced roles of various molecular, neuroendocrine, and pheromonal elements in regulating exophogenesis in *C. elegans*. Our findings highlight the pivotal roles of male-derived ascarosides, hermaphrodite volatile signals, and specific sensory neurons AQR, PQR, and URX and their associated neuropeptides, FLP-8 and FLP-21, in modulating muscle exopher production. The discussion then expands on the implications of these findings, considering the trade-offs between reproduction and somatic well-being, the adaptations in response to social stresses, and the potential relevance of our findings to protein aggregation and clearance in neurodegenerative conditions. Moreover, we speculate on the broader implications of our work, pondering the existence of analogous mechanisms in more complex organisms and the potential implications for understanding diseases like Alzheimer's or Parkinson's.

Minor point:

45. *The Discussion is quite speculative and could benefit from addition of a summary clarification of what can and cannot be concluded about multiple responses and factors in the external environment and their impact on muscle exophers. Hermaphrodite and male influences, how much is actually shown to require proper ascaroside synthesis, str-173, eggs? Can male conditioned medium work but not hermaphrodite? The authors might consider a summary table as there are multiple implications of data and multiple pathways, but the precise model for a given influence does not easily emerge.*

Response: We greatly appreciate the reviewer's feedback and agree that our discussion does appear speculative due to the complexity of the pathways involved and the multiple influences at play. To address this, we have added new data addressing some of the reviewer's concerns, revised the discussion section, included a new model section detailing the mechanism of exopher regulation, and made a summary table with all the observed experimental outcomes as suggested.

In the revised discussion, we have further elaborated on the complexities of exopher regulation in *C. elegans*, shedding light on the vital roles of male-derived ascarosides and hermaphrodite volatile signals. We discussed the strategic evolutionary adaptation of *C. elegans* to allocate resources in response to reproductive endeavors and the potential trade-offs between reproduction and somatic well-being. We also addressed the contrary effects of male proximity and hermaphrodite density on exophogenesis and highlighted the importance of understanding the hierarchical dynamics of these cues under diverse conditions. Furthermore, we delved into the potential broader biological significance of our findings, particularly concerning protein aggregation in neurodegenerative conditions. Finally, we suggested possible parallels with human physiology and areas for future exploration, such as the potential role of olfactory-driven exopher regulatory mechanisms in cardiovascular disease.

Overall, the revised discussion provides a comprehensive and nuanced analysis of our findings while addressing the reviewer's concerns about the speculative nature of our initial discussion.

Reviewer #2 (Remarks to the Author):

Banasiak et al. investigated how secreted metabolites that are possibly related to ascaroside pheromone affect the genesis of large extracellular vehicles (exophogenesis), a biologically important process that remains very poorly understood. The authors further report the involvement of GPCR STR-173 and sensory neurons AQR, PQR, and URX.

Major points:

46. *In the first two sections, the authors showed that worms grown in the presence of male secretome had an effect on exophogenesis that is opposite to that of hermaphrodite-derived secretome, and further, that*

exophergenesis is affected by peroxisomal beta-oxidation genes and secreted molecules. These observations suggest that ascaroside pheromones are involved, since ascaroside biosynthesis requires peroxisomal b-oxidation. However, ascarosides are a highly diverse class of molecules - more than 100 different structures have been reported, many of which are produced in a sex-specific manner. In addition, the production (and possibly their secretion) many other lipids likely depends on peroxisomal b-oxidation. Therefore, whether ascarosides are involved or not must be tested. Based on the existing knowledge of C. elegans sex-specific ascaroside biosynthesis, the authors could have developed testable hypotheses on the molecular identities of potentially involved ascarosides to perform validation assays. Without such validation experiments, I don't think any firm conclusions can be drawn from the data presented here. Importantly, the observation that maoc-1 mutants display a reduction in exopher production, whereas the daf-22 and acox-1 mutants display an increase, may actually speak against the involvement of ascarosides, since most production of ascarosides is abolished in both maoc-1 and daf-22 worms. Thus, while the data clearly show that peroxisomal b-oxidation is involved, it is entirely unclear whether ascarosides are involved or not.

Response: Thank you for your thoughtful evaluation. You are correct in noting the diversity of ascarosides and the possibility that other lipids dependent on peroxisomal 13-oxidation could be involved. Indeed, our new results section is aimed at addressing this very concern.

Our results specifically show that two different ascarosides, ascr#10 and ascr#18, oppositely regulate exophergenesis (Fig. 3). While we acknowledge that there may still be other types of ascarosides (as well as other molecules dependent on peroxisomal 13-oxidation) involved in this process, our manuscript in its current version convincingly show that ascaroside signaling is involved in the regulation of exophergenesis.

Minor points:

47. *Though only included in a supplementary figure, one of the most interesting claims in this manuscript is that STR-173 may be involved in sensing of "pheromone" (though it's unclear whether this is ascarosides). However, to support that STR-173 functions as a GPCR sensor in the exopher context one would expect more molecular evidence – at least testing isolated metabolites or synthetic candidate compounds.*

Response: Thank you for your interest in our findings surrounding STR-173. We value your input and agree that further elucidating the role of this receptor can substantially fortify our study's conclusions.

In response to your suggestions, we took it upon ourselves to delve deeper into exploring STR-173's role in exopher regulation in our recent experiments. It led us to a significant breakthrough where we could affirm that STR-173 binds to the male pheromone ascr#10, a binding that interestingly stimulated an increase in exopher production in hermaphrodites. This outcome, achieved through testing with synthetic candidate compounds, not only confirms STR-173's pivotal role as a sensor in this context but also unveils it as a novel receptor for ascr#10, one of the central ascarosides produced by *C. elegans*.

This finding significantly elevates our understanding of the mechanisms underlying exophergenesis, pinpointing a clear pathway of pheromone sensing that is pivotal in regulating this process. We are excited to include these robust findings in our revised manuscript, illustrating a more detailed pathway whereby pheromones can mediate exopher production through STR-173.

48. *STR-173 does not appear to be expressed in the relevant neurons, therefore a logical connection is missing.*

Response: Thank you for your comment concerning the expression of STR-173 in relevant neurons. We recognize the critical importance of demonstrating a clear link between the receptor and the neurons implicated in the response to pheromones. In our study, we used gap-15p::GFP as a marker for ASK, ASH, and ADL neurons. In the revised manuscript, we have included images with a higher level of zoom to better visualize the colocalization of STR-173 expression with ASK-expressed GFP from gap-15 promoter (Fig. 5b). These modified panels convincingly show that STR-173 is expressed in ASK neurons, providing a more representative and clearer illustration of our findings.

49. Fig. 2a: using ‘very long chain ascarosides’ as the starting point might match better in this case.

Response: Thank you for your suggestion regarding Fig. 2a (now Fig 3a). We have considered your feedback and included “very long chain ascarosides” in the figure legend.

Reviewer #3 (Remarks to the Author):

*This is an exciting paper on the role of exosensory neurons in ascaroside signaling. The authors discovered that the hermaphrodites grown with hermaphrodites produce fewer exophers, but have the same number of eggs in utero. Conversely, hermaphrodites grown with males or on a male-conditioned plate produce more exophers and also produced more eggs in utero. These experiments allowed the authors to conclude that exopher formation is influenced by the pheromones worms sense. Since pheromones influence exopher production, the authors wanted to see if genes involved in ascaroside synthesis, *maoc-1*, *daf-22*, and *acox-1* affect exopher production. The authors found that *maoc-1* worms had decreased exopher and egg production while *daf-22*, and *acox-1* worms had an increase in both exopher and egg production. This caused the authors to conclude that MAOC-1 plays a role in exopher-mediated pheromone synthesis. Since neurons involved in ascaroside detection are known, the authors wanted to see if these neurons also mediate pheromone-induced exopher production. They additionally discovered that the ASK, AWC, and ADL are the main neurons required for normal exopher production. The authors also found that the ASK, ASH, and ASI neurons were required for a hermaphrodite pheromone-induced decrease in exopher production whereas the ASK, ADL, and AWB were required for male pheromone-induced increase in exopher production and identified *str-173* as a candidate gene because it was differentially expressed in worms grown alone and worms grown with conspecifics.*

Overall, this paper supports that exopher formation is influenced by the ascarosides a worm is sensing, and this effect may be related to the role that ascaroside signaling plays in reproduction.

I RECOMMEND THE PAPER FOR SOME REVISIONS.

50. In figure 1, the authors indicate that there is a relationship between pheromone exposure, exophogenesis, and embryogenesis. The exposure to male pheromones increases both exopher formation and egg count (Fig 1e-j). Since various sensory neurons are implicated in the male pheromone dependent increase (ASK, ADL, and AWB) of exopher formation, do these same neurons affect embryo formation too? If these neurons were found to also modulate the increased egg production upon exposure to male pheromones, I think it would help support that these phenotypes are related.

Response: Thank you for your insightful question regarding the relationship between sensory neurons, exophogenesis, and embryo retention in the uterus. As suggested, we have verified whether sensory neurons that regulate exophogenesis also regulate in utero embryo retention. We observed that worms with genetically ablated neurons, which showed diminished exophogenesis, also displayed fewer eggs in utero (Fig. 4c-d). Additionally, we found that the removal of ASK, AWB, or ADL neurons prevented the increase in the number of embryos in utero and nullified the enhancement in exopher production driven by male-emitted pheromones (Fig. 4f; Supplementary Fig. 3a). On the other hand, exposing worms to *ascr#10* at 1 nanomolar concentration increased exopher production (via ASK-expressed STR-173 GPCR) but not eggs retention. Accordingly, in the revised model, we have included two potential scenarios for ascaroside-mediated exopher production with and without the involvement of embryo-maternal signaling.

51. In Figure 2a, there is a schematic showing how each of the genes tested is involved in ascaroside synthesis. This figure implies that each of these genes functions in the same pathway. If this were true, it would seem that all of the mutant secretions would create the same phenotype. Could you add text explaining why you think why the secretions of each mutant creates different exopher numbers (fig 2e) when all three mutants are deficient in ascaroside production? In other words, what is a specific difference between the mutant secretions could these results (in fig 2e) be attributed to?

Response: Thank you for pointing out the potential confusion arising from Fig. 2a (now Figure 3a). To offer a more detailed representation, we have revised Fig. 2a to portray shifts in the ascaroside profile brought on by the different mutants, essentially to illustrate that the ascaroside constellation is modified rather than completely abolished, hence affecting muscle exopherogenesis in diverse ways. We have also added the explanation mentioned above to the manuscript. Moreover, we now discuss changes in male-enriched ascaroside ascr#10 levels in these mutants, as this ascaroside is a key factor in regulating male-induced exopher production.

52. In Figure 3d-3f, the authors examine how ablating various cells affects the exopher response. They also show how masculinizing the hermaphrodite nervous system with the addition of the CEM neurons changes exopher response. I would prefer if the CEM data were separated because it feels like it's answering a different question with this data (which part of the male nervous system is required for exopher response?) Whereas the ablation data is trying to answer which part of the hermaphrodite nervous system is involved in exopher pheromone response. With this, if you're going to include the CEM data, it might make sense to just include the WT male response because I'm not really sure what that would look like as a baseline. If you see that WT males and hemaphrodites with CEMs have the same response, then you can conclude that the CEM neurons are required for exopher response in males.

Response: Thank you for your thoughtful feedback on our presentation of data in Figure 3d-3f. Addressing your concerns:

- 1. Differentiation between CEM data and ablation data:** We concur with your observation that the CEM and ablation data cater to different scientific inquiries. The intent behind presenting them in proximity was to provide a comprehensive perspective on how many olfactory neurons, including male-specific CEM neurons, influence exopher response. Specifically, our results indicate that while the introduction of CEM neurons to hermaphrodites did not change exopher levels in a general context (Fig. 4c), their presence in a masculinized olfactory circuit led to a distinct outcome, with decreased exopher production in solitary animals and no further decrease when the population density was increased (Fig. 4g). This was an essential point for us to convey, showing that the CEM neurons' role is not solely based on their presence but is context-dependent within the neural circuitry.
- 2. Including the WT male response:** We acknowledge your recommendation regarding the potential inclusion of the WT male response for clarity. Our current presentation focused on exploring and comparing the effects within hermaphrodites, especially in the context of CEM introduction and neuron ablation. While a comparison with WT males, who do not produce exophers at all (Turek et al. EMBO Reports, 2020), might offer additional insights, we intended to shed light on the complexities within hermaphrodites, given the unique approach of introducing male-specific neurons.

In summary, while we recognize the potential benefits of separating the datasets more distinctly or including additional comparisons, our current presentation aims to offer an integrated perspective, considering the intertwined nature of the underlying mechanisms.

53. In the intro, could there be more text about how exophers are related to reproduction? It would be helpful to have more background on this when reading the paper (unless more is not known).

Response: Thank you for your feedback and suggestion to elaborate on the relationship between exophers and reproduction in the introduction.

Your suggestion is valid. The interplay between exophers and reproduction is indeed a fascinating and complex area. As you rightly observed, there isn't a vast repository of literature on the topic as of now. We have, however, tried to touch upon the crux of it.

In our introduction, we have stated:

" In our previous work, we showed that the body wall muscles (BWMs) of *C. elegans* release exophers that transport muscle-synthesized yolk proteins to support offspring development, increasing their odds of development and survival⁶. However, how exopherogenesis is regulated in response to external factors impacting animal development and reproduction is unclear."

This statement encapsulates the key role exophers play in reproductive processes, particularly concerning the developmental support they offer to offspring. Moreover, it hints at the complex interplay between embryonic signaling, the presence of developing embryos, and exopher production. While we recognize that a more in-depth discussion could further enrich the introduction, given the limited available literature and our current understanding, we have tried to convey the essence of the relationship between exophers and reproduction as comprehensively as possible. In its current form, we hope our introduction provides readers with a sufficient foundation for understanding the subsequent sections of our study and the larger context within which our findings fit.

Reviewer #4 (Remarks to the Author):

54. This work demonstrates that signals released to the environment by conspecifics regulate the production of muscle exopheres in C. elegans hermaphrodites. While signals from hermaphrodites decrease exosphere production, signals from males increase production. Furthermore, the authors identify 2 distinct groups of sensory neurons mediating either signal. Overall, I don't feel that the results presented here contribute significantly to advance the field of extracellular vesicle communication. The mechanisms by which the signals affect exosphere production are unknown. In addition, it is not clear at all what the adaptability of this decrease or increase would be according to the sex of the conspecifics.

Response: Thank you for the feedback on our manuscript and for drawing attention to the importance of outlining the implications and significance of our findings.

Based on our latest investigations, we've developed a comprehensive model that elaborates on the multifaceted mechanism of exopher regulation in *C. elegans*. Here is a summary that directly addresses your concerns:

1. Male pheromones: Males produce ascarosides, including ascr#10, that increase exopher production levels. These signals predominantly act through ASK, ADL, and AWB sensory neurons. The critical role in this pathway is played by the ASK-expressed STR-173 G protein-coupled receptor, which binds ascr#10 to potentiate exopher production.
2. Hermaphrodite pheromones: Hermaphrodites release ascarosides, including ascr#18, that reduce exopher production levels. These signals predominantly act through ASI, ASH, and ASK (via ascr#18) sensory neurons.
3. Embryo accumulation effect: The rise in embryos inside the hermaphrodite triggers a series of pro-exopher signals. Significantly, increased in utero embryo accumulation can be mediated by a blend of male-released pheromones.
4. Neuropeptide control: AQR/PQR/URX pseudocoelomic cavity-opened neurons release FLP-8 and FLP-21 neuropeptides that negatively regulate exopherogenesis. This modulation type is important for the decrease in exopherogenesis dependent on hermaphrodite pheromones but not for the increase in reliance on male pheromones.
5. Integrated response: Exopher production is modulated by a blend of internal processes and external cues, whether from male or hermaphrodite secretions. This balance ensures that exopherogenesis is regulated in tandem with various internal and external environmental conditions.

While the direct mechanisms behind exopher communication may seem specific to *C. elegans*, the broader implications of our work extend far beyond. This study not only delves into the minutiae of cellular processes in response to internal and external cues but also sets the foundation for understanding how such processes may manifest in other, more complex organisms. Fig. 8, as referenced, provides a visual summary of this intricate regulatory system.

In response to the concerns regarding adaptability and the role of conspecific signals, our research indeed postulates that *C. elegans* has developed sophisticated mechanisms to modulate cellular processes based on conspecific cues. This adaptive response ensures that the worm can optimize its cellular activities based on its immediate environment, which likely impacts reproductive success and overall survival.

Our study provides pivotal insights into the interconnectedness of reproductive signals, sensory neurons, and cellular processes in *C. elegans*. These findings can potentially deepen our understanding of nematode biology and shed light on the possible evolutionary significance of extracellular vesicle communication in the animal kingdom.

We hope our response offers a clearer perspective on the relevance and implications of our work.

55. The data in this manuscript should be pretty straight forward to follow and understand. However, the paper is written in such an unclear manner that is extremely hard to interpret the experiments performed and follow the logic and conclusions of the authors. An example in point is all the data in figure 1. The manipulations carried out are very unclear to me. On growing worms in presence of males and having an impact on exospherogenesis: why do the authors say male scent? It may be contact by males during development, or mating in adulthood, etc. at this stage of the data presented they don't know what aspect of male presence is the cause. Is it having male essence after L4? Or is it growing with males? When is the effect produced? I just don't understand what they did in line 117, are they changing genotype (now N2) and protocol (removing the males which they didn't with him-5 mutants)? Then you can't distinguish between the two.

I think the sentence "Growing hermaphrodites on male-conditioned plates increased exophere production to the same degree as when hermaphrodites were grown with males until the L4 larvae stage (Fig. 1i)" doesn't make sense because they condition the plates by growing herms with males, so the sentence should say "growing N2 herms with males increased exophere production to the same degree as when growing him-5 hermaphrodites", right? Otherwise, I don't understand what they did or how many factors they changed. And how is this different from "Furthermore, adult hermaphrodites exposed to males' secretions as larvae showed no further increase in exophere production" ?

Response: Thank you for the detailed feedback on our manuscript. We appreciate the time taken to review our work and would like to address the concerns raised.

- 1. Clarity of Writing:** The clarity of a manuscript is paramount, and we apologize for any confusion caused. We intended to present the data in a manner that best captures the essence of our findings, and we believe the results are presented accurately. Moreover, we have rewritten the manuscript in such a manner that it first focuses on male-related regulation of exophers formation, followed by the data on hermaphrodite regulation of exopher formation. Next, we present the data on molecular mechanisms responsible for exopher formation. Finally, we prepared a table (Supplementary Table 2) summarizing all phenotypes presented in the manuscript and their regulators. We believe that this makes the manuscript easier to follow and understand.
- 2. On the Term 'Male Scent':** The terminology "male scent" was employed to denote that in this experiment, hermaphrodites were grown on a plate that had previously housed males. After 48 hours, the males were removed, leaving only secreted substances associated with males. This mixture of substances was referred to as "male scent." Therefore, observed an increase in exopher production in hermaphrodites could not be attributed to contact with males during development or mating in adulthood. Instead, we have concluded that the hermaphrodites could sense substances left by males on the plate.
The data in Figure 1 was not meant to isolate a specific factor of male influence but rather to illustrate a detectable change in exospherogenesis due to some substances secreted by males.

We acknowledge that further experiments are required to determine the exact nature of this influence. These data are presented in the following parts of the manuscript.

3. **Explanation of Figure 1:** Regarding line 117 and the experiments involving N2 and *him-5* mutants, our objective was to understand if there's a genotype-specific response to male presence or male-conditioned plates. It's crucial to differentiate between the presence of males and the residual effects of a male-conditioned environment. The N2 and *him-5* mutants serve to demonstrate that the observed effect is consistent across genotypes and that the findings are not an artifact of a particular experimental condition. Considering your comment, in the current version of the manuscript in all the figures, we have also made it clear what genotypes have worms used in the experiments.
4. **Clarification on the Challenged Sentence:** The statement, "Growing hermaphrodites on male-conditioned plates increased exophers production to the same degree as when hermaphrodites were grown with males until the L4 larvae stage (Fig. 1i)" highlights the overarching theme of our experiments: that the environment conditioned by male presence (without the physical presence of males) has a similar impact on exophers production as direct male-hermaphrodite cohabitation until L4 larval stage. The difference between the two scenarios you have pointed out emphasizes that the effect is observed during developmental stages and not necessarily post-maturity. Through additional experiments, we have identified the developmental stage at which male influence (via pheromones) on exophogenesis in hermaphrodites is most pronounced (Fig. 1h).
5. **Differentiating Experiments:** We acknowledge that we have miswritten the sentence "Furthermore, adult hermaphrodites exposed to males' secretions as larvae showed no further increase in exophers production," we are sorry for any confusion it has caused. For the part mentioned above in the manuscript, we have presented an experimental scheme demonstrating how it was performed in its current version. Moreover, we have clarified this sentence, and it reads: "Longer exposure to male-conditioned plates or co-culture with males beyond larval development did not further increase exopher production in *him-5* mutant hermaphrodites (Supplementary Fig. 1a-c)."

Finally, with additional experiments, we found that 24 hours of exposure to males' pheromones, regardless of the life stage at which the exposure occurred, was sufficient to increase exophogenesis in hermaphrodites. This was observed even when hermaphrodites were exposed only during the larval development, indicating that exposure to male pheromones during the larval stage can influence exopher levels in adult hermaphrodites. Specifically, the most potent effect was observed in animals exposed during the L4 larval stage to young adults' day 1 stage, mirroring the effects of continuous exposure to males' secretions (Fig. 1h-i).

In summary, our experiments and the associated results offer valuable insights into the influence of male presence on exophogenesis. We hope this response clarifies our experimental logic and conclusions.

56. It is also unclear why the authors purify/extract secretions from starving animals in dauer to mimic the signals from well-fed adult conspecifics that regulate exosphere production in their essays.

Response: Thank you for your insights. Our experiment with isolated extracts from starved animals aimed not to show that it mimics the signal from well-fed adult conspecific but to obtain yet another proof that substances secreted by *C. elegans* modulate exopher formation. However, in reference to your concerns, we decided to remove these results from the current version of the manuscript as presenting these data in the current version might detract from the core focus of our manuscript, introducing elements of uncertainty and speculation. Because as you pointed out, for instance, we do not know the pheromone profile of the starving swimming population, especially in terms of *ascr#10* and *ascr#18* amounts and ratio, which we show are crucial in exopher regulation.

57. The results with the mutants in ascaroside production are also complex to interpret. Is the effect on exosphere production due to an increase in long-chain ascarosides or a decrease in short-chain ascarosides (both result from manipulating the synthesis pathway)?

Response: Thank you for highlighting the complexities surrounding the influence of ascaroside chain length on exopher production. We appreciate the opportunity to explain our findings further.

From our experiment (using a solitary exophers reporter worm plus nine mutant hermaphrodites, either *acox-1(ok2257)*, *maoc-1(ok2645)*, or *daf-22(ok693)*):

1. ***acox-1* mutants:** These are characterized by high levels of medium-length *ascr#10* and significant amounts of *ascr#18*. In our experiments, *acox-1* mutants demonstrated a notable increase in exopher levels. Given the prominence of *ascr#10* in male pheromone profiles and its elevated presence in *acox-1* hermaphrodites, we determined its potent role in stimulating exopher production. However, it's essential to understand that the presence of *ascr#18* alongside *ascr#10* in *acox-1* mutants can have modulating effects.
2. ***maoc-1* mutants:** These mutants produce *ascr#18*, albeit in moderated amounts, but produce only minimal amounts of *ascr#10*. Our observations showed a decrease in exopher production when exposed to the *maoc-1* secretome, underlining the inhibitory role of *ascr#18* in the absence of the stimulatory *ascr#10*.
3. ***daf-22* mutants:** These animals only produce very long-chain ascarosides. Interestingly, co-culture with *daf-22* animals showed no discernible change in exopher levels, suggesting that very long-chain ascarosides, in our experimental conditions, didn't have a pronounced impact on exopher production.

Furthermore, our direct assays with synthetic *ascr#10* and *ascr#18* reinforced their contrasting roles in exophogenesis. Exposure to *ascr#10* significantly promoted exopher production, while *ascr#18* exhibited an inhibitory effect, even at low concentrations.

Collectively, our findings underscore a delicate balance between *ascr#10* and *ascr#18* in regulating exopher production. In situations like the *maoc-1* mutant, the absence of stimulatory *ascr#10* coupled with the presence of inhibitory *ascr#18* leads to reduced exophogenesis. Conversely, the elevated levels of both ascarosides in *acox-1* mutants might create a favorable balance, pushing towards increased exopher production mediated by *ascr#10*. In addition, we cannot exclude that additional ascarosides and other metabolites dependent on peroxisomal β -oxidation play a role in exopher formation, and we added this caveat to the Discussion section.

Additionally, based on your valuable comments, we have reworked much of the manuscript to make it more intuitive and straightforward for readers. We hope our revisions provide clarity and enhance the comprehensibility of our findings.

58. In all the experiments of sensory neuron manipulation the authors should just present the data set where they compare mutant animals raised singly and mutant animals raised with 10 others. This is the correct experiment to assess the contribution of sensory neurons to sensing the exopheregenesis-regulating signal without having confounding autonomous effect of sensory neurons on exosphere production. The presence of the other data set where they assess exopheregenesis in a mutant population does not add any useful information and makes data interpretation confusing.

Response: Thank you for your thoughtful feedback regarding the presentation of our data on sensory neuron manipulation. We appreciate the emphasis on clarity, especially when interpreting complex datasets. Based on your comments, we have taken time to address your concerns:

1. We agree that comparing mutant animals raised singly and mutant animals raised in ten hermaphrodite populations will exclude the autonomous effect of sensory neurons on exopher production.
2. On the other hand, performing similar experiments on hermaphrodites exposed or not exposed to male pheromones allows us to assess which sensory neurons are mediating males-dependent increase in exophogenesis without having a confounding autonomous effect of sensory neurons on exopher production.

3. Finally, the data set where we assess exophergenesis in a neuronal mutants population broadens the perspective on how ablation mutants of specific neurons, by their inherent characteristics, influence exopher production and *in utero* eggs retention.

We believe this segmented approach provides a more detailed and structured narrative on the complex interplay of sensory neurons, pheromones, and exopher production.

To ensure that the information is communicated as clearly as possible, we are meticulous in our graphical representations, ensuring that the distinctions between various experimental setups are well-demarcated.

We hope this clarifies our rationale and methodology. We are confident that these considerations will help to eliminate any potential ambiguities and lead to a more precise interpretation of our findings.

59. *In figure 3d, why is ASH ablation significantly different from control but AWB ablation is not? The graph would suggest otherwise.*

Response: Thank you for your insightful observation on Figure 3d. We understand the source of confusion based on the graphical representation. The distinction between the ASH ablation and the control is primarily due to the overall shift in the distribution of exopher quantities under ASH depletion conditions. While the graph might give the impression that AWB ablation exhibits a significant difference, it is important to note that statistical significance is determined not just by the apparent difference in means or medians but also by factors in the spread and distribution of the data.

To better represent this distribution and clarify the results, we have modified the graphical presentation of the data. This provides a clearer visual cue about the underlying data distributions and help distinguish the effects of neuron ablation more effectively.

60. *And why do the authors single out the effect of CEMs in the text to say that adding CEMs leads to a reduction in exosphere production in single animals? It also happens by removing ADL, ASK, AWC but they do not mention it. In addition they need statistical analysis comparing to control if they want to make that claim on CEMs.*

Response: Thank you for your comment regarding the singling out of the effect of CEMs in the text. Upon reflection, we acknowledge that our manuscript did not adequately comment on the influence of adding CEMs to the hermaphrodite nervous system compared to other data. In the current version of the manuscript we have made it consistent with a description for other neuronal ablation and have stated that: "Remarkably, the masculinization of a hermaphrodite olfactory circuit through the introduction of CEM male-specific neurons resulted in an absence of exopher production decrease in ten hermaphrodite population comparing to solitary worms (Fig. 4g)".

61. *Also, why do they say ASK has less of a role for pheromone response than ASH and ASI? How do they reach that conclusion?*

Response: Our conclusion was based on the analysis of the results from experiments where we grew hermaphrodites with genetically ablated different classes of olfactory neurons as a single worm on a plate or in ten-hermaphrodites population (Fig. 4g). Contrary to ASH and ASI neuronal mutants in ASK neuronal mutants we could see a non-significant decrease in exopher production. However, we agree with the reviewer that such an observation is insufficient to conclude that ASH and ASI neurons control hermaphrodite pheromone-driven decrease in exophergenesis more than ASK neurons. Therefore, in the current version of the manuscript, we have replaced the original sentence with: "Our analysis with strains showing impaired olfaction revealed that ASH, ASI, and ASK neurons are pivotal in this process (Fig. 4g)"

62. Finally, regarding the role of AQR, URX and PQR in exopheregenesis, their data suggests that the effect goes through egg production, as egg production is required for these neurons to increase the production of exopheres (Fig 4h and i). Please state this accordingly.

Response: Thank you for highlighting the connection between AQR, PQR, and URX neurons and their impact on exopheregenesis through egg production. In light of your feedback, we would like to clarify our findings and articulate their implications more explicitly.

1. Genetically ablating AQR, PQR, and URX neurons led to a marked increase in exopheregenesis (Fig. 6a). However, this wasn't due to embryo-maternal signaling, as these worms had fewer eggs in utero than wild-type controls (Fig. 6b).
2. Optogenetics further established these neurons' role: inactivation increased exophers, while activation decreased them (Fig. 6c, d).
3. AQR, PQR, and URX neurons respond to hermaphrodite, but not male pheromones, and cannot bypass fertility role in exopheregenesis (Fig. 6f-i).
4. Mechanistically, FLP-8 and FLP-21 neuropeptides, linked to these neurons, negatively regulate exopher production, independent of embryo-maternal signaling (Fig. 7a, b). Therefore, fertility/egg production is necessary for AQR, PQR, and URX neurons to be able to control exopheregenesis, however, this regulation is not modulated by embryo-maternal communication. In summary, in the current version of the manuscript we have added data that show how AQR, PQR, and URX regulate exopheregenesis via neuropeptides and restructured our manuscript to present AQR, PQR, and URX role in exopheregenesis more clearly.

63. And, could they assess the same for the effect of male pheromones on exosphere production? Male pheromones increase both egg production and exopher production. Could the authors dissect whether these are linked or not?

Response: Thank you for pointing out the intricate interplay between male pheromones, egg production, and exopher generation in hermaphrodite muscles. We've re-evaluated our data and present the following insights:

1. **Direct influence of male pheromones on exopher production:** As outlined in our results (Section 1), male pheromones indeed escalate exopher production in hermaphrodite muscles. This was evident in our findings from the *him-5* mutant experiments, which showed increased exophers in the presence of males. However, interestingly, the upsurge in exopher numbers is also associated with an elevation in in utero embryos, indicating that male pheromones also play a role in embryo retention in the hermaphrodite's uterus.
2. **Link between egg production and exophers:** As highlighted, while there is a concurrent increase in both exophers and the number of in utero embryos in the presence of male pheromones, the exact relationship remains complex. In experiments involving ascaroside mutants (Section 2), notably *acox-1(ok2257)*, we noticed variations in exopher production without significant shifts in embryo counts in utero. This indicates that exopheregenesis in response to pheromonal stimulation can occur independent of embryo-maternal signaling.
3. **Dissecting male pheromone effects:** On examining the influence of the *ascr#10* male pheromone, we discovered it specifically amplifies exopheregenesis without affecting embryo retention, reinforcing the idea that specific pheromones can modulate exopher production distinctly from their effects on reproduction.

In conclusion, male pheromones can simultaneously stimulate egg retention and exopher formation. However, while often observed together, these two outcomes can also be dissociated under specific conditions or in response to specific pheromonal cues, such as the *ascr#10* male pheromone. We have elaborated on these findings in the revised manuscript to provide a more detailed understanding of this intricate balance. Accordingly, in the revised model, we have included two potential scenarios for male pheromones-mediated exopher production with and without embryo-maternal signaling.

Additional data provided after the revision and not requested by the reviewers:

In the updated manuscript, we have voluntarily incorporated new findings that were not specifically requested by the reviewers. This data showing that AQR, PQR, and URX neurons regulate exophogenesis via URX-expressed FLP-8 and AQR/PQR/URX-expressed neuropeptides. This conclusion is supported by genetic (Fig. 7a, b) as well as optogenetic data (Fig. 7c-e) and enriches the mechanistic underpinnings detailed in this manuscript, allowing for a more comprehensive understanding of the exophogenesis regulation by the nervous system.

REVIEWER COMMENTS

Reviewer #1 (Remarks to the Author):

The authors have responded extensively in a detailed and well argued rebuttal letter. Figure 8 Model and the listed summary around line 323 are major improvements in the revised submission. The S2 Table is also a good addition. ASK-mediated STR-173 ascr#10 and ascr#18 data are quite interesting, with a neuronal signaling component expanded. Overall, the revised version addresses concerns and presents new information on social signaling that influences muscle exopher production, a novel and interesting contribution.

There remain minor elements of the manuscript that should be addressed prior to publication.

****Section on ASCR#10, #18 and ascarosides in general.**

-Writing on the rationale for testing ascaroside roles, ~line 148-170. This section, as written, impressed as confusing -it seems odd that a “looking for male signal” screen is anchored in hermaphrodite observations (hermaphrodite observations for WT having been just discussed and are reported; even with discussion of high male ascr#10, the approach seems unfocused). The initial sticking point is that these are hermaphrodite-hermaphrodite experiments while the intro emphasis was on male outcome— data shows potential, but does not resolve the male signal. The authors should consider alternative presentation: a) test males of the ascaroside biosynthesis mutants initially and lead with that (this is not reported, but would be a strong addition; not essential) or b) begin the lead in to ascarosides from the hermaphrodite side, emphasizing the potential for secreted ascarosides to modulate behavior and continuing from there.

--Figure 3a;3d report that daf-22 mutants, which do not alter exopher levels, do not produce ascr#10 or ascr#18 and yet have little impact on exopher production. Why would the absence of biosynthesis of an important bioactive compound confer no change in the response?? The authors should highlight and discuss the possible reasons. General readers might appreciate a comment on the likely large and systemic impact of deletions, and the differences in chemical addition of individual ascarosides. Chemical supplementation is strong, but should be emphasized to be targeting one of several threads of influence likely operative under normal circumstances; differences from biosynthesis.

****Overall, the identification of ASCR#10 and #18 as having impact on exopher production is a major selling point of the significance of this paper in the revision, and the text section should be reworked to better summarize the complexities.**

Line 115 day1 stage. Add a space

Line 176. Bacterial diet section is a bit jarring to the flow and might be reduced to a conclusion statement or two: *E. coli* variant or food viability had minimal impact on the response to social exposure.

Line 215 or so—considerations on AWC heat sensitivity and CEMs also might be taken out or put into the supplement as they draw off from the main flow a bit.

Line 215 and Fig. 4F, S3A--The ASK, AWB, ADL roles in the male social paradigm appear tightly tied to the egg stimulus, which should be better noted at the end of this paragraph.

Line 493. Begin sentence with *him-5* no capitals and italics, *him-5* later with italics.

Fig. 2a legend should indicate the life stage at which ascarosides were measured in the paper referenced.

Fig. 2g appears to report an experiment of adult with larvae only, possibly suggesting that crowding rather than an adult specific signal is operative. The text explaining this experiment needs to be clarified. Line 137 raising individual adult hermaphrodites in the presence of larvae...

Line 564 statistics suggest that 1nM does influence embryo retention, so title should be reworded.

**Figure 5d –the relationship between the two “null” *str-173* alleles is confusing. *wwa2* appears to be as responsive to males as WT, but *wwa1* does not. This requires some explanation.

Line 612. Sentence begins *str-173* (no capital S)

Line 680. Should be does not

Line 684 *him-5* in italics

Fig S1b is confusing

S6. Legend should note that these are from WT cultures, L4 stage, as AD2 could be quite different.

S7. a and b should have *flp-8* and *-21* in italics as in c.

Strain list As listed, strain TUR61 should also ablate the ASH neuron, but is not listed as doing so in Table 2.

Discussion line 379. Statement that exophers probably act as biological executors and carriers of information in inter-animal communication is not really supported by this work—pheromones, yes, exophers are only characterized for internal activities.

Reviewer #4 (Remarks to the Author):

The manuscript is much better written and the experiments more clearly explained. In addition, there is a clearly laid out mechanistic model underlying exosphere regulation by social cues. Despite this, there are still some concerns that need to be addressed, described below.

- End of section 1 (page 118) they conclude that induction of exopher production by male pheromones occurs through increase in egg retention but this has not been tested at that point. They have not linked or unlinked the two effects: exosphere production and egg retention.

- It seems that the dependency between egg retention and exosphere production is tested later, in section 3. However, it is unclear in this section whether what the authors are testing is male or hermaphrodite pheromone. The section starts addressing the role of male pheromone but then they conclude that it is the hermaphrodite pheromone the one that acts independently of egg production. The authors were already advised in the first review to be more clear to distinguish when they were addressing male or hermaphrodite pheromone. It is very frustrating to see they still have not done this and are still missing it in a very unclear manner.

- In figure 4, are these animals raised in isolation? This should be done to establish first the effect of sensory neurons on basal level of exophere production, independently of con-specific pheromones

- Section 4. The statement “Interestingly, the introduction of male-specific, pheromone-sensing CEM neurons in hermaphrodites (via *ceh-30* gain-of-function mutation²⁵) did not lead to changes in exopher level (Fig. 4c).” needs to be backed up by statistical analysis, which is missing in Fig. 4C

- Similarly, the statement in line 191 “This increase was inhibited in the absence of AWC substantiating the temperature-dependent control of exophergenesis by these neurons (Fig. 4e).” is not backed up by the statistical analysis in which the authors detect a statistically significant increase in exopher production in AWC ablated animals at 25C compared to 20C. Please be more precise in the statement and say the effect is diminished but not abolished.

- Line 270 “This emphasizes the specificity of STR-173 binding to *ascr#10* rather than any other type of ascaroside.” . The authors do not show any experiment to state that STR-173 is the receptor of *Ascr# 10*. That statement is an overinterpretation and I find quite worrying that the authors dare make such a strong statement from a piece of data that shows that effects of the pheromone require the STR-173 in ASK. All the authors can conclude is specificity of STR-173 in mediating the effects of *Ascr# 10*.

- The statement that STR-173 binds *ascr#10* is repeated in the final model, line 328. Please delete as this is by no means demonstrated. To demonstrate receptor- ligand interactions, the authors would need a cellular readout of GPCR signal activation and show that this activation is absolutely dependent on ligand. The data presented in the current manuscript does not rule out a model in which for example STR-173 binds a different molecule, such as a neuropeptide, which modulates ASK neurotransmission to downstream targets.

- It should be clearly stated in the main text that the optogenetic experiments were done at 1 day and 2 days adulthood. Also indicate at which day the data shown in figure 6 is performed. Otherwise, the statement “ This modulation was evident at the adult day 1 (AD1) stage for ReaChr-based activation but not ArchT-based inactivation, underscoring the potential significance of an optimal embryo count or direct uterus interaction “ comes out of nowhere.

- In line 304- “However, their heightened activity could not override the indispensable role of fertility in exophergenesis (Fig. 6h-i).” please rephrase as “However, the increase in exophergenesis induced by these neurons’ inactivation was still dependent on fertility” or something like that

- Placing flp-8 and flp-21 downstream of AQR, PQR and URX activation in the regulation of exopherogenesis is good but they still need to place them in the pathway of socially induced reduction of exopher production. They need to compare the mutants' exopher production in isolation versus grown with 10 hermaphrodites

Reviewer #5 (Remarks to the Author):

As requested by the editor I focussed my evaluation on the responses to reviewer 2. The authors have, in my opinion, adequately addressed the initial concerns from the reviewer.

Reviewer #6 (Remarks to the Author):

In Figure 1 and supp. Fig. 1 the authors show increased embryo production and increased exopher production and embryo retention in hermaphrodites when the hermaphrodites were raised in the presence of male *C. elegans*. Can the authors show images of increased exopher production? All the data indicates number, but one example image or video would help the reader.

In Figure 2 the authors show that increasing worms on the plate or using media conditioned with other hermaphrodites reduces exopher production. They also show no decrease in eggs retained per *C. elegans* indicating that maybe there is no direct correlation between exopher production and eggs retained? This is also highlighted in the next result section.

In Figure 3 the authors show that ascr#10 addition increases exopher production while ascr#18 addition reduces exopher production indicating a complex link between exopher production and ascarosides.

In Supp. Fig. 2 the authors show that HT115 allows for increase in exopher production in comparison to OP50. This result section feels abrupt, and I do not know why this is added as a separate result, the authors could consider either removing this section.

In Figure 4 and Supp Fig. 3 the authors show that a subset of the ciliary neurons that respond to male pheromones also allow for increased exopher formation, and this increase is lost in the absence of some

of these neurons. The authors could consider adding 4 e to supplemental data as it detracts from the main objective of this result section.

Figure 5 and supp. Fig. 4 the authors identify and show the role of str-173 in mediating increased exopher production in the presence of ascr#10. STR-173 potentially functions through the ASK neuron.

In Figure 6,7 and supplemental figs. 5-7 the authors discuss the role of AQR, PQR and URX neuronal activity in regulating exopherogenesis. They show that loss of these neurons/decreased activity of these neurons leads to increased exopher formation while increased neuronal activity in these neurons shows decreased exopher formation. They also show that FLP-8 and FLP-21 from these neurons inhibit exopherogenesis.

The manuscript is well written and the results convincing. The one point that I am very confused about is the relationship between exopherogenesis and eggs retained, this seems like a complicated relationship and could maybe be discussed further. The manuscript flows well till Figure 5 and then becomes fairly complicated especially with the neuropeptides as the receptors for these peptides is not elucidated with respect to exopherogenesis. It also leaves other questions unanswered like is it just neuropeptide or would neurotransmitters from these neurons (AQR and PQR are glutamatergic neurons) also be involved as the ablation/activation experiments would affect aspects of exopherogenesis. Answering these questions would involve neuron specific rescue of the peptides as well as finding the receptors which I realise would be beyond the scope of this manuscript. I would suggest not adding the neuropeptidergic signaling work in this manuscript as it tends to confuse the reader.

Finally, I feel the authors have responded adequately to the comments of the reviewers' in this revised version of the manuscript. As indicated above, I have only minor concerns with the manuscript.

Thank you for the constructive feedback on our manuscript. We have thoroughly addressed the points raised by the reviewers, focusing on clarity and scientific accuracy. Following the reviewer's recommendation, we conducted additional experiments to examine exopher production in *flp-8* and *flp-21* mutants under varying social conditions, including isolation, co-culture with hermaphrodites, and exposure to male pheromones. Our results reveal the pivotal role these neuropeptides play in modulating exopherogenesis in response to different social stimuli and affirm their positioning downstream of the AQR, PQR, and URX neurons. Furthermore, we have clarified the distinct effects of male and hermaphrodite pheromones on exopher production and egg retention, enhancing the manuscript's focus on the interplay between these factors. Finally, in the revised manuscript, we have shifted our emphasis from *ascr#18* to *ascr#3*, following additional experiments and analysis, based on the fact that *ascr#3* is the by far most abundant hermaphrodite-secreted pheromone. Importantly, the effects of *ascr#3* are more consistent and significant compared to *ascr#18*, providing a clearer understanding of the roles of the opposite effects of male- and hermaphrodite-produced pheromones in exopher production in *C. elegans*. Overall, we believe these revisions comprehensively address the reviewers' concerns and significantly enrich our study, providing a more nuanced view of exopherogenesis regulation in *C. elegans*. Please find a detailed description of the edited paragraphs below (the reviewers' comments are in *italics* and our responses are in blue font):

Reviewer #1 (Remarks to the Author):

*The authors have responded extensively in a detailed and well argued rebuttal letter. Figure 8 Model and the listed summary around line 323 are major improvements in the revised submission. The S2 Table is also a good addition. ASK-mediated STR-173 *ascr#10* and *ascr#18* data are quite interesting, with a neuronal signaling component expanded. Overall, the revised version addresses concerns and presents new information on social signaling that influences muscle exopher production, a novel and interesting contribution.*

There remain minor elements of the manuscript that should be addressed prior to publication.

General remarks

Minor points:

****Section on ASCR#10, #18 and ascarosides in general.**

*1. Writing on the rationale for testing ascaroside roles, ~line 148-170. This section, as written, impressed as confusing -it seems odd that a "looking for male signal" screen is anchored in hermaphrodite observations (hermaphrodite observations for WT having been just discussed and are reported; even with discussion of high male *ascr#10*, the approach seems unfocused). The initial sticking point is that these are hermaphrodite-hermaphrodite experiments while the intro emphasis was on male outcome—data shows potential, but does not resolve the male signal. The authors should consider alternative presentation: a) test males of the ascaroside biosynthesis mutants initially and lead with that (this is not reported, but would be a strong addition; not essential) or b) begin the lead in to ascarosides from the hermaphrodite side, emphasizing the potential for secreted ascarosides to modulate behavior and continuing from there.*

Response: Thank you for your insightful comments regarding our manuscript. In the original version, we indeed began exploring the roles of ascarosides in the context of hermaphrodite observations, with a primary focus on the male pheromone *ascr#10*'s stimulatory effect on exopher production in hermaphrodites. We acknowledge that this approach may have seemed unfocused and counterintuitive to the reader. Therefore, as you suggested, we have revised the section to initiate the discussion of ascarosides from the hermaphrodite pheromones side. In the revised manuscript, we specifically tested the involvement of *ascr#3*, one of the main hermaphrodite-enriched ascarosides, in exopher regulation. Our data indicate that *ascr#3* downregulates exopherogenesis,

consistent with findings of diminished exopher production in animals from higher-density populations. Considering the well-studied nature of ascr#3 and its evident role in modulating behavior and physiological responses in the hermaphrodite population, we believe this approach is more logical and reader-friendly than our previous data presentation.

Furthermore, in the original draft, we explored the roles of another hermaphrodite-enriched ascaroside, ascr#18, in hermaphrodite exopher production. However, the effects of ascr#18 on exopher production are much weaker – and less consistent – than the effects of ascr#3, which is also much more abundantly secreted by hermaphrodites than ascr#18. Consequently, we have decided not to include the data for ascr#18 in the current manuscript version and focus on the role of ascr#3 in the regulation of exopher production.

Following the section on ascr#3-dependent decrease in exopher production, we present the results of ascr#10-dependent increase in exopher production. This allows for a clear demonstration that two different ascarosides can differentially regulate exopherogenesis. The paragraph concludes by illustrating the complex interactions between ascaroside biosynthesis and exopherogenesis. For instance, *acox-1(ok2257)* mutants, with increased ascr#10 and notable ascr#3 levels, exhibited enhanced exopher production, suggesting a delicate balance between these ascarosides in regulating exopher dynamics. Conversely, *maoc-1(ok2645)* mutants, characterized by low ascr#10 and absent short-chain ascarosides, demonstrated reduced exopher production, indicating a possible repressive role of e.g., side-chain hydroxylated medium-chain ascarosides. Furthermore, *daf-22(ok693)* mutants, lacking ascarosides like ascr#3 and ascr#10, did not significantly alter exopher production, hinting at potential redundancy in the exopher production pathway. This complex interplay underscores the nuanced roles of ascarosides in *C. elegans*, a key focus of our revised manuscript.

We believe these additions and revisions address your concerns and enhance the manuscript's clarity and focus on ascaroside signaling. Regarding your suggestion to test males of the ascaroside biosynthesis mutants, we agree that this would be a strong addition to our study. While not initially included in our experimental design, we acknowledge its potential to provide more direct insights into the male signaling aspect. We will consider incorporating such experiments in future studies to further elucidate the roles of ascarosides in male signaling.

2. *Figure 3a;3d report that daf-22 mutants, which do not alter exopher levels, do not produce ascr#10 or ascr#18 and yet have little impact on exopher production. Why would the absence of biosynthesis of an important bioactive compound confer no change in the response?? The authors should highlight and discuss the possible reasons. General readers might appreciate a comment on the likely large and systemic impact of deletions, and the differences in chemical addition of individual ascarosides. Chemical supplementation is strong, but should be emphasized to be targeting one of several threads of influence likely operative under normal circumstances; differences from biosynthesis.*

Response: *daf-22* mutant worms produce neither ascr#3 nor ascr#10 and do not impact exopher levels. However, the nematode regulatory network may feature additional levels of complexity, where the absence of one bioactive compound can be mitigated by other components within the system. Furthermore, the systemic effects of gene deletions, such as in *daf-22* mutant, often differ markedly from the impact of chemically supplementing individual ascarosides. Gene deletions can trigger broader compensatory responses within the organism, which may not be apparent when examining the effects of a single, isolated compound. To address this complexity, our manuscript both in results and discussion sections now includes paragraphs on the broader implications of gene deletions and the nuanced differences between genetic and chemical perturbations. This aims to provide readers with a more comprehensive understanding of the multifaceted nature of nematode signaling and regulation.

3. ***Overall, the identification of A SCR#10 and #18 as having impact on exopher production is a major selling point of the significance of this paper in the revision, and the text section should be reworked to better summarize the complexities.*

Response: Following your suggestion, we have revised the section about ascarosides to better summarize the complexities of ascaroside signaling, particularly the roles of ascr#10 and ascr#3

(instead of ascr#18), in modulating exopher production. By focusing on these two ascarosides, our manuscript presents a clearer and more accurate representation of our findings, reflecting the nuanced interplay of these molecules in *C. elegans* biology. This adjustment ensures that our study accurately communicates the most significant and robust aspects of our research, aligning with the overarching objectives of understanding nematode molecular signaling.

4. Line 115 *day1 stage. Add a space.*

Response: We have made appropriate changes in the text.

5. Line 176. *Bacterial diet section is a bit jarring to the flow and might be reduced to a conclusion statement or two: E. coli variant or food viability had minimal impact on the response to social exposure.*

Response: As you suggested, the whole paragraph was reduced to half of the sentence (underlined): “**Growing hermaphrodites on male-conditioned plates increased exopher production (Fig. 1f) to the same degree as when hermaphrodites were grown with males until the L4 larvae stage (Fig. 1c) regardless of the *E. coli* variant used as a food source (Supplementary Fig. 1a).**” Moreover, we have moved Supplementary Fig. 2c to Supplementary Fig. 1a. and Supplementary Fig. 2a-b were removed from the manuscript completely to improve its coherence.

6. Line 215 or so—*considerations on AWC heat sensitivity and CEMs also might be taken out or put into the supplement as they draw off from the main flow a bit.*

Response: As you suggested, we have moved Fig. 4e (data on AWC heat sensitivity) to the supplemental data (Supplementary Fig. 3b) in order to improve manuscript coherence. We decided to keep data on CEM neurons as these results nicely show the complexity of the exopher regulation by the nervous system by highlighting that not only removal but also the addition of one class of sensory neurons has a vast effect on exopher production in response to social cues.

7. Line 215 and Fig. 4F, S3A--*The ASK, AWB, ADL roles in the male social paradigm appear tightly tied to the egg stimulus, which should be better noted at the end of this paragraph.*

Response: Our revised text now explicitly notes the interdependence of these neurons' roles in exopherogenesis with the stimulus provided by male pheromones in terms of egg production. This clarification enhances the understanding of the critical role of these neurons not just in pheromone detection but also in their interaction with reproductive cues, providing a more comprehensive view of the sensory neuron-mediated regulation of exopherogenesis in response to male pheromones.

8. Line 493. *Begin sentence with him-5 no capitals and italics, him-5 later with italics.*

Response: We have made appropriate changes in the text.

9. Fig. 2a legend should indicate the life stage at which ascarosides were measured in the paper referenced.

Response: We have added the information to the Fig. 3e legend that the ascarosides levels in the paper referenced were obtained from mixed-stage worms populations.

10. Fig. 2g appears to report an experiment of adult with larvae only, possibly suggesting that crowding rather than an adult specific signal is operative. The text explaining this experiment needs to be clarified. Line 137 raising individual adult hermaphrodites in the presence of larvae.

Response: Thank you for your feedback regarding the clarity of our experiment description. We have made appropriate changes in the text.

11. Line 564 statistics suggest that *InM* does influence embryo retention, so title should be reworded.

Response: In the revised manuscript, we have removed the data pertaining to ascr#18, since the effects of ascr#3 on exopher production are much more consistent. To keep the paper focused on the largely opposite activities of the male- and hermaphrodite-produced ascr#10 and ascr#3, we also removed mention of the modest effects of 1 nM ascr#18 on embryo retention. These results are interesting starting points for a detailed analysis of potential roles of other, less abundant ascarosides.

12. Line 612. Sentence begins *str-173* (no capital S)

Response: We have made appropriate changes in the text.

13. Line 680. *Should be does not*

Response: We have made appropriate changes in the text.

14. Line 684 *him-5* in italics

Response: We have made appropriate changes in the text.

15. Fig S1b is confusing

Response: We have simplified the scheme to make it easier to understand.

16. S6. Legend should note that these are from WT cultures, L4 stage, as AD2 could be quite different.

Response: We have made appropriate changes in the figure legend.

17. S7. a and b should have *flp-8* and *-21* in italics as in c.

Response: We have made appropriate changes in the figure labeling.

18. Strain list As listed, strain TUR61 should also ablate the ASH neuron, but is not listed as doing so in Table 2.

Response: We have verified the description of TUR61 and can confirm its accuracy. According to the literature, the transgene *qrls2 [sra-9::mCasp1]* (originally from the PS6025 strain available from CGC) causes genetic ablation of ASK neurons only. It has been used multiple times for this purpose in the literature (e.g., Srinivasan et al., PLOS Biology 2012; Ludewig et al., Nat Chem Biol. 2020; Huang et al., Current Biology 2023).

19. Discussion line 379. Statement that *exophers probably act as biological executors and carriers of information in inter-animal communication is not really supported by this work—pheromones, yes, exophers are only characterized for internal activities.*

Response: We acknowledge your observation that our current work primarily characterizes exophers for their internal activities and does not directly support the notion of exophers acting as biological executors and carriers of information in inter-animal communication.

In response, we have revised the relevant section of our manuscript to more accurately reflect the findings of our study. The revised text emphasizes that while our study indicates exopher production is influenced by external cues, particularly pheromones, it does not establish a direct role for exophers in inter-animal communication. We have also clarified that the potential for such a role, akin to EVs in other species like *Drosophila*, remains an open question for future research.

Reviewer #4 (Remarks to the Author):

The manuscript is much better written and the experiments more clearly explained. In addition, there is a clearly laid out mechanistic model underlying exophere regulation by social cues. Despite this, there are still some concerns that need to be addressed, described below.

20. End of section 1 (page 118) they conclude that induction of exopher production by male pheromones occurs through increase in egg retention but this has not been tested at that point. They have not linked or unlinked the two effects: exophere production and egg retention.

Response: In response to your comment regarding the conclusion at the end of section 1 about the induction of exopher production by male pheromones, we have reviewed our manuscript and data to ensure accuracy in our statements. In the revised version of the manuscript, we have clearly stated in the discussion, that while the *ascr#10*-mediated increase in exopher production demonstrates that male pheromones can upregulate exophogenesis without the involvement of embryo accumulation *in utero*, additional data from male-conditioned plates show that male pheromones can also induce embryo retention in the uterus. This also suggests that *ascr#10* is just

one among multiple small molecule signals secreted by males that influence exopher formation, potentially with or without engaging embryo-maternal signaling pathways.

21. *It seems that the dependency between egg retention and exosphere production is tested later, in section 3. However, it is unclear in this section whether what the authors are testing is male or hermaphrodite pheromone. The section starts addressing the role of male pheromone but then they conclude that it is the hermaphrodite pheromone the one that acts independently of egg production. The authors were already advised in the first review to be more clear to distinguish when they were addressing male or hermaphrodite pheromone. It is very frustrating to see they still have not done this and are still missing it in a very unclear manner.*

Response: Regarding the clarity in distinguishing between male and hermaphrodite pheromones in our study, we have taken your feedback into consideration and revised the relevant sections. In the original version, we indeed initiated our exploration into ascaroside roles in the context of hermaphrodite observations, primarily focused on *ascr#10*'s stimulatory effect on exopher production in hermaphrodites. We recognize that this approach may have appeared unfocused, especially given the earlier emphasis on male signaling outcomes. To address this, we have revised the section to begin with the differential roles of *ascr#3* and *ascr#10*, secreted by hermaphrodites and males, respectively, in modulating behavior and physiological responses in the nematode population. We believe this revised introduction provides a more logical segue into the specific experiments and findings related to ascarosides. Furthermore, we have incorporated additional experimental observations to clarify the role of *ascr#3* and *ascr#10* in exopher production. This includes details about the developmental stage-specific responses to ascaroside exposure and the effects of different ascaroside concentrations. Additionally, we have included detailed observations from our studies on *C. elegans* mutants with disrupted ascaroside biosynthesis, particularly in peroxisomal β -oxidation. Our data reveal complex interactions between ascaroside biosynthesis and exophogenesis. For instance, *acox-1(ok2257)* mutants, with increased *ascr#10* and notable *ascr#3* levels, exhibited enhanced exopher production, suggesting a delicate balance between these ascarosides in regulating exopher dynamics. Conversely, *maoc-1(ok2645)* mutants, characterized by low *ascr#10* and absent short-chain ascarosides, demonstrated reduced exopher production, indicating a possible repressive role of e.g., side-chain hydroxylated medium-chain ascarosides. Furthermore, *daf-22(ok693)* mutants, lacking ascarosides like *ascr#3* and *ascr#10*, did not significantly alter exopher production, hinting at potential redundancy in the exopher production pathway. This complex interplay underscores the nuanced roles of ascarosides in *C. elegans*, a key focus of our revised manuscript.

We are fully aware that you suggested a complete distinction between male pheromone-dependent and hermaphrodite pheromone-dependent regulation of exopher production. We have attempted various approaches to implement this distinction. However, due to shared elements in certain pathways, such as neuronal regulation, we would either need to repeat information or refer back to data presented much earlier in the text. After presenting the data and text in this manner, we observed that it unnecessarily complicates the narrative and adversely affects its readability. Consequently, we have made efforts to separate the two pathways as much as possible and have further clarified the sections as per your suggestion.

22. *In figure 4, are these animals raised in isolation? This should be done to establish first the effect of sensory neurons on basal level of exosphere production, independently of con-specific pheromones*

Response: Thank you for your inquiry regarding the conditions under which the animals were raised for the experiments presented in Figure 4c. To clarify, the animals were not raised in isolation but in a population of approximately 50 age-synchronized hermaphrodites. Following your suggestion to assess the basal level of exopher production independently of conspecific pheromones, we have extracted the data for solitary animals from Figure 4e and presented them as a separate Supplementary Figure 3a to directly compare the data for different neuronal mutants. The results obtained for solitary animals align with the data for animals grown in a population of 50 hermaphrodites, further solidifying our conclusions regarding the involvement of certain sensory neurons in exophogenesis regulation.

23. Section 4. The statement “Interestingly, the introduction of male-specific, pheromone-sensing CEM neurons in hermaphrodites (via *ceh-30* gain-of-function mutation²⁵) did not lead to changes in exopher level (Fig. 4c).” needs to be backed up by statistical analysis, which is missing in Fig. 4C.

Response: We appreciate your attention to the necessity of statistical backing for the observations in Figure 4c. In response, we have now included a result of statistical analysis to support the statement regarding the effects of the *ceh-30* gain-of-function mutation on exopher levels.

24. Similarly, the statement in line 191 “This increase was inhibited in the absence of AWC substantiating the temperature-dependent control of exopherogenesis by these neurons (Fig. 4e).” is not backed up by the statistical analysis in which the authors detect a statistically significant increase in exopher production in AWC ablated animals at 25°C compared to 20°C. Please be more precise in the statement and say the effect is diminished but not abolished.

Response: Thank you for pointing out the need for precision in our statement regarding the role of AWC neurons in temperature-dependent control of exopherogenesis. We have revised the statement to accurately reflect that the increase in exopher production at 25°C, compared to 20°C, in AWC-ablated animals is statistically significant but diminished rather than completely abolished.

25. Line 270 “This emphasizes the specificity of STR-173 binding to *ascr#10* rather than any other type of”. The authors do not show any experiment to state that STR-173 is the receptor of *ascr#10*. That *ascr#10* is an overinterpretation and is quite worrying that the authors dare make such a strong statement from a piece of data that shows that effects of the pheromone require the STR-173 in ASK. All the authors can conclude is specificity of STR-173 in mediating the effects of *ascr#10*.

The statement that STR-173 binds *ascr#10* is repeated in the final model, line 328. Please delete as this is by no means demonstrated. To demonstrate receptor-ligand interactions, the authors would need a cellular readout of GPCR signal activation and show that this activation is absolutely dependent on ligand. The data presented in the current manuscript does not rule out a model in which for example STR-173 binds a different molecule, such as a neuropeptide, which modulates ASK neurotransmission to downstream targets

Response: We acknowledge that our experimental setup and data do not directly demonstrate STR-173 as the receptor for *ascr#10*, but rather indicate the specificity of STR-173 in mediating the effects of *ascr#10*. The conclusion that STR-173 binds *ascr#10* was inferred from the observed phenotypic outcomes, which may not conclusively establish a direct receptor-ligand relationship. In light of your feedback, we have revised our manuscript to more accurately reflect the nature of our findings. We have removed the statements suggesting direct binding of *ascr#10* to STR-173 and have rephrased these sections to focus on the specificity of STR-173 in mediating the effects of *ascr#10*, without implying direct receptor-ligand interaction. Additionally, we agree that demonstrating a receptor-ligand interaction requires specific experimental evidence, such as a cellular readout of GPCR signal activation that is dependent on the ligand. Our study does not include such direct evidence and, as you rightly pointed out, does not exclude other models where STR-173 might interact with different molecules that modulate ASK neurotransmission.

Response: We acknowledge that our experimental setup and data do not directly demonstrate STR-173 as the receptor for *ascr#10*, but rather indicate the specificity of STR-173 in mediating the effects of *ascr#10*. The conclusion that STR-173 binds *ascr#10* was inferred from the observed phenotypic outcomes, which may not conclusively establish a direct receptor-ligand relationship.

In light of your feedback, we have revised our manuscript to more accurately reflect the nature of our findings. We have removed the statements suggesting direct binding of *ascr#10* to STR-173 and have rephrased these sections to focus on the specificity of STR-173 in mediating the effects of *ascr#10*, without implying direct receptor-ligand interaction.

Additionally, we agree that demonstrating a receptor-ligand interaction requires specific experimental evidence, such as a cellular readout of GPCR signal activation that is dependent on the ligand. Our study does not include such direct evidence and, as you rightly pointed out, does not exclude other models where STR-173 might interact with different molecules that modulate ASK neurotransmission.

26. It should be clearly stated in the main text that the optogenetic experiments were done at 1 day and 2 days adulthood. Also indicate at which day the data shown in figure 6 is performed. Otherwise, the statement “This modulation was evident at the adult day 1 (AD1) stage for ReaChr-based activation but not ArchT-based inactivation, underscoring the potential significance of an optimal embryo count or direct uterus interaction” comes out of nowhere.

Response: In response to your feedback, we have revised the main text to explicitly state that the optogenetic experiments were performed at both day 1 and day 2 of adulthood. Additionally, we have now clearly indicated in the manuscript the specific day on which the data shown in Figure 6 was obtained.

We have also refined the statement regarding the modulation observed at the adult day 1 (AD1) stage for ReaChr-based activation and ArchT-based inactivation. The revised text provides context for this observation, linking it to the specific timing of the experiments and the relevance of the embryo count or direct uterus interaction at this developmental stage.

27. In line 304- “However, their heightened activity could not override the indispensable role of fertility in exophogenesis (Fig. 6h-i).” please rephrase as “However, the increase in exophogenesis induced by these neurons’ inactivation was still dependent on fertility” or something like that.

Response: We have made appropriate changes in the text.

28. Placing *flp-8* and *flp-21* downstream of AQR, PQR and URX activation in the regulation of exophogenesis is good but they still need to place them in the pathway of socially induced reduction of exopher production. They need to compare the mutants’ exopher production in isolation versus grown with 10 hermaphrodites

Response: Thank you for emphasizing the need to clarify the role of FLP-8 and FLP-21 in regulating exopher production in response to social cues. In response to your suggestion, we have conducted comprehensive experiments to assess exopher production in *flp-8* and *flp-21* mutants, both in isolation and when co-cultured with other hermaphrodites. To maintain consistency with other data in the manuscript, we have also measured exopher production in *flp-8* and *flp-21* mutants after exposing them to male pheromones. Our results reveal that the removal of FLP-8 and FLP-21 does not lead to a reduction in exopher levels in hermaphrodite-dense environments. This response aligns with observations in strains devoid of AQR/PQR/URX neurons. Furthermore, the absence of these neuropeptides did not inhibit the increased exophogenesis observed in a reporter strain cultured on male-conditioned plates, a response similar to observations in worms devoid of AQR/PQR/URX neurons. These findings indicate that FLP-8 and FLP-21, released by AQR/PQR/URX neurons, are integral in regulating exophogenesis, particularly in response to hermaphrodite signals.

Reviewer #5 (Remarks to the Author):

29. As requested by the editor I focused my evaluation on the responses to reviewer 2. The authors have, in my opinion, adequately addressed the initial concerns from the reviewer.

Response: We are grateful for your time and effort in evaluating our manuscript, particularly in the context of the concerns raised by reviewer 2.

Reviewer #6 (Remarks to the Author):

Minor points:

30. In Figure 1 and supp. Fig. 1 the authors show increased embryo production and increased exopher production and embryo retention in hermaphrodites when the hermaphrodites were raised in the presence of male *C. elegans*. Can the authors show images of increased exopher production? All the data indicates number, but one example image or video would help the reader.

Response: In response to your comment, we have included representative images in the revised manuscript to visually demonstrate the increased exopher production. Moreover, we have also included representative images for decreased exopher production when worms were grown in a higher population density. These images complement the quantitative data presented in Figure 1, Figure 2, and Supplementary Figure 1 offering a more comprehensive understanding of the phenomena observed. We also believe that these visual additions will greatly enhance the reader's grasp of the significant increase in exopher production under the specified conditions.

31. In Supp. Fig. 2 the authors show that HT115 allows for increase in exopher production in comparison to OP50. This result section feels abrupt, and I do not know why this is added as a separate result, the authors could consider either removing this section.

Response: Following the reviewers’ suggestion, we have reduced the whole paragraph to half of the sentence (underlined): “Growing hermaphrodites on male-conditioned plates increased exopher production (Fig. 1f) to the same degree as when hermaphrodites were grown with males until the L4 larvae stage (Fig. 1c) regardless of the *E. coli* variant used as a food source (Supplementary Fig. 1a).” Moreover, we have moved Supplementary Fig. 2c to Supplementary Fig. 1a. and Supplementary Fig. 2a-b were removed from the manuscript completely to improve its coherence.

32. In Figure 4 and Supp Fig. 3 the authors show that a subset of the ciliary neurons that respond to male pheromones also allow for increased exopher formation, and this increase is lost in the absence of some of these neurons. The authors could consider adding 4 e to supplemental data as it detracts from the main objective of this result section.

Response: As you suggested, we have moved Fig. 4e to the supplemental data (Supplementary Fig. 3b) in order to improve manuscript coherence.

33. *The manuscript is well written and the results convincing. The one point that I am very confused about is the relationship between exophergenesis and eggs retained, this seems like a complicated relationship and could maybe be discussed further.*

Response: In response to your comment, we have thoroughly reviewed and revised our manuscript to clarify these aspects. Our investigations using *C. elegans* mutants with disrupted ascaroside biosynthesis, specifically those like *acox-1(ok2257)* which have elevated levels of *ascr#10*, revealed intricate interactions between ascaroside biosynthesis and exophergenesis. These findings suggest a significant role for *ascr#10* in enhancing exopher production. In the revised version of the manuscript, we have clearly stated in the discussion, that while the *ascr#10*-mediated increase in exopher production demonstrates that male pheromones can upregulate exophergenesis without the involvement of embryo accumulation *in utero*, additional data from male-conditioned plates show that male pheromones can also induce embryo retention in the uterus. This also suggests that *ascr#10* is just one among multiple small molecule signals secreted by males that influence exopher formation, potentially with or without engaging embryo-maternal signaling pathways.

In addition, we have taken steps to clearly distinguish between the effects of male and hermaphrodite pheromones throughout our study. Our data indicate that the modulation of exophergenesis by hermaphrodite pheromones is largely independent of embryo-maternal signaling pathways. The revised manuscript now distinctly outlines when we are discussing the effects of male pheromones in general, *ascr#10* specifically, or hermaphrodite pheromones (such as *ascr#3*) regarding their influence on exopher production and egg retention. We hope these revisions address your concerns and provide a clearer understanding of the complex dynamics between exopher production, egg retention, and ascaroside signaling in *C. elegans*.

34. *The manuscript flows well till Figure 5 and then becomes fairly complicated especially with the neuropeptides as the receptors for these peptides is not elucidated with respect to exophergenesis. It also leaves other questions unanswered like is it just neuropeptide or would neurotransmitters from these neurons (AQR and PQR are glutamatergic neurons) also be involved as the ablation/activation experiments would affect aspects of exophergenesis. Answering these questions would involve neuron specific rescue of the peptides as well as finding the receptors which I realise would be beyond the scope of this manuscript. I would suggest not adding the neuroptidergic signaling work in this manuscript as it tends to confuse the reader.*

Response: We appreciate your suggestion of removing the neuroptidergic signaling data from the manuscript. However, after careful consideration, we have decided to keep these data in the manuscript as in our opinion they allow for a more comprehensive understanding of the exophergenesis regulation by the nervous system.

REVIEWERS' COMMENTS

Reviewer #1 (Remarks to the Author):

The authors have revised in response to previous review, with some shift in emphasis on specific ascarosides of focus. The paper reads with improved accessibility and rationale. The story is complex and the discussion is a bit long and speculative (for example muscle excitability and locomotion extend out from the focus here), but the discussion will probably have to be trimmed for length considerations in final revision. Overall, data on potential links of exopher production and social pheromone signaling, and associated molecular and cellular player identification, are of high interest in the field.

Typographical issues:

Line 36. EV should be in the singular

Line 83 exopher should be in the singular

Line 496. More clear: him-5 mutant hermaphrodites and him-5 mutant males were co-cultured from the L1 stage until the L4 stage and then hermaphrodites were transferred to a male-free plate.

Line 146 decreased

Line 208 independently

Line 221 che-13 should be italicized

231 ...under higher population density (Fig. 4c) but caused a decrease in exopher production when animals were grown in isolation.

261 pheromone sensing

Figure 3e wild type pie has extra lines between similar colors—better to eliminate those.

574 worm

726 accessibility

731 10-hermaphrodite populations

741 period

762 influences

Reviewer #4 (Remarks to the Author):

Finally, this has become a quite complete story.

The authors may want to consider improving the clarity of some sections, particularly the one addressing the results in Figure 4. The interpretation of this data and conclusions extracted, although correct, are hard to follow in the way they are currently written.

Reviewer #6 (Remarks to the Author):

The authors have sufficiently addressed my concerns. I am fine with the manuscript being accepted for publication.

We would like to express our appreciation for the thoughtful and constructive feedback provided on our manuscript. The positive evaluation and subsequent acceptance, in principle, for publication have been both encouraging and deeply gratifying. We are particularly grateful for the Reviewers' attention, time, and insightful criticism, which have been instrumental in elevating the quality and maturity of our manuscript. We have addressed the remaining concerns as follows:

Reviewer #1:

1. We acknowledge the Reviewer concern regarding the length and speculative nature of our discussion. After careful consideration, we have decided to retain the current form of the discussion. We believe that the detailed narrative is essential to fully convey the complexity and implications of our findings, fostering a comprehensive understanding among a broad audience and stimulating further research in this area.
2. We have meticulously addressed each of the typographical and technical errors highlighted:
 - Corrections made on lines 36, 83, 496, 146, 208, 221, 231, 261, 574, 726, 731, 741, as specified.
 - In Figure 3e there are no extra lines that has to be removed. In WT pie chart line between blue colors separate ascr#10 from other saturated ascarosides while line between red colors separate ascr#3 from α , β -unsaturated ascarosides.

Reviewer #4:

While we are thankful for the Reviewer suggestions for improving the clarity of the section discussing Figure 4, we have opted to retain the current presentation. We believe the existing structure and narrative best convey the intricate details and significance of our findings, providing a balanced and accessible explanation suitable for both expert and broader audiences.

Reviewer #6:

We are pleased that the Reviewer concerns have been sufficiently addressed and appreciate the support for our manuscript's acceptance for publication.